# Sec61β maintains cytoplasmic proteostasis via ARIH1-mediated translational repression upon ER stress

Hisae Kadowaki [1✉], Tomohisa Hatta[2], Kazuma Sugiyama[1], Tomohiro Fukaya[3], Takao Fujisawa [4], Takashi Hamano [5], Naoya Murao [1], Yasunari Takami[1], Shuya Mitoma [3], Tohru Natsume[6], Katsuaki Sato[3], Hiromi Hirata [5], Tamayo Uechi[7] & Hideki Nishitoh [1,8✉]

## Abstract

Disrupted proteostasis causes various degenerative diseases, and organelle homeostasis is therefore maintained by elaborate mechanisms. Endoplasmic reticulum (ER) stress-induced preemptive quality control (ERpQC) counteracts stress by reducing ER load through inhibiting the translocation of newly synthesized proteins into the ER for their rapid degradation in the cytoplasm. Here, we show that Sec61β, a translocon component, prevents the overproduction of ERpQC substrates, allowing for their efficient degradation by the proteasome. Sec61β inhibits the binding of translation initiation factor eIF4E to the mRNA 5′ cap structure by recruiting E3 ligase ARIH1 and eIF4E-homologous protein 4EHP, resulting in selective translational repression of ERpQC substrates. Sec61β deficiency causes overproduction of ERpQC substrates and reduces proteasome activity, leading to cytoplasmic aggresome formation. We also show that Sec61β deficiency causes motor dysfunction in zebrafish, which is restored by exogenous ARIH1 expression. Collectively, translational repression of ERpQC substrates by the Sec61β–ARIH1 complex contributes to maintain ER and cytoplasmic proteostasis.

**Keywords** ERpQC; ER Stress; Proteostasis; Translational Regulation
**Subject Categories** Post-translational Modifications & Proteolysis; Translation & Protein Quality

## Introduction

Approximately one-third of newly synthesized proteins are translocated into the endoplasmic reticulum (ER) by cotranslational or posttranslational targeting via the translocon in cells (Rapoport, 2007). During cotranslational translocation, the signal sequence of nascent polypeptides emerging from the ribosome is recognized by the signal recognition particle (SRP), and the ribosome-nascent chain complex (RNC) docks onto the SRP receptors (SRα/β) near the translocon (Keenan et al, 2001). When the SRP dissociates from the RNC, the newly synthesized polypeptide is delivered into the ER (Mandon et al, 2009). Translocated polypeptides are correctly folded in the ER and either secreted or transported to the membrane compartments (Braakman and Hebert, 2013). Disturbed ER homeostasis due to environmental changes or excessive protein folding load leads to the accumulation of unfolded proteins in the ER, causing ER stress. Cells attempt to restore ER homeostasis by two distinct strategies (Hetz and Papa, 2018; Kadowaki and Nishitoh, 2013; Walter and Ron, 2011). One is the restoration of ER folding capacity by reducing unfolded proteins. To this end, unfolded proteins are refolded by ER chaperones or degraded by the ubiquitin-proteasome system (UPS) via ER-associated degradation (ERAD) or by selective autophagy in the ER (Bhattacharya and Qi, 2019; Christianson and Carvalho, 2022; Hetz et al, 2020; Kadowaki and Nishitoh, 2019; Krshnan et al, 2022; Wodrich et al, 2022). Another strategy is reducing the protein transport load into the ER. To achieve this, novel protein synthesis in the ER is inhibited by translational attenuation (Harding et al, 1999) and selective degradation of ER-associated mRNA (Hollien et al, 2009; Ottens et al, 2024) through the activation of the ER stress sensors PERK and IRE1, respectively. However, protein synthesis in the ER is not completely halted by translational attenuation and mRNA degradation, and some of the ER-targeting proteins continue to be translated (Kang et al, 2006). This is

[1]Laboratory of Biochemistry and Molecular Biology, Faculty of Medicine, University of Miyazaki, 5200 Kihara, Kiyotake, Miyazaki 889-1692, Japan. [2]Robotic Biology Institute Inc., 1-5-45 Yushima, Bunkyo-ku, Tokyo 113-8510, Japan. [3]Division of Immunology, Department of Infectious Diseases, Faculty of Medicine, University of Miyazaki, 5200 Kihara, Kiyotake, Miyazaki 889-1692, Japan. [4]Laboratory of Cell Signaling, Graduate School of Pharmaceutical Sciences, The University of Tokyo, Tokyo 113-0033, Japan. [5]Department of Chemistry and Biological Science, College of Science and Engineering, Aoyama Gakuin University, Sagamihara 252-5258, Japan. [6]Cellular and Molecular Biotechnology Research Institute, National Institute of Advanced Industrial Science and Technology, 2-3-26 Aomi, Koto-ku, Tokyo 135-0064, Japan. [7]Laboratory of Medical Biology, Faculty of Medicine, University of Miyazaki, 5200 Kihara, Kiyotake, Miyazaki 889-1692, Japan. [8]Frontier Science Research Center, University of Miyazaki, 5200 Kihara, Kiyotake, Miyazaki 889-1692, Japan. ✉E-mail: kadowaki@med.miyazaki-u.ac.jp; nishitoh@med.miyazaki-u.ac.jp

reasonable since the expression of ER chaperones or ERAD-related molecules should be induced under stress conditions. Conversely, ER transport of proteins unrelated to ER quality control, such as secretory proteins, overloads the ER and reduces the efficiency of the unfolded protein response. It has been reported that restricting the translocation of such ER-targeting proteins into the ER contributes to maintaining ER homeostasis (Braunstein et al, 2015; Glinka et al, 2014; Kang et al, 2006; Kim et al, 2013; Orsi et al, 2006; Rane et al, 2008; Rutkowski et al, 2007).

We previously reported the physiological role and molecular mechanisms of ER stress-induced preemptive quality control (ERpQC), a load-reduction system of ER-targeting proteins associated with cotranslational translocation (Kadowaki et al, 2015; Kadowaki et al, 2018). In ERpQC, several secretory proteins possessing signal sequences change their location of synthesis from the ER to the cytoplasm and are immediately degraded by the UPS during ER stress. In response to ER stress, ERpQC is triggered by the recruitment of Derlin family proteins (Derlins; Derlin-1, Derlin-2, and Derlin-3 in mammals), which are components of the ERAD complex, to the translocon and SRα/β (Kadowaki et al, 2015). Derlins interact with the 54-kDa subunit of the SRP (SRP54) on the RNC through their carboxyl (C) terminus, and newly synthesized polypeptides emerging from ribosome are rerouted from the ER translocation pathway to the cytoplasmic degradation pathway (rerouting step). Thus, Derlins function as factors in determining the fate of ER-targeting proteins during ER stress. Rerouted newly synthesized polypeptides (ERpQC substrates) are ubiquitinated by the ER membrane-localized E3 ligase HRD1, which forms a large ERpQC complex with the translocon and Derlin-1 during ER stress (Kadowaki et al, 2018), and transported to the proteasome by the AAA-ATPase p97 and the cytoplasmic chaperone Bag6 (degradation step) (Kadowaki et al, 2015). This ERpQC-mediated degradation system is also called cotranslational degradation (Oyadomari et al, 2006). The ERpQC substrates exhibit selectivity, and ER chaperones such as BiP, PDI, and GRP94 evade capture by Derlins even under ER stress conditions and translocate into the ER, where they refold unfolded proteins.

Here, we show the novel role of Sec61β, a translocon component that binds to Derlins. Sec61β selectively inhibits the translation of ERpQC substrate mRNAs via the recruitment of the RING-in-between-RING (RBR) E3 ligase Ariadne homologue 1 (ARIH1) and the mRNA cap-binding eukaryotic translation initiation factor (eIF) 4E-homologous protein (4EHP). Disruption of this translational repression due to Sec61β dysfunction risks reducing cytoplasmic proteostasis and increasing aberrant protein aggregation. Thus, Derlins–Sec61β–ARIH1 complex formation ensures the strict translational repression of ERpQC substrates captured by Derlins, mitigating the burden of proteasomal degradation and ultimately maintaining cytoplasmic proteostasis. Such a rigorous quality control system with ERpQC is crucial for maintaining proteostasis for the cytoplasm as well as the ER.

## Results

### Derlins interact with translation-related proteins

Newly synthesized ERpQC substrates are rerouted from the ER to the cytoplasm via the interaction of Derlins with the SRP–SR complex, and

effectively degraded by the UPS via HRD1, Bag6, and p97 (Kadowaki et al, 2015; Kadowaki et al, 2018). To examine the detailed mechanism of Derlins-mediated ERpQC, we performed mass spectrometry (MS)-based proteomics analysis using Flag-tagged Derlin-1, -2, and -3 as bait. Pulldown assays identified that 135 molecules increased their binding to Derlin-1, -2, or -3 in an ER stress-dependent manner (Dataset EV1). The Gene Ontology molecular function analysis revealed that Derlins unexpectedly interacted with RNA-binding molecules in addition to protein-binding molecules in the ER stressor thapsigargin (Tg; an inhibitor of the sarco-ER $Ca^{2+}$ ATPase)-treated cells (Fig. 1A; Dataset EV1). Interactions of RNA-binding molecules with Derlins were also observed in cells treated with the proteasome inhibitor MG132 in addition to Tg (Fig. EV1A). Since capture of ERpQC substrates by Derlins may be coupled with cotranslational translocation, we focused on RNA-binding molecules, specifically the translocon component Sec61β and the RNA helicase eIF4A1, a component of the translation initiation factor complex eIF4F that plays a central role in cap-dependent translation initiation (Dataset EV1). In HEK293 cells co-expressing exogenous Sec61β and Derlins, Sec61β interacted with Derlin-1, -2, and -3, and treatment with Tg and MG132 enhanced the binding of Sec61β with Derlin-2 and -3 (Fig. 1B). Although exogenous Derlin-1 interacted with exogenous Sec61β independently of ER stress, an ER stress-induced endogenous interaction between Derlin-1 and Sec61β was observed in human hepatoma HepG2 cells, which actively synthesize secretory proteins (Fig. 1C). These results suggest that Sec61β physiologically interacts with Derlins in an ER stress-dependent manner. Next, since eIF4A1, together with the cap-binding protein eIF4E and the scaffolding protein eIF4G, forms eIF4F that plays a crucial role in translation initiation, we hypothesized that eIF4F might be recruited to Derlins. To test this hypothesis, we examined whether both eIF4A1 and eIF4E interact with Derlins. Endogenous eIF4A1 interacted with exogenous Derlin-1, Derlin-2, and Derlin-3, and treatment with Tg and MG132 enhanced the interactions of eIF4A1 with Derlin-2 and -3 in HEK293 cells (Fig. EV1B). The physiological interactions of endogenous eIF4A1 with Derlin-1 and Derlin-2 were confirmed in HepG2 cells (Fig. EV1C,D). Endogenous eIF4E also interacted with exogenous Derlins in an ER stress-dependent manner (Fig. EV1B). ER stress-dependent interaction of endogenous eIF4E with endogenous Derlin-2 was also confirmed (Fig. EV1D), but not with endogenous Derlin-1 due to experimental limitations. However, ER stress-dependent interaction between exogenous eIF4E and endogenous Derlin-1 was detected (Fig. EV1E). Derlin-3, unlike Derlin-1 and -2, is expressed only in limited cell types and antibodies recognizing endogenous Derlin-3 were not available. In any case, our findings suggest that eIF4F (composed of eIF4A, eIF4E, and eIF4G) is physiologically recruited to Derlins in an ER stress-dependent manner. We have previously reported that Derlins interact with Sec61α during ER stress (Kadowaki et al, 2015) and that SRP binding to the C-terminal cytoplasmic domain of Derlins is required for ERpQC (Kadowaki et al, 2018). Taken together, it is possible that the RNC of ERpQC substrate including eIF4F and SRP is captured by Derlins and its translation is initiated in the vicinity of Derlins–Sec61β complex.

### Sec61β reduces the accumulation of ERpQC substrates

The translocon comprises Sec61α, β, and γ. Sec61α and γ are essential for the translocation of ER-targeting proteins across the

A

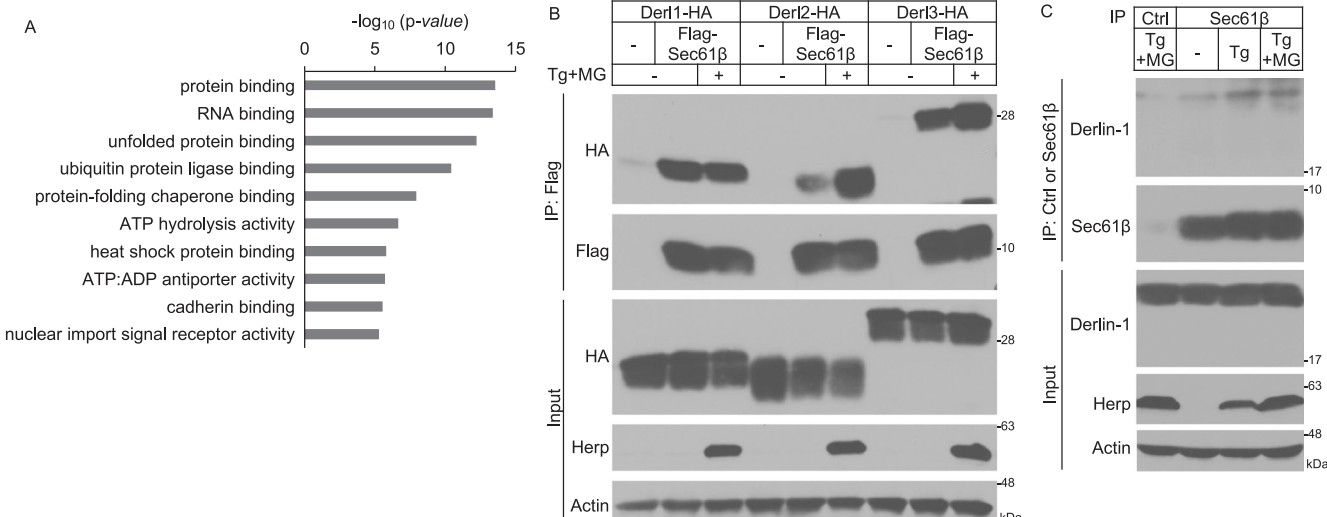

-log₁₀ (p-value)

protein binding
RNA binding
unfolded protein binding
ubiquitin protein ligase binding
protein-folding chaperone binding
ATP hydrolysis activity
heat shock protein binding
ATP:ADP antiporter activity
cadherin binding
nuclear import signal receptor activity

D

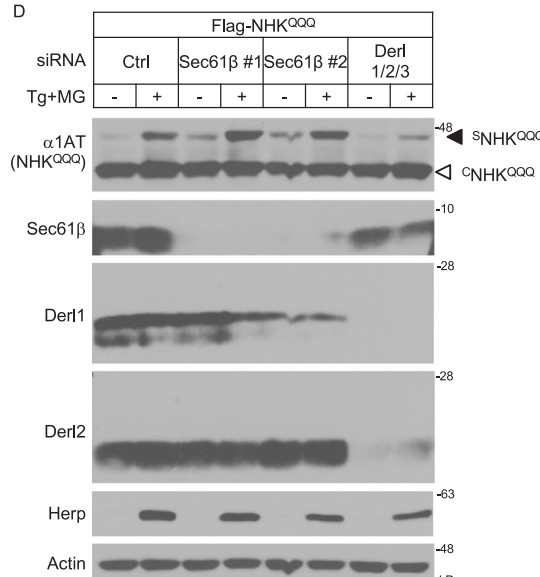

E

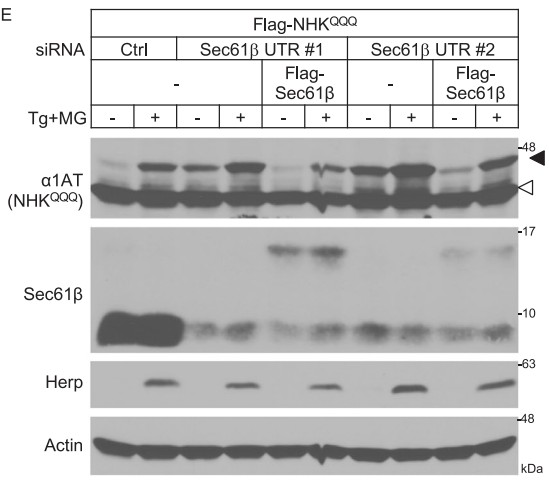

F

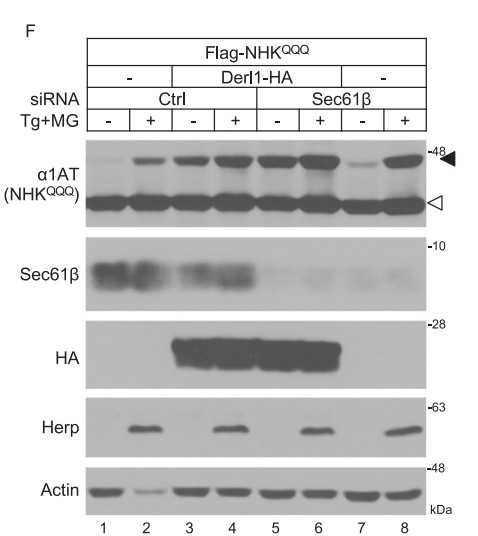

G

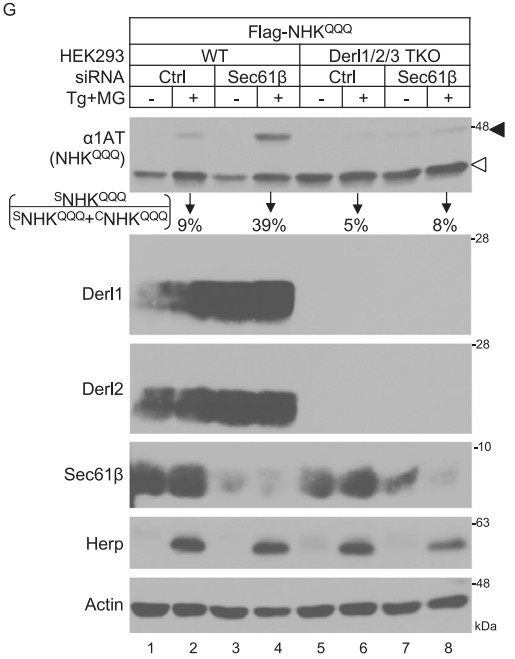

◄ **Figure 1. Interactions between Sec61β and Derlins and inhibition of ERpQC substrate accumulation by Sec61β.**

(A) Gene Ontology (GO) analysis of Derlins-interacting proteins by thapsigargin (Tg) treatment. The bar graph shows the top 10 GO molecular function terms with a false discovery rate of <0.05 calculated from the DAVID online tool. $P$ values were calculated using the modified Fisher's exact test implemented in DAVID; 37 proteins identified as RNA binding in terms of molecular function are listed in Dataset EV1. (B) Interactions of Derlins with Sec61β. HEK293 cells transfected with indicated plasmids and treated with or without 50 nM Tg and 200 nM MG132 for 16 h were immunoprecipitated (IPed) with an anti-Flag antibody and immunoblotted with indicated antibodies. (C) Endogenous interaction of Derlin-1 with Sec61β. Immunoprecipitation (IP) with anti-Sec61β antibody or control (Ctrl) IgG using Protein G Sepharose and immunoblotting (IB) with indicated antibodies in HepG2 cells treated with or without 200 nM Tg and/or 500 nM MG132 for 16 h. (D–F) IB of ERpQC substrates in HEK293 cells transfected with indicated siRNAs and plasmids and treated with or without 50 nM Tg and 200 nM MG132 for 16 h. All samples were immunoblotted with indicated antibodies. Black arrowhead, signal peptide-uncleaved NHK$^{QQQ}$ ($^S$NHK$^{QQQ}$); white arrowhead, signal peptide-cleaved NHK$^{QQQ}$ ($^C$NHK$^{QQQ}$). (G) IB of ERpQC substrate in wild-type (WT) or *Derlin-1, -2,* and *-3* triple knockout (TKO) HEK293 cells transfected with indicated siRNAs and plasmid for NHK$^{QQQ}$ and treated with or without 50 nM Tg and 200 nM MG132 for 16 h. All samples were immunoblotted with indicated antibodies. Expression levels of $^S$NHK$^{QQQ}$ were calculated and shown as the percentage of $^S$NHK$^{QQQ}$ out of the total amount of NHK$^{QQQ}$ ($^S$NHK$^{QQQ}$ and $^C$NHK$^{QQQ}$). Black arrowhead, $^S$NHK$^{QQQ}$; white arrowhead, $^C$NHK$^{QQQ}$. Source data are available online for this figure.

ER membrane, whereas Sec61β is not (Kalies et al, 1998). Sec61β is located opposite the lateral gate of the translocon, which is a region required for the lateral translocation of the transmembrane domain of the polypeptide, and is considered dispensable for the function of the translocon complex (Rapoport, 2007). To investigate whether Sec61β is required for the translocation of ER-targeting proteins, we examined the secretion into the culture medium of endogenous secretory protein transthyretin (TTR) using HepG2 cells lacking Sec61β. Sec61α deficiency inhibited the secretion of TTR and induced the accumulation of intracellular TTR, whereas Sec61β deficiency did not (Fig. EV2A). These results suggest that Sec61β, unlike Sec61α, does not appear to play a major role in the translocation of TTR into the ER. We next examined the effect of Sec61β deficiency in ERpQC by using the null Hong Kong mutant of secretory protein α1-antitrypsin (α1AT) lacking N-glycosylation sites (NHK$^{QQQ}$) and TTR as ERpQC substrates. NHK$^{QQQ}$ and TTR were engineered to carry amino (N)-terminal Flag tags, allowing the identification and purification of signal peptide-uncleaved forms of NHK$^{QQQ}$ ($^S$NHK$^{QQQ}$) and TTR ($^S$TTR), as ERpQC substrates. These forms can be distinguished by their differences in molecular weight and by the retention of the Flag tag, which is removed upon signal peptide cleavage in the mature form, with minimal impact on translocation efficiency (Kadowaki et al, 2015). In HEK293 cells transfected with control siRNA, Tg and MG132 treatment induced the accumulation of $^S$NHK$^{QQQ}$ and $^S$TTR, which are ERpQC substrates (Figs. 1D and EV2B, black arrowheads). As we previously reported, the loss of Derlin-1, -2, and -3 reduced the accumulation of $^S$NHK$^{QQQ}$ (Fig. 1D). This is because substrates not captured by Derlins are inserted into the ER (Kadowaki et al, 2015). Surprisingly, despite binding to Derlins, Sec61β deficiency enhanced the accumulation of $^S$NHK$^{QQQ}$ and $^S$TTR (Figs. 1D and EV2B, black arrowheads). The accumulation of $^S$NHK$^{QQQ}$ and $^S$TTR in Sec61β-deficient HEK293 cells was abrogated by exogenous Flag-Sec61β (Figs. 1E and EV2C, black arrowheads). We used HepG2 cells to investigate the endogenous ERpQC substrate α1AT. Sec61α deficiency clearly increased the amount of α1AT with altered molecular weight in addition to nonglycosylated or glycosylated α1AT (Fig. EV2D, lane 9, red arrowheads). Deglycosylation via Endo-H treatment did not affect the amount of α1AT with altered molecular weight, suggesting that these proteins may be untranslocated α1AT ($^*$α1AT), considered to be the signal peptide-uncleaved form (Fig. EV2E, red arrowheads). Sec61β deficiency increased the expression of $^*$α1AT during ER stress (Fig. EV2D, lanes 7,8). In contrast, Sec61β deficiency did not affect the ER translocation of endogenous BiP and GRP94 (Fig. EV2D, arrows) and exogenous BiP and PDI (Fig. EV2F), which are not ERpQC substrates (Kadowaki et al, 2015). Since Sec61β interacts with the ERpQC substrate-rerouting factor Derlins, we next examined whether Sec61β-mediated inhibition of substrate accumulation occurs in concert with Derlins in Tg and MG132-treated cells. The overexpression of Derlin-1 enhanced the accumulation of $^S$NHK$^{QQQ}$ (Fig. 1F, lanes 3 and 4, black arrowhead), and this accumulation was further accelerated by Sec61β deficiency (Fig. 1F, lanes 5 and 6). However, Sec61β deficiency had little effect on the accumulation of $^S$NHK$^{QQQ}$ in *Derlin-1, -2,* and *-3* triple knockout cells as compared with that in wild-type cells (Fig. 1G, lanes 2, 4, 6, and 8, see percentages of $^S$NHK$^{QQQ}$ expression relative to the total expression). These findings suggest that Sec61β inhibits the accumulation of ERpQC substrates rerouted by Derlins.

## The translation of ERpQC substrate mRNAs is suppressed by Sec61β

There are two possible mechanisms by which Sec61β inhibits the accumulation of ERpQC substrates: promoted degradation or inhibited synthesis of ERpQC substrates. To investigate the possibility of Sec61β-mediated degradation of ERpQC substrates, we performed a pulse-chase experiment using Flag-NHK$^{QQQ}$ immunoprecipitated (IPed) by an N-terminal Flag tag and found that Sec61β deficiency did not affect the degradation of $^S$NHK$^{QQQ}$ (Fig. EV3A,B). Next, to investigate the possibility that Sec61β regulates the synthesis of ERpQC substrates, newly synthesized ERpQC substrates were analyzed in a pulse-label experiment. The amounts of newly synthesized $^S$NHK$^{QQQ}$ and $^S$TTR between 3- and 10-min labeling with [$^{35}$S]-methionine/cysteine were increased by Sec61β deficiency, whereas the amounts of signal peptide-cleaved NHK$^{QQQ}$ ($^C$NHK$^{QQQ}$) and TTR ($^C$TTR) were unchanged (Fig. 2A,B). Unlike ERpQC substrates, the synthesis of BiP and CL1, a cytoplasmic protein containing a degron sequence, was unaffected by Sec61β deficiency (Fig. 2C,D). To confirm these results, newly synthesized Flag-NHK$^{QQQ}$ was labeled with puromycin for 10 min followed by immunoprecipitation (IP) with an anti-Flag antibody and immunoblotting (IB) with an anti-puromycin antibody. Sec61β deficiency had little effect on the total amount of newly synthesized proteins labeled with puromycin (Fig. EV3C, Input), but increased the amount of newly synthesized Flag-NHK$^{QQQ}$ (Fig. EV3C, IP:

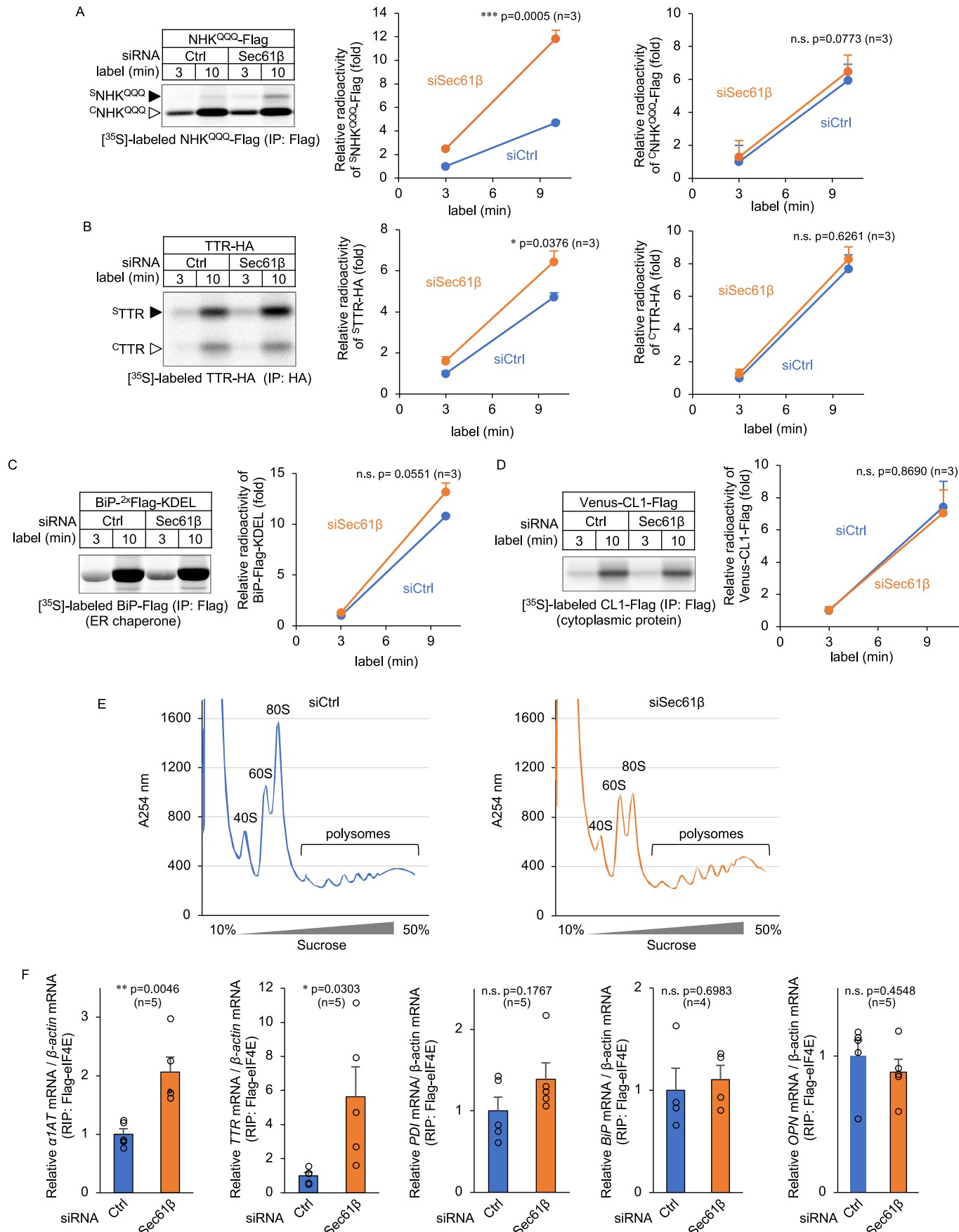

**Figure 2.  Translational repression of ERpQC substrates by Sec61β.**

(A–D) IP of [$^{35}$S]-methionine/cysteine-labeled NHK$^{QQQ}$-Flag (A), TTR-HA (B), BiP-$^{2\times}$Flag-KDEL (C), or Venus-CL1-Flag (D) for indicated times in HEK293 cells transfected with indicated siRNAs and plasmids and treated with or without 50 nM Tg and 200 nM MG132 for 16 h using antibodies against tag. Graphs show the increase in radioactivity of indicated [$^{35}$S]-labeled proteins from 3 to 10 min, relative to the intensity observed for 3-min labeling in siCtrl-transfected cells ($n = 3$). Black arrowhead, $^{S}$NHK$^{QQQ}$ or signal peptide-uncleaved TTR ($^{S}$TTR); white arrowhead, $^{C}$NHK$^{QQQ}$ or signal peptide-cleaved TTR ($^{C}$TTR). (E) Polysome profiles of lysates from siCtrl- or siSec61β-transfected HEK293 cells treated with 50 nM Tg and 200 nM MG132, assessed by 10–50% sucrose gradient ultracentrifugation. (F) RNA immunoprecipitation (RIP) of the association of $^{3\times}$Flag-eIF4E with ERpQC substrate mRNA, *α1AT* and *TTR* or non-substrate mRNA, *PDI*, *BiP*, and *OPN*. $^{3\times}$Flag-eIF4E was overexpressed and IPed using an anti-Flag antibody in HepG2 cells transfected with siCtrl or siSec61β and treated with 50 nM Tg and 200 nM MG132 for 16 h. The levels of the indicated mRNAs (normalized to *β-actin* mRNA) in $^{3\times}$Flag-eIF4E-bound mRNA were analyzed by RT-qPCR ($n = 5, 5, 5, 4$ and 5 from left to right, respectively). Data are means ± SEM. *$P < 0.05$, **$P < 0.01$, ***$P < 0.001$; n.s., not significant. Two-tailed unpaired *t* test for siCtrl vs siSec61β at 10 min labeling (A–D) and for siCtrl vs siSec61β (F). Source data are available online for this figure.

Flag-IB: Puromycin). Thus, Sec61β may specifically suppress the translation of ERpQC substrate mRNAs rerouted by Derlins. PERK-mediated eIF2α phosphorylation pathway is classically known as a mechanism for ER stress-induced translational attenuation (Harding et al, 1999). To determine whether Sec61β-mediated translational repression is related to the canonical eIF2α pathway, we first examined eIF2α phosphorylation levels in Sec61β-deficient HepG2 cells under ER stress conditions. Tg and MG132 treatment increased eIF2α phosphorylation to a similar extent in both control and Sec61β knockdown cells, with no significant difference between them (Fig. EV3D), suggesting that ER stress-induced eIF2α phosphorylation occurs independently of Sec61β. Next, we used ISRIB, an inhibitor of the integrated stress response (ISR), to examine the involvement of eIF2α phosphorylation-mediated translational attenuation in the Sec61β-mediated translational repression of ERpQC substrates. ISRIB treatment increased the expression of $^{S}$NHK$^{QQQ}$ under ER stress conditions (Fig. EV3E, lanes 1 and 3). Importantly, Sec61β deficiency further enhanced the accumulation of $^{S}$NHK$^{QQQ}$, but not that of $^{C}$NHK$^{QQQ}$ or BiP, in ISRIB-treated cells (Fig. EV3E, lanes 3 and 4). These results suggest that the PERK–eIF2α pathway induces broad translational attenuation in the vicinity of the ER membrane under stress conditions, whereas Sec61β-mediated translational regulation specifically suppresses the accumulation of ERpQC substrates through a mechanism distinct from the ISR. In other words, the ERpQC substrate may be subject to dual translational regulation mediated by the Sec61β–ARIH1 axis in addition to the PERK–eIF2α pathway.

Sucrose density gradient sedimentation analysis revealed that Sec61β deficiency increased the ribosome density of polysomal fractions in Tg- and MG132-treated cells (Fig. 2E), suggesting that Sec61β attenuates polysome formation under ER stress conditions. To investigate the involvement of Sec61β in translational repression in more depth, we performed RNA IP (RIP) of the mRNA cap-binding protein eIF4E with ERpQC substrate or non-substrate mRNA in HepG2 cells transfected with control or Sec61β siRNA. Endogenous mRNA for *α1AT*, *TTR*, *PDI*, *BiP*, and secretory protein osteopontin (*OPN*), which is not an ERpQC substrate (Kang et al, 2006), was analyzed via RIP using Flag-eIF4E IPed with an anti-Flag antibody. In Sec61β-deficient HepG2 cells, *α1AT* and *TTR* mRNAs, but not the non-ERpQC substrates *PDI*, *BiP*, and *OPN* mRNAs, were significantly enriched in IPed eIF4E (Fig. 2F). Collectively, our findings suggest that Sec61β inhibits polysome formation by specifically suppressing eIF4E-dependent translation of ERpQC substrate mRNAs captured via Derlins during ER stress.

## Interaction between Sec61β and ARIH1 recruits 4EHP to the ERpQC complex

To elucidate the molecular mechanism of Sec61β-mediated translational repression, Sec61β-binding molecules were analyzed via MS-based quantitative proteomics. We identified 22 molecules whose interaction with Sec61β was significantly increased by ≥2-fold by Tg and MG132 treatment (Dataset EV2) and focused on the RBR ubiquitin E3 ligase ARIH1, which demonstrated the most enhanced binding (Fig. 3A,B). In $^{3\times}$Flag-tagged *Sec61β* knock-in HEK293 cells, an endogenous interaction between Sec61β and ARIH1 was confirmed, and the amount of the Sec61β–ARIH1 complex increased under ER stress conditions (Fig. 3C). In addition to Sec61β, Derlin-1 also exhibited enhanced interaction with ARIH1 upon ER stress (Fig. 3D). To confirm the ER stress-dependent physiological interactions among Derlin-1, Sec61β, and ARIH1, we performed in vivo experiment in which intraperitoneal administration of tunicamycin (Tun; an N-glycosylation inhibitor) induces acute ER stress in the mouse liver, a highly secretory organ with active protein synthesis in the ER (Eura et al, 2020; Yamamoto et al, 2010; Zinszner et al, 1998). Importantly, endogenous interactions of Derlin-1 with Sec61β and ARIH1 were observed in livers of Tun-treated mice (Fig. 3E). These results suggest that ARIH1 is physiologically recruited to the ERpQC complex under ER stress conditions. ARIH1 has been reported to bind to and modify 4EHP, which interacts with the 5′ cap structure of mRNA instead of eIF4E to inhibit translation, thereby contributing to translation inhibition upon DNA damage (Tan et al, 2003; von Stechow et al, 2015). Exogenous 4EHP bound to endogenous ARIH1, Sec61β, and Derlin-1, and all these interactions were increased by ER stress (Fig. 3F). Moreover, the interaction of 4EHP with Sec61β and Derlin-1 was inhibited by ARIH1 deficiency (Fig. 3G). Thus, ER stress may promote the complex formation of Derlins with Sec61β and ARIH1, thereby recruiting 4EHP via ARIH1. Next, we examined the subcellular localization of the Sec61β–ARIH1–4EHP complex. The interactions between Sec61β and ARIH1 or 4EHP were visualized using a proximity ligation assay (PLA). In ER stress-treated HepG2 cells, PLA signals representing the interactions of Flag-Sec61β with HA-ARIH1 or HA-4EHP were observed along the reticular structures defined by coexpression of the ER marker GFP-KDEL (Fig. EV4A–D). Taken together, Sec61β–ARIH1–4EHP complex formation may occur on the ER membrane via Derlins.

We hypothesized that ARIH1 may regulate the binding of 4EHP to the mRNA 5′ cap structure instead of eIF4E, inhibiting the translation of ERpQC substrate mRNAs. If this hypothesis is

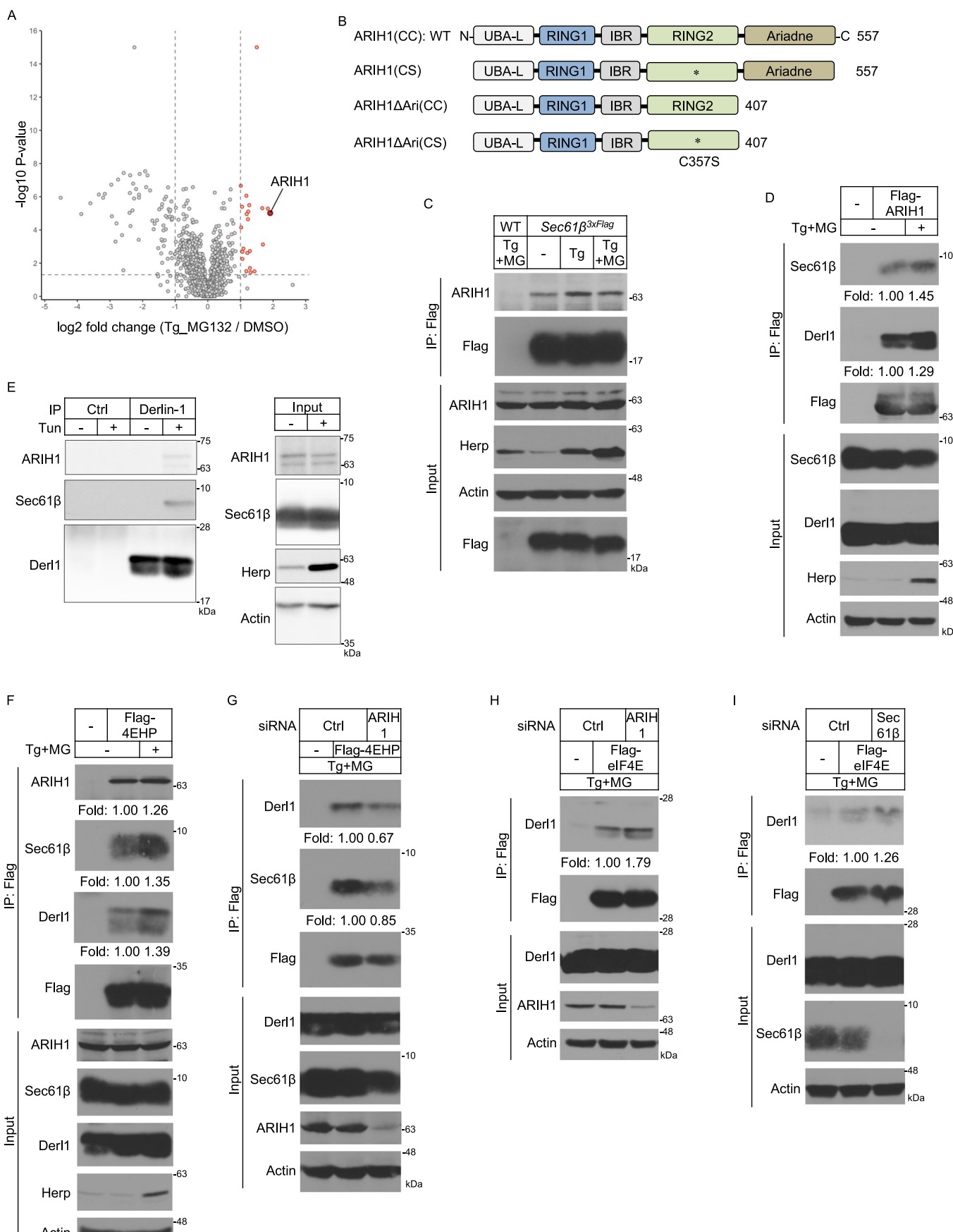

◄ **Figure 3. ER stress-induced enhancement of Sec61β interactions with ARIH1 and 4EHP.**

(A) Volcano plots of the quantitative proteomic analysis of ER stress-induced Sec61β interactome. Plots indicate the fold change in the abundance of identified proteins in Tg and MG132-treated samples compared with that in DMSO-treated samples (log2 fold change [Tg_MG132/DMSO], x axis) against its significance (−log10 P value, y axis) (n = 4). Orange, significant fold change (Tg_MG132/DMSO) ≥ 2, P value ≤ 0.05; red, ARIH1. See also Dataset EV2. (B) Domain structures of human ARIH1 and truncated forms with or without mutation. UBA-L, ubiquitin-associated domain-like; RING1, RING domain 1; IBR, in-between RING domain; RING2, RING domain 2; C357S (CS), catalytically inactive serine mutant of Cys357, the active site of Ub ligase activity; ΔAri, mutant lacking inhibitory Ariadne domain. (C) Endogenous interaction of Sec61β with ARIH1 during ER stress. IP with an anti-Flag antibody and IB with indicated antibodies in WT or 3×Flag-tagged *Sec61β* knock-in HEK293 cells and treated with or without 50 nM Tg and/or 200 nM MG132 for 16 h. (D) Interaction of endogenous Sec61β and Derlin-1 with exogenous ARIH1 during ER stress. IP with an anti-Flag antibody and IB with indicated antibodies in HEK293 cells transfected with Flag-ARIH1 and treated with or without 50 nM Tg and 200 nM MG132 for 16 h. The amounts of co-IPed proteins with Flag-ARIH1 were normalized by the amount of input for each protein and shown as fold increases relative to the control lane. (E) Endogenous interaction of Sec61β and ARIH1 with Derlin-1 during ER stress. Wild-type C57BL/6 J mice (12 to 14-week-old), matched for sex, were given a single 2 μg/gram body weight intraperitoneal injection of a 0.1 mg/ml suspension of tunicamycin (Tun) in PBS or vehicle (PBS) alone. After 20 h, mice were deeply anesthetized and transcardially perfused with PBS. Whole cell lysates were prepared by homogenizing livers for 60 s × five times in lysis buffer. Cell lysates were IPed with anti-Derlin-1 antibody or control IgG using Protein G Sepharose. All samples were immunoblotted with indicated antibodies. (F, G) Interaction of endogenous ARIH1 (F), Sec61β (F, G), or Derlin-1 (F, G) with exogenous 4EHP in HEK293 cells (F) and ARIH1-deficient HEK293 cells (G). IP with an anti-Flag antibody and IB with indicated antibodies in HEK293 cells transfected with indicated siRNAs (G) and Flag-4EHP (F, G) and treated with or without 50 nM Tg and 200 nM MG132 for 16 h. The amounts of co-IPed proteins with Flag-4EHP were normalized by the amount of input for each protein and shown as fold changes relative to the control lane. (H, I) Interaction of endogenous Derlin-1 with exogenous eIF4E in ARIH1- (H) or Sec61β- (I) deficient cells. IP with an anti-Flag antibody and IB with indicated antibodies in HEK293 cells transfected with indicated siRNAs and Flag-eIF4E and treated with 50 nM Tg and 200 nM MG132 for 16 h. The amount of co-IPed Derlin-1 with Flag-eIF4E was normalized by the amount of input Derlin-1 and shown as fold increase relative to the control lane. Source data are available online for this figure.

correct, since Derlins recruit the RNC of ERpQC substrates via their interaction with eIF4F in an ER stress-dependent manner (Fig. EV1B–E), ARIH1 may prevent subsequent recruitment of eIF4F to the 5′ cap structure of ERpQC substrate mRNAs. To test this idea, we examined the role of ARIH1 in the ER stress-dependent binding of Derlin-1 to eIF4E and found that ARIH1 deficiency enhanced the interaction between endogenous Derlin-1 and exogenous eIF4E (Fig. 3H). Similar results were observed in Sec61β-deficient cells (Fig. 3I). These results suggest that the ER stress-induced Sec61β–ARIH1 complex may keep eIF4F away from ERpQC substrate mRNAs through the recruitment of 4EHP to the mRNA 5′ cap structure of Derlins-bound RNCs.

## ARIH1 and 4EHP suppress the translation of ERpQC substrate mRNAs

We next investigated the expression levels of $^S$NHK$^{QQQ}$ to test whether ARIH1 and 4EHP regulate ERpQC substrate translation. Similar to Sec61β deficiency, ARIH1 or 4EHP deficiency significantly increased the accumulation of $^S$NHK$^{QQQ}$ in Tg- and MG132-treated cells (Fig. 4A,B), suggesting that ARIH1 and 4EHP suppress the expression of the ERpQC substrate. The effect of 4EHP deficiency was rescued by exogenous 4EHP (Fig. 4C). 4EHP has been reported to bind to cap analogs with lower affinity than eIF4E in in vitro experiments (Zuberek et al, 2007). However, 4EHP can compete with eIF4E through its interaction with the eIF4E-binding protein 4E-T (eIF4E transporter), which enhances the affinity of 4EHP for the 5' cap structure and enables suppression of translation initiation (Chapat et al, 2017). Moreover, 4EHP is recruited to the 5' cap structure via the GIGYF1/2 adaptor complex and mediates selective translational repression of collided ribosomes (Morita et al, 2012; Weber et al, 2020). It is also known that the translation of *Dual specificity phosphatase 6* (*DUSP6*) mRNA, encoding an ERK1/2 phosphatase, is repressed by 4EHP (Jafarnejad et al, 2018). Therefore, it is important to verify whether 4EHP actually binds to the 5' cap structure of ERpQC substrate mRNAs. To address this, we analyzed the association of mRNAs for *DUSP6* and the ERpQC substrates *α1AT* and *TTR* with Flag-4EHP

expressed in HepG2 cells under ER stress conditions using RIP. The amount of *DUSP6* mRNA bound to Flag-4EHP did not change upon ER stress (Fig. EV4E). Conversely, the amounts of *α1AT* (Fig. 4D) and *TTR* (Fig. EV4F) mRNAs bound to Flag-4EHP increased in an ER stress-dependent manner. Furthermore, Sec61β deficiency reduced the binding of *α1AT* (Fig. 4E) and *TTR* (Fig. EV4G) mRNAs to Flag-4EHP. These results suggest that 4EHP may be recruited to the 5' cap structure of specific ERpQC substrate mRNAs via Sec61β under ER stress conditions, thereby contributing to their translational repression. However, the precise mechanism by which the Sec61β–ARIH1 complex facilitates the association of 4EHP with the 5' cap structure and potentially displaces eIF4F remains unclear.

ARIH1 has been reported to ubiquitinate 4EHP or conjugate it with the ubiquitin-like protein ISG15 (ISGylation) in its E3 ligase activity-dependent manner (Okumura et al, 2007; Tan et al, 2003; von Stechow et al, 2015). However, whether these modifications of 4EHP positively or negatively regulate translation remains controversial. A structural study has revealed that the C-terminal Ariadne domain masks the catalytic cysteine (C357) of ARIH1, resulting in autoinhibition of its E3 ligase activity under basal conditions (Duda et al, 2013). Therefore, ARIH1 activation requires a conformational change that removes the autoinhibitory masking mediated by the Ariadne domain. To assess the requirement of the E3 ligase activity of ARIH1 for ERpQC, we developed a constitutively active form of ARIH1 lacking the inhibitory Ariadne domain [ARIH1ΔAri(CC)] and its inactive form with a serine mutation of C357 [ARIH1ΔAri(CS)] (Fig. 3B). The E3 ligase activity of full-length (FL) ARIH1 was required for its interaction with 4EHP and Sec61β (Fig. EV4H, lanes 2 and 3), whereas ARIH1ΔAri interacted with them independently of its E3 ligase activity (Fig. EV4H, lanes 4 and 5). These results suggest that the E3 ligase activity may be necessary for the conformational change in ARIH1, enabling 4EHP binding. Supporting this notion, the phospho-mimetic mutation on the Ariadne domain [ARIH1(S427D)], which relieves the autoinhibitory activity by opening the catalytic site (Reiter et al, 2022), also bound to 4EHP and Sec61β independently of its E3 ligase activity (Fig. EV4I, lanes 4 and 5). Thus, the E3 ligase activity may facilitate a conformational transition of ARIH1 from a closed (inactive) to an

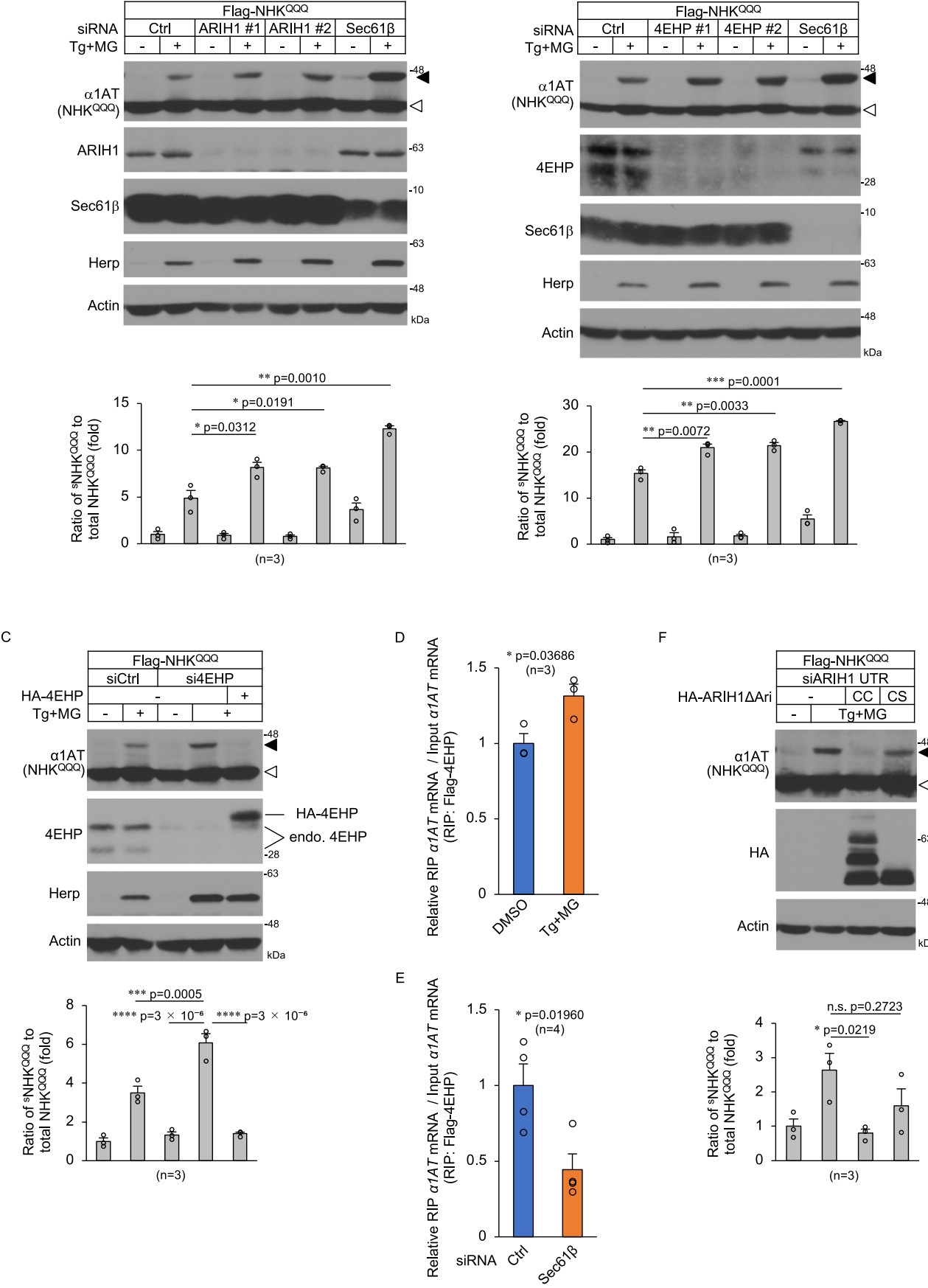

**Figure 4.   Inhibition of ERpQC substrate accumulation by ARIH1 and 4EHP.**

(A, B) IB of ERpQC substrate in HEK293 cells transfected with siRNAs against ARIH1 (A), 4EHP (B), or Sec61β (A, B) and Flag-NHK$^{QQQ}$ and treated with or without 50 nM Tg and 200 nM MG132 for 16 h; samples were immunoblotted with indicated antibodies. Bar graph: Ratio of the expression level of $^{S}$NHK$^{QQQ}$ to the total amount of NHK$^{QQQ}$ was calculated and shown ($n = 3$). (C) IB of ERpQC substrates in HEK293 cells transfected with siRNA against 4EHP and cDNAs for Flag-NHK$^{QQQ}$ and HA-4EHP and treated with or without 50 nM Tg and 200 nM MG132 for 16 h; samples were immunoblotted with indicated antibodies. Bar graph: Ratio of expression level of $^{S}$NHK$^{QQQ}$ to the total amount of NHK$^{QQQ}$ was calculated and shown ($n = 3$). (D, E) RIP of Flag-4EHP with ERpQC substrate mRNA, $a1AT$. HepG2 cells were transfected with Flag-4EHP alone (D) or together with siCtrl or siSec61β (E) and treated with or without 50 nM Tg and 200 nM MG132 for 16 h (D) or with 50 nM Tg and 200 nM MG132 for 16 h (E). Flag-4EHP was IPed using an anti-Flag antibody, and the levels of the indicated mRNAs (normalized to input) in Flag-4EHP-bound mRNA were analyzed by RT-qPCR (D; $n = 3$, E; $n = 4$). (F) IB of ERpQC substrates in HEK293 cells transfected with siRNA against ARIH1 UTR and cDNAs for Flag-NHK$^{QQQ}$ and HA-ARIH1ΔAri(CC) or (CS) and treated with or without 50 nM Tg and 200 nM MG132 for 16 h; samples were immunoblotted with indicated antibodies. Bar graph: Ratio of expression level of $^{S}$NHK$^{QQQ}$ to the total amount of NHK$^{QQQ}$ was calculated and shown ($n = 3$). Data are means ± SEM. $^*P < 0.05$, $^{**}P < 0.01$, $^{***}P < 0.001$, $^{****}P < 0.0001$; n.s., not significant. Two-tailed unpaired $t$ test for siCtrl vs siARIH1 (A), siCtrl vs si4EHP (B) or siCtrl vs siSec61β (A, B); one-way ANOVA with Tukey's multiple comparisons test (C). Two-tailed unpaired $t$ test for DMSO vs Tg+MG (D) or siCtrl vs siSec61β (E); one-way ANOVA with Tukey's multiple comparisons test (F). For (A–C, F), black arrowhead, $^{S}$NHK$^{QQQ}$; white arrowhead, $^{C}$NHK$^{QQQ}$. Source data are available online for this figure.

open (active) state, thereby enabling the recruitment of Sec61β and 4EHP under ER stress conditions. Collectively, these results suggest that opening of ARIH1 catalytic site appears to be crucial for Sec61β–ARIH1–4EHP complex formation under ER stress conditions. We next examined the requirement of the E3 ligase activity of ARIH1 for the translational repression of ERpQC substrates. Increased $^{S}$NHK$^{QQQ}$ accumulation in ARIH1-deficient cells was cancelled by the expression of ARIH1ΔAri(CC), but not by the expression of ARIH1ΔAri(CS), despite its ability to bind 4EHP (Fig. 4F). Although ISGylation of 4EHP has been shown to be involved in translational repression (Okumura et al, 2007), non-ISGylatable 4EHP mutant (4KR; K121R/K130R/K134R/K222R) clearly inhibited the accumulation of $^{S}$NHK$^{QQQ}$ accumulation in Sec61β knockdown cells similarly to wild-type 4EHP (Fig. EV4J). Therefore, ISGylation of 4EHP might be unlikely to play a major role in the translational repression of ERpQC substrates. Nevertheless, we do not possess clear evidence that completely rules out the possibility that ISGylation or other posttranslational modifications of 4EHP might contribute to this process. Taken together, the E3 ligase activity of ARIH1 appears to be essential not only for driving the conformational change required for interactions with 4EHP and Sec61β, but also for repressing the translation of ERpQC substrate mRNAs. However, the detailed mechanisms underlying the release of ARIH1 autoinhibition and the posttranslational modification of its E3 ligase substrates have not yet been elucidated, and the precise role of ARIH1 E3 ligase activity in the translational repression of ERpQC substrates remains to be clarified.

## Sec61β inhibits the accumulation of ERpQC substrates through the interaction with ARIH1 and 4EHP

Next, we examined the role of ARIH1 bound to Sec61β in the translational repression of the ERpQC substrates. The accumulation of $^{S}$NHK$^{QQQ}$ in Sec61β-deficient cells was suppressed by exogenous ARIH1ΔAri(CC) in an E3 ligase activity-dependent manner (Fig. 5A). Similar effect was also observed with exogenous 4EHP (Fig. 5B). Moreover, the coexpression of ARIH1ΔAri(CC) and 4EHP completely suppressed $^{S}$NHK$^{QQQ}$ accumulation in Sec61β-deficient cells (Fig. 5C). We next investigated whether 4EHP cooperates with Sec61β and ARIH1 to regulate the expression of ERpQC substrates. Intriguingly, 4EHP deficiency significantly cancelled the effect of ARIH1ΔAri(CC) in Sec61β-

deficient cells (Fig. 5D). These results suggest that 4EHP may inhibit the translation of ERpQC substrate mRNAs by forming a complex with Sec61β and ARIH1. Sec61β is a tail-anchored ER membrane protein with an N-terminal intrinsically disordered region (IDR; amino acids 1–54) exposed to the cytoplasm (Fig. 5E). This IDR of Sec61β was required for the interaction with ARIH1 and 4EHP (Fig. 5F). Consistent with the result of interaction, Sec61βΔIDR, unlike Sec61βWT, failed to reduce $^{S}$NHK$^{QQQ}$ accumulation in Sec61β-deficient cells (Fig. 5G), suggesting that Sec61β regulates the translation of ERpQC substrate mRNAs through the recruitment of ARIH1 and 4EHP.

## Sec61β maintains cytoplasmic proteostasis during ER stress

In considering the physiological role of Sec61β, we hypothesized that Sec61β may inhibit the overproduction of ERpQC substrates containing hydrophobic signal peptides in the cytoplasm, thereby reducing the burden on cytoplasmic proteostasis. We examined the proteasome activity in Sec61β-deficient HepG2 cells under ER stress conditions. Sec61β deficiency using three different siRNAs all significantly reduced proteasome activity upon treatment with ER stressors, Tg and Tun (Figs. 6A and EV5A). We next examined the degradation of cytoplasmic protein via pulse-chase experiments using the CL1 degron protein. Sec61β deficiency had no effect on the degradation of the ERpQC substrate (Fig. EV3A,B), whereas that of Venus-CL1-Flag was significantly delayed (Figs. 6B and EV5B). As delayed degradation of cytoplasmic proteins due to Sec61β deficiency was predicted to induce the aggregation of unfolded proteins in the cytoplasm, we performed aggresome staining using ProteoStat, which monitors protein aggregate formation, in HepG2 cells. The ProteoStat fluorescence signal was increased in Sec61β-deficient HepG2 cells compared with that in control cells (Fig. 6C). The protein aggregates in Sec61β-deficient cells did not colocalize with the ER membrane protein calnexin (Fig. EV5C), suggesting that Sec61β inhibits aggresome formation in the cytoplasm. Fluorescence-activated cell sorting (FACS) analysis revealed that Sec61β deficiency significantly increased the number of cells with protein aggregates during ER stress (Fig. 6D). Since the IDR of Sec61β is essential for its interaction with ARIH1 and 4EHP and for the suppression of ERpQC substrate accumulation (Fig. 5F,G), we performed FACS

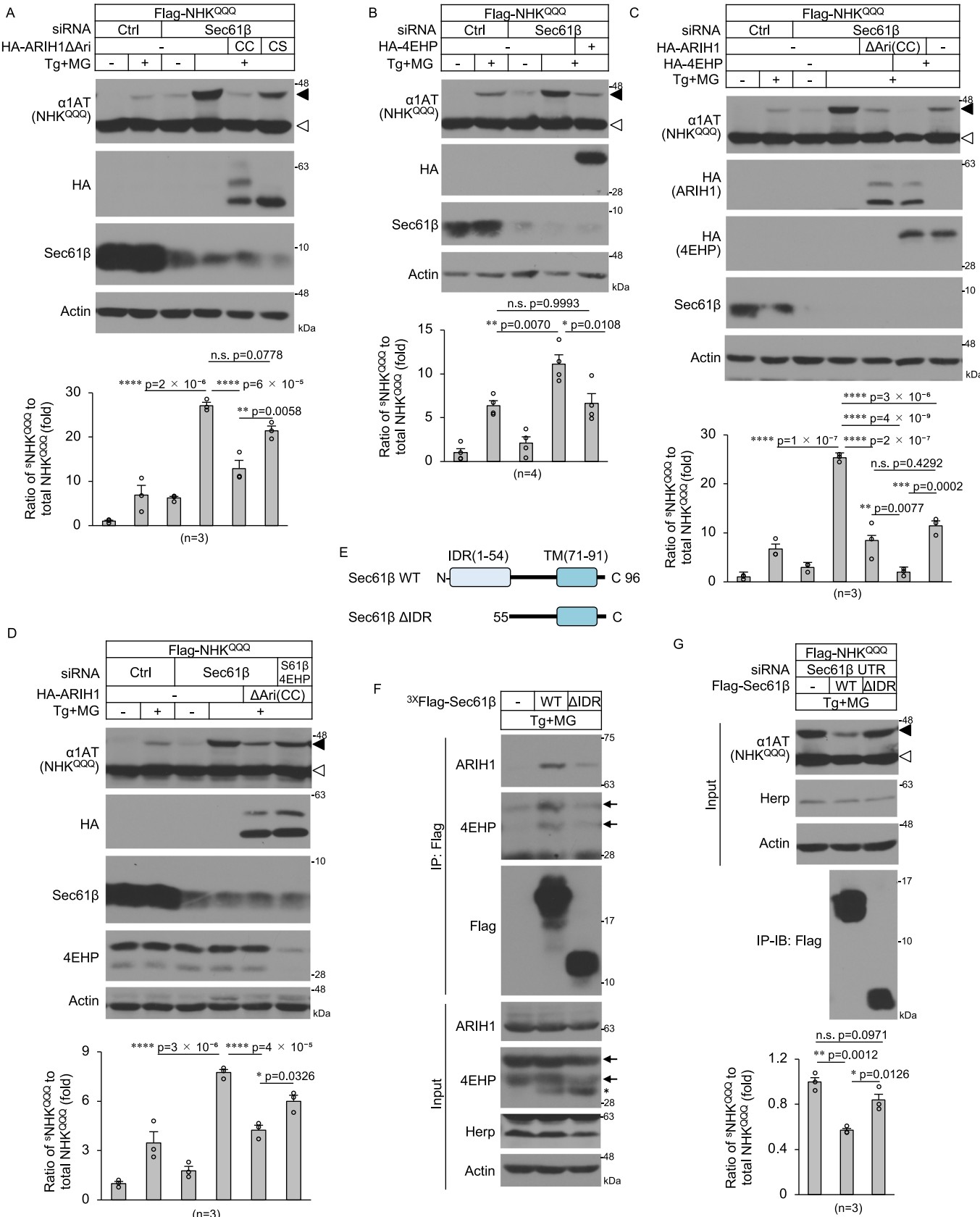

**Figure 5.   Inhibition of ERpQC substrate accumulation through the interaction of Sec61β with ARIH1 and 4EHP.**

(A–D) IB of ERpQC substrate in HEK293 cells transfected with indicated siRNAs and plasmids and then treated with or without 50 nM Tg and 200 nM MG132 for 16 h; samples were immunoblotted with indicated antibodies. Bar graph: Ratio of the expression level of $^S$NHK$^{QQQ}$ to the total amount of NHK$^{QQQ}$ ($n = 3$ for A, C, D and $n = 4$ for B). (E) Domain structure of human Sec61β and its truncated form. IDR intrinsically disordered region, TM transmembrane domain. (F) Interaction of endogenous ARIH1 and 4EHP with exogenous Sec61β. IP with an anti-Flag antibody and IB with indicated antibodies in HEK293 cells transfected with indicated plasmids and treated with 50 nM Tg and 200 nM MG132 for 16 h. Two endogenous 4EHP bands were detected in the input, and both interacted with Sec61βWT (arrows). These bands may reflect post-translational modified 4EHP, including previously reported ubiquitination or ISGylation, in addition to the possibility of splicing products. Arrows, 4EHP; asterisk, a non-specific band unrelated to 4EHP that appears due to overexpression of $^{3\times}$Flag-Sec61β. (G) IB of ERpQC substrate in HEK293 cells transfected with Sec61β UTR siRNA and indicated plasmids treated with or without 50 nM Tg and 200 nM MG132 for 16 h. For exogenous Flag-Sec61β, lysates were IPed with an anti-Flag antibody and immunoblotted with an anti-Flag antibody; input samples were immunoblotted with indicated antibodies. Bar graph: ratio of the expression level of $^S$NHK$^{QQQ}$ to the total amount of NHK$^{QQQ}$ was calculated and shown ($n = 3$). Data are means ± SEM. *$P < 0.05$, **$P < 0.01$, ***$P < 0.001$, ****$P < 0.0001$; n.s., not significant. One-way ANOVA with Tukey's multiple comparisons test. For (A–D, G), black arrowhead, $^S$NHK$^{QQQ}$; white arrowhead, $^C$NHK$^{QQQ}$. Source data are available online for this figure.

analysis to examine whether the Sec61β IDR is also required to prevent aggresome formation in Sec61β-deficient HepG2 cells. Sec61βΔIDR-expressing cells showed significantly higher Proteo-Stat fluorescence than Sec61βWT-expressing cells (Fig. 6E). Taken together, Sec61β may suppress the proteasomal degradation load by inhibiting the accumulation of ERpQC substrates and contribute to the maintenance of cytoplasmic proteostasis through its interaction with ARIH1 and 4EHP.

## Sec61β contributes to motor function of zebrafish via the E3 ligase activity of ARIH1

Finally, we investigated the effect of Sec61β deficiency in zebrafish by knockdown with an antisense morpholino oligonucleotide (MO) designed against the translation initiation codon of *Sec61β* (Sec61β-atg MO). Sec61β-atg MO clearly reduced Sec61β expression, whereas Sec61β-atg-5mis MO, a control MO with five mismatches, did not (Fig. EV5D). At 3 days post-fertilization (dpf), a very small number of Sec61β-deficient larvae exhibited morphological abnormalities (Fig. EV5E, arrowhead). Their body length tended to be shorter than that of water- or Sec61β-atg-5mis MO-injected larvae (Fig. EV5F). Detailed behavioral analysis revealed that Sec61β-deficient larvae were unable to move their tails smoothly during swimming. To further assess this swimming performance, larvae at 3 dpf were video-recorded while their heads and yolks were immobilized in agarose, allowing free movement of the trunk and tail. The amplitude of tail movement in Sec61β-deficient larvae was significantly smaller than that in control water- or Sec61β-atg-5mis MO-injected larvae (Fig. 6F,G). The degree of morphological and swimming abnormalities was scored from 0 to 3 at 4 dpf (Fig. EV5G). Sec61β-deficient larvae swam with their heads swinging from side to side and exhibited significantly higher abnormal scores than control larvae (Fig. 6H). Notably, this defect could be partially but significantly restored by the expression of human ARIH1ΔAri(CC), but not by that of the catalytically inactive ARIH1ΔAri(CS) (Figs. 6I and EV5H), suggesting that the E3 ligase activity of ARIH1 bound to Sec61β is required for morphogenesis and motor function in zebrafish. The result obtained in Sec61β-deficient larvae expressing ARIH1ΔAri is consistent with the finding that the E3 ligase activity of ARIH1ΔAri is required for suppressing the accumulation ERpQC substrates in Sec61β-deficient cells (Fig. 5A). Collectively, the maintenance of intracellular proteostasis via the Sec61β-ARIH1 axis may play an important physiological role in vivo.

## Discussion

Although many studies have elucidated the molecular mechanisms that maintain proteostasis in the cytoplasm and individual organelles, how proteostasis is coordinated across organelle membranes remain poorly understood. Excessive accumulation of misfolded proteins in the cytoplasm disturbs the ER quality, which has been implicated in pathologies such as neurodegenerative diseases (Nishitoh et al, 2008; Nishitoh et al, 2002). This is probably because reduced proteasome activity in the cytoplasm impairs ERAD function. However, whether disruption of the ER quality control system reciprocally affects cytoplasmic proteostasis remains unknown. If so, it is crucial to understand how ER homeostasis is linked to cytoplasmic homeostasis. ERpQC is a system that maintains ER homeostasis by inhibiting the translocation of newly synthesized proteins into the ER and degrading them immediately in the cytoplasm (Kang et al, 2006; Orsi et al, 2006; Oyadomari et al, 2006). We previously demonstrated the mechanisms of ERpQC, whereby ER stress induces the formation of the ERpQC complex comprising the Sec61 translocon, Derlins, and HRD1 on the ER membrane. ER-targeting proteins are rerouted from the ER translocation pathway to the cytoplasmic degradation pathway by the interaction of Derlins with the SRP–SR complex, ubiquitinated by HRD1, and degraded by the proteasome via p97 and Bag6 (Kadowaki et al, 2015; Kadowaki et al, 2018) (Fig. 7). The substrates degraded by ERpQC exhibit selectivity, and many ER chaperones evade ERpQC-mediated degradation and continue to be transported into the ER to ameliorate misfolded proteins. Conversely, ER-targeting proteins that are not required for ER quality control, such as secretory proteins that escape the PERK-mediated translational attenuation or the IRE1-mediated mRNA decay, are degraded by ERpQC, thereby reducing the folding load in the ER. This study demonstrates that ERpQC maintains cytoplasmic proteostasis as well as ER proteostasis through the translational regulation of specific substrates. Under ER stress conditions, the translocon component Sec61β, which interacts with Derlins, recruits the E3 ligase ARIH1 and its binding partner, the cap-binding protein 4EHP, via its cytoplasmic IDR. The RNC of ERpQC substrates is captured by the Derlins–SR–SRP complex, thereby ARIH1 recruited to the ER membrane enables the association of 4EHP with the 5′ cap structure of the ERpQC substrate mRNAs (Fig. 7). Consequently, 4EHP bound to ARIH1 may prevent the recruitment of eIF4F to the mRNA, leading to the translational repression of ERpQC substrates. However, the precise

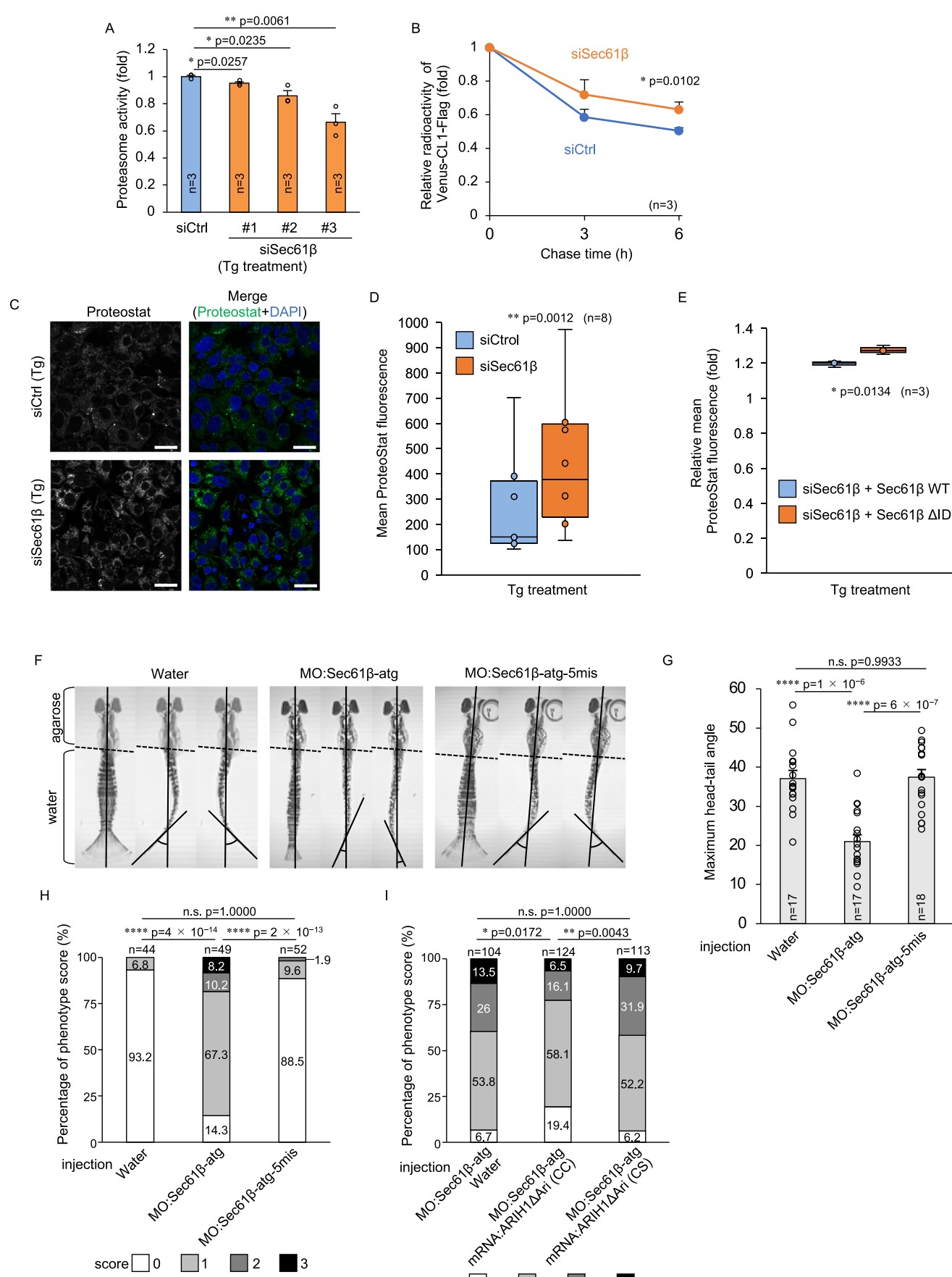

**Figure 6. Sec61β-mediated maintenance of cytoplasmic proteostasis and motor function of zebrafish.**

(A) Proteasome chymotrypsin-like peptidase activity of cell extracts from HepG2 cells transfected with siCtrl or siSec61β (#1, #2, or #3) and treated with 200 nM Tg for 16 h was measured using Suc-LLVY-AMC as a substrate. Fluorescence intensity was normalized to cell viability in each condition. Proteasome activity is shown as fold decrease relative to that of siCtrl-transfected cells (n = 3). (B) Degradation of the cytoplasmic protein CL1 degron after a 15-min pulse of [$^{35}$S]-methionine/cysteine metabolic labeling, followed by the indicated chase periods. Lysates of HEK293 cells transfected with indicated siRNAs and Venus-CL1-Flag and stimulated with 50 nM Tg for 16 h were IPed with an anti-Flag antibody and resolved via SDS-PAGE. The relative radioactivity of Venus-CL1-Flag at different chase times was calculated and shown as fold decreases relative to the intensity observed at 0 h chase (n = 3). (C) Representative fluorescence images of HepG2 cells transfected with siCtrl or siSec61β and stimulated with 50 nM Tg for 10 h, followed by staining with ProteoStat (green, protein aggregation) and DAPI (blue, nuclei). Scale bars, 25 μm. (D, E) Quantification of protein aggregation in Sec61β-deficient HepG2 cells using ProteoStat and flow cytometry. Mean ProteoStat fluorescence in HepG2 cells transfected with siCtrl or siSec61β and stimulated with 50 nM Tg for 4 h (D) and relative mean ProteoStat fluorescence in HepG2 cells transfected with siSec61β and indicated cDNAs and stimulated with 50 nM Tg for 4 h (E) were analyzed by flow cytometry and software. Boxes represent the 25th–75th percentiles with the median indicated; whiskers represent the minimum and maximum values (n = 8 for D or n = 3 for E). (F) Superimposed images of tail movements at 3 dpf and a scheme of the quantitative analysis of head-tail angle. Compared with larvae injected with water or control Sec61β-atg-5mis MO, Sec61β-deficient zebrafish exhibited impaired swimming behavior. (G) Histogram showing the maximum head-tail angles at 3 dpf. Sec61β-deficient zebrafish (n = 17) showed reduced maximum head-tail angles compared to control animals (n = 17 for water injection or n = 18 for Sec61β-atg-5mis MO injection). (H) Histogram showing the distribution of phenotype scores of zebrafish at 4 dpf. The degree of morphological and swimming abnormalities was scored from 0 to 3 as shown in Fig. EV5G. Sec61β-deficient zebrafish (n = 49) showed greater abnormality scores than control animals (n = 44 for water injection or n = 52 for Sec61β-atg-5mis MO injection). (I) Histogram showing the rescue of abnormal phenotype in Sec61β-deficient zebrafish by ARIH1 at 4 dpf. Exogenous expression of human ARIH1ΔAri(CC) (n = 124), but not ARIH1ΔAri(CS) (n = 113), reduced the abnormal phenotype score in Sec61β-deficient zebrafish (n = 104). Data are means ± SEM. *P < 0.05, **P < 0.01, ***P < 0.001, ****P < 0.0001; n.s., not significant. Two-tailed unpaired t test for siCtrl vs siSec61β (A, B); two-tailed paired t test for siCtrl vs siSec61β (D); two-tailed unpaired t test for Sec61β WT vs Sec61β ΔIDR (E); one-way ANOVA with Tukey's multiple comparisons test (G); Fisher's exact test followed by Bonferroni's post hoc test (H, I). Source data are available online for this figure.

mechanism by which ARIH1 facilitates the binding of 4EHP to the 5' cap structure and displaces eIF4F remains unclear, and the definitive evidence excluding the involvement of other molecules has not been obtained. Therefore, our conclusion is limited to the possibility that 4EHP acts as a mediator in the Sec61β-ARIH1-dependent translational repression of ERpQC substrates. Such a Sec61β-ARIH1-mediated quality control system not only efficiently degrades proteins that disrupt ER proteostasis but also contributes to maintaining cytoplasmic proteostasis by preventing the over-production of unnecessary proteins (e.g., polypeptides with signal sequence) in the cytosol. Indeed, disturbance of this mechanism by Sec61β deficiency triggers cytoplasmic protein aggregation in hepatocytes and impairs motor function in zebrafish (Fig. 7).

In cotranslational translocation, the SRP is considered to preferentially bind to the RNC of secretory proteins before the signal sequence emerges. Consequently, ER-targeting proteins are recognized and translocated by prerecruited SRP immediately after the signal sequence emerges from the ribosomal tunnel (Chartron et al, 2016). ER-targeting proteins are regulated by strict quality control to ensure proper ER targeting of nascent polypeptides during the pioneer round of translation (Kramer et al, 2019). Therefore, the binding of the RNC of ERpQC substrates to SRP and its subsequent recruitment to Derlins during the pioneer round of translation may be the first step in determining substrate specificity and degrading them effectively (Fig. 7). During the pioneer round of translation on mRNAs encoding ER-targeting proteins, the nuclear cap-binding complex (CBC) represses the aberrant translation of nascent polypeptides on the RNC–SRP complex by directly binding to the SRP until the nascent polypeptides are correctly targeted to the ER (Park et al, 2021). This surveillance by the CBC–RNC–SRP complex is also important for maintaining cytoplasmic proteostasis. It seems inconsistent that ER stress enhanced the interaction of Derlins not only with eIF4F but also with 4EHP (Figs. EV1 and 3). In the case of ERpQC substrates, translation may be repressed by the Sec61β–ARIH1–4EHP complex recruited to Derlins without replacement of CBC by eIF4E in the CBC–RNC–SRP complex formed during the pioneer round of translation. The regulation of translation initiation reduces the

number of ribosomes on the mRNA, thereby giving more time for proper folding and improving the maturation of misfolded membrane proteins on the ER membrane (Lakshminarayan et al, 2020). Therefore, inhibition of polysome formation via the Derlins–Sec61β–ARIH1 complex may allow minimal degradation of ERpQC substrates by the proteasome.

As a quality control mechanism similar to ERpQC, mislocalized ER-targeting proteins are degraded by Bag6 and the E3 ubiquitin ligase RNF126 to prevent aberrant aggresome formation in the cytoplasm (Hessa et al, 2011; Rodrigo-Brenni et al, 2014). In yeast, the ribosome collision sensor Hel2 (ZNF598 in mammals) maintains proteostasis in the secretory pathway by reducing the mistargeting of ER proteins into mitochondria due to SRP dysfunction (Matsuo and Inada, 2021). Although ERpQC differs from the quality control of mislocalized proteins that are not captured by Derlins, both systems are important for maintaining cytoplasmic proteostasis. Various factors, such as the mRNA secondary structure and the lack of stop codons, cause ribosome stalling on the mRNA, and stalled ribosomes and remaining aberrant nascent polypeptides are efficiently removed by ribosome quality control (RQC) (Inada, 2020; Yip and Shao, 2021). For stalled ribosomes near the translocon, the E3 ligase Listerin ubiquitinates nascent polypeptides on the 60S subunit for proteasomal degradation (Brown et al, 2015; Shao et al, 2013; von der Malsburg et al, 2015). Since ERpQC substrates are ubiquitinated by HRD1, but rarely by Listerine (Kadowaki et al, 2018), the mechanism of ERpQC will differ from that of RQC related to ribosome stalling near the ER membrane. Age-dependent increased ribosome stalling causes RQC overload and aggregation of nascent polypeptides, which ultimately affects lifespan (Stein et al, 2022). Therefore, although the mechanisms of ERpQC and RQC differ, the strict regulation of cytoplasmic proteostasis via Sec61β in ERpQC may also affect biological functions, including lifespan as well as motor function.

The N-terminal cytoplasmic domain of Sec61β is poorly conserved among species, contains an unstructured IDR, and can interact with ribosomes (Levy et al, 2001). Sbh1, the yeast homologue of Sec61β, promotes the ER translocation of proteins with suboptimal targeting sequences, and its activity is regulated by the phosphorylation of N-terminal IDR (Barbieri et al, 2023).

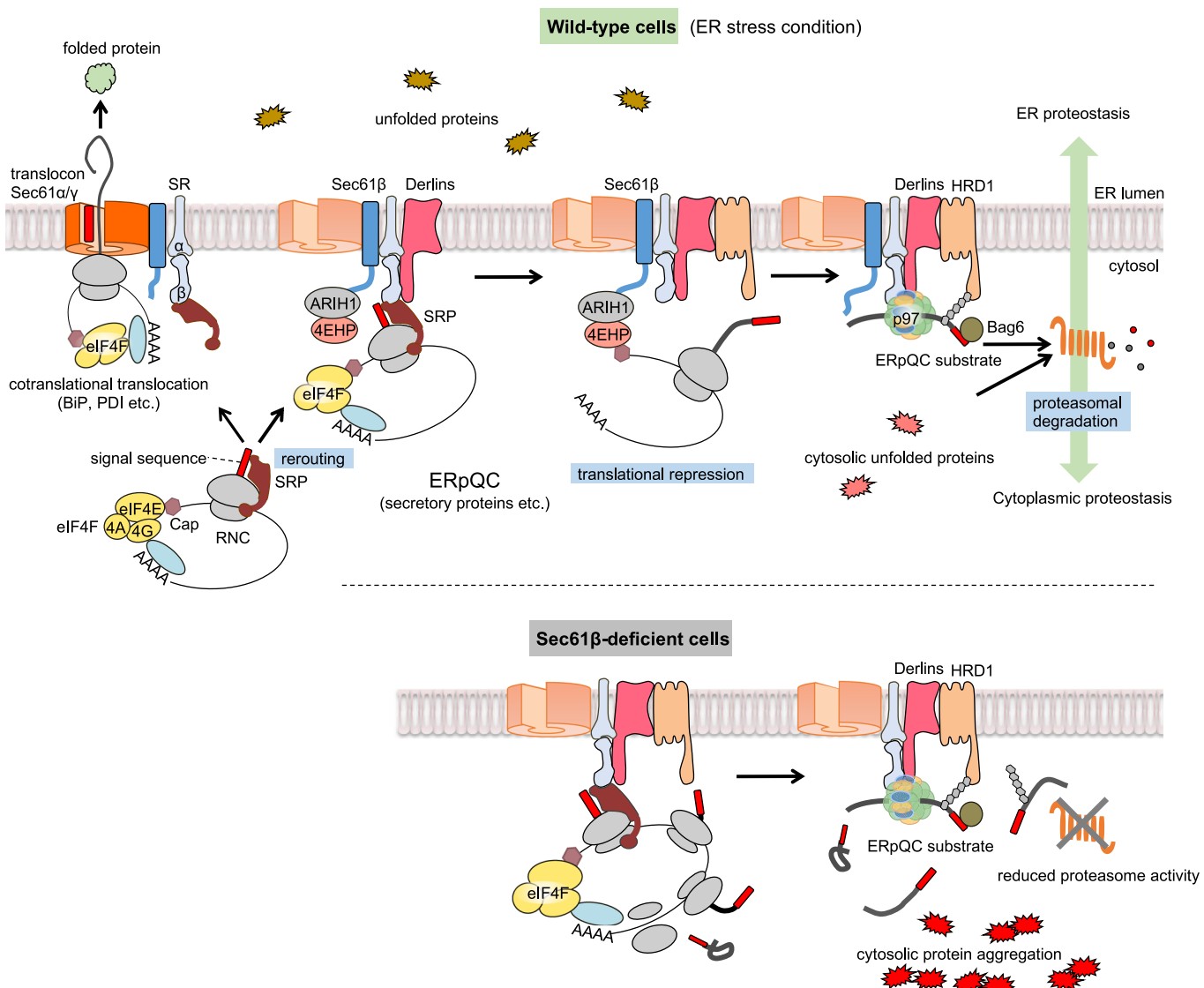

**Figure 7. Working model for the translational repression of ERpQC substrates via the Derlins–Sec61β–ARIH1 complex.**

Under ER stress conditions, chaperones such as BiP and PDI continue to be synthesized by cotranslational translocation into the ER lumen (upper left). The signal sequence of nascent polypeptides emerging from the ribosome is recognized by SRP, and RNC–SRP complex docks onto SRα/β. When the SRP dissociates from the RNC, the nascent polypeptide is delivered into the ER, where it undergoes proper folding. By contrast, in ERpQC (upper right), some secretory proteins are degraded by the proteasome without translocation into the ER. ERpQC is triggered by recruitment of Derlins to the Sec61 translocon and SR in response to ER stress. Derlins interact with SRP54 on the RNC through their C-terminus, thereby rerouting the nascent polypeptide from the ER translocation pathway to the cytoplasmic degradation pathway (rerouting). Sec61β bound to Derlins recruits the E3 ligase ARIH1 via its IDR upon ER stress. ARIH1, guided in the vicinity of the ER membrane, enables 4EHP to associate with the 5′ cap structure of the ERpQC substrate mRNAs captured by Derlins, thereby causing translational repression of ERpQC substrates (translational repression). Rerouted ERpQC substrates that have already initiated translation are ubiquitinated by HRD1 and effectively transported to the proteasome by p97 and Bag6 (proteasomal degradation). Derlins–Sec61β–ARIH1 complex-mediated translational repression contributes to reduced proteasomal degradation load of cytosolic unfolded proteins, leading to inhibition of aggresome formation and maintenance of cytoplasmic proteostasis. Conversely, Sec61β deficiency allows eIF4F recruitment to the mRNA 5′ cap structure of the Derlins-bound RNC and polysome formation, resulting in the overproduction of ERpQC substrates and disruption of cytosolic proteostasis (lower).

However, Sec61β is not considered to be essential for the translocation of ER-targeting proteins (Kalies et al, 1998), and the precise role of its IDR remains largely unknown. In this study, we showed that the Sec61β IDR was required for the interaction with ARIH1, for translational repression of ERpQC substrates, and for inhibition of aggresome formation (Figs. 5 and 6). However, we failed to demonstrate the interaction of the Sec61β IDR with ARIH1, possibly due to difficulties in expressing the IDR alone

without aggregation in cells. Structural elucidation of the mechanism by which the IDR of Sec61β bound to Derlins recruits ARIH1 to the vicinity of the ER membrane may help provide ways to improve ER and cytoplasmic proteostasis during ER stress, which is often observed in conformational diseases.

ARIH1 induces the ubiquitination or ISGylation of 4EHP in the presence of the E2 conjugating enzymes UBCH7 or UBCH8, respectively. These modifications promote the cap-binding capacity

of 4EHP and inhibit the recruitment of eIF4F, thereby repressing translation in immune and genotoxic stress responses (Okumura et al, 2007; von Stechow et al, 2015). Conversely, ARIH1-mediated polyubiquitination of 4EHP causes its degradation (Tan et al, 2003). In our study, non-ISGylatable 4EHP mutant (K121R/K130R/K134R/K222R) effectively suppressed the accumulation of ERpQC substrates in Sec61β-deficient cells, similar to wild-type 4EHP (Fig. EV4J). Although 4EHP ISGylation may be unlikely to play a major role in ERpQC, we cannot entirely exclude the possibility that ISGylation or other modifications of 4EHP contribute to translational repression. As a mechanism for ARIH1 activation, ARIH1 is known to be a component of Cullin-RING ubiquitin ligase (CRL) complexes and is activated by neddylated Cullin–RBX1 complex to catalyze the monoubiquitination of CRL substrates (Kelsall et al, 2013; Scott et al, 2016). Although ARIH1 has been the subject of a large amount of research in recent years, its role, particularly regarding the function of E3 ligase, remains controversial (Villa et al, 2017; Wu et al, 2021; Xiong et al, 2022). While the role of 4EHP modification in ERpQC complex formation is still unclear, the E3 ligase activity of ARIH1 is required for its interaction with 4EHP and for the suppression of ERpQC substrate accumulation (Figs. 4F, 5A, and EV4H). One possibility is that the E3 ligase activity is required for the conformational change in ARIH1 itself, which opens its inhibitory Ariadne domain that masks the RING domain with the catalytic cysteine. As a result, ARIH1 associates with Sec61β and recruits 4EHP to the vicinity of the ER membrane. Another possibility is that ARIH1 modifies 4EHP or unknown molecules related to the translation of ERpQC substrate mRNAs. Both possibilities may be implicated. Further analysis will be needed to elucidate the mechanism of selective translational repression of ERpQC substrates under ER stress conditions.

In conclusion, in the ERpQC system, which maintains ER quality by altering the location of translation from the ER to the cytoplasm via Derlins upon ER stress, the selective regulation of translational efficiency of ERpQC substrates reduces the overload on the degradation machinery in the cytoplasm, resulting in the maintenance of proteostasis in the cytoplasm as well as in the ER. We show that Sec61β, an ER stress-dependent Derlins-binding partner, enables the translational repression of ERpQC substrates by recruiting ARIH1 and 4EHP to the vicinity of the translocon. Future detailed studies will clarify the precise mechanism of Derlins–Sec61β–ARIH1 complex-mediated translational repression of ERpQC substrates, and pharmacological approaches targeting the ERpQC mechanism will be expected to overcome various diseases related to impaired ER and cytoplasmic proteostasis due to ER stress.

# Methods

### Reagents and tools table

| Reagent/resource | Reference or source | Identifier or catalog number |
| --- | --- | --- |
| **Experimental models** | | |
| HEK293A cells (*H. sapiens*) | ATCC | R70507 |
| HepG2 cells (*H. sapiens*) | ATCC | HB-8065 |

| Reagent/resource | Reference or source | Identifier or catalog number |
| --- | --- | --- |
| *Derlin-1, -2,* and *-3* triple knockout HEK293 cells | Kadowaki et al, 2015 | N/A |
| 3 × Flag-tagged *Sec61β* knock-in HEK293 cells | This study | |
| Zebrafish: AB strain (WT) | Natural breeding | N/A |
| C57BL/6J (*M. musculus*) | Jackson Lab | B6.129P2Gpr37 tm1Dgen/J |
| **Recombinant DNA** | | |
| pcDNA3/GW Derlin-1-Flag | Kadowaki et al, 2015 | N/A |
| pcDNA3/GW Derlin-2-Flag | Kadowaki et al, 2015 | N/A |
| pcDNA3/GW Derlin-3-Flag | Kadowaki et al, 2015 | N/A |
| pcDNA3/GW Flag-eIF4A1 (CDS of NM_001416.4) | This study | N/A |
| pcDNA3/GW Flag-eIF4E (CDS of NM_001968.5) | This study | N/A |
| pcDNA3/GW 3 × Flag-eIF4E (CDS of NM_001968.5) | This study | |
| pcDNA3/GW Flag-Sec61β (CDS of NM_006808.3) | This study | N/A |
| pcDNA3/GW 3 × Flag-Sec61β (CDS of NM_006808.3) | This study | N/A |
| pcDNA3/GW Flag-Sec61βΔIDR (55-96) (CDS of NM_006808.3) | This study | N/A |
| pcDNA3/GW 3 × Flag-Sec61βΔIDR (55-96) (CDS of NM_006808.3) | This study | N/A |
| pcDNA3/GW Derlin-1-HA | Kadowaki et al, 2015 | N/A |
| pcDNA3/GW Derlin-2-HA | Kadowaki et al, 2015 | N/A |
| pcDNA3/GW Derlin-3-HA | Kadowaki et al, 2015 | N/A |
| pcDNA3/GW Flag-ARIH1 (CDS of NM_005744.5) | This study | N/A |
| pcDNA3/GW Flag-ARIH1 C357S (CDS of NM_005744.5) | This study | N/A |
| pcDNA3/GW Flag-ARIH1 S427D (CDS of NM_005744.5) | This study | N/A |
| pcDNA3/GW Flag-ARIH1 C357S/S427D (CDS of NM_005744.5) | This study | N/A |
| pcDNA3/GW Flag-ARIH1ΔAri (1-407) (CDS of NM_005744.5) | This study | N/A |
| pcDNA3/GW HA-ARIH1ΔAri (1-407) (CDS of NM_005744.5) | This study | N/A |
| pcDNA3/GW Flag-ARIH1Δ Ari (1-407) C357S (CDS of NM_005744.5) | This study | N/A |
| pcDNA3/GW HA-ARIH1Δ Ari (1-407) C357S (CDS of NM_005744.5) | This study | N/A |

| Reagent/resource | Reference or source | Identifier or catalog number |
| --- | --- | --- |
| pcDNA3/GW Flag-4EHP (CDS of NM_004846.4) | This study | N/A |
| pcDNA3/GW HA-4EHP (CDS of NM_004846.4) | This study | N/A |
| pcDNA3/GW HA-4EHP 4KR (K121/130/134/222 R) (CDS of NM_004846.4) | This study | N/A |
| pcDNA3/GW Flag-6His-NHK^QQQ-HA | Kadowaki et al, 2015 | N/A |
| pcDNA3/GW Flag-6His-NHK^QQQ-Myc | This study | N/A |
| pcDNA3/GW NHK^QQQ-Flag | Kadowaki et al, 2015 | N/A |
| pcDNA3/GW Flag-TTR-HA | Kadowaki et al, 2015 | N/A |
| pcDNA3/GW TTR-Flag | Kadowaki et al, 2015 | N/A |
| pcDNA3/GW BiP-2 × Flag-KDEL | Kadowaki et al, 2015 | N/A |
| pcDNA3/GW PDI-2 × Flag-KDEL | Kadowaki et al, 2015 | N/A |
| pcDNA3/GW Venus-CL1-Flag | Nishitoh et al, 2008 | N/A |
| pcDNA3.1 GFP-KDEL | Bannai et al, 2004 | N/A |
| pCS2+ HA-ARIH1Δ Ari (1-407) C357S (CDS of NM_005744.5) | This study | N/A |
| pMK344-3 x Flag | This study | N/A |
| pX330 | Addgene | Cat. #42230 |
| **Antibodies** | | |
| Mouse monoclonal anti-DYKDDDDK tag (clone FLA-1, IB: 1:5000, IF: 1:10,000) | MBL | Cat. #185-3 L; RRID: AB_11123930 |
| Rat monoclonal anti-HA tag (clone 3F10, IB: 1:1000) | Roche Diagnostics | Cat. #11867431001; RRID: AB_390919 |
| Rabbit monoclonal anti-HA tag (IF: 1:1600) | Cell Signaling Technology | Cat. #3724; RRID: AB_1549585 |
| Rabbit polyclonal anti-Herp (IB: 1:4000) | Fujisawa et al, 2012 | N/A |
| Rabbit polyclonal anti-α1AT (IB: 1:2000) | Dako | Cat. #A0012 |
| Rabbit polyclonal anti-TTR (IB: 1:2000) | Dako | Cat. #A0002 |
| Rabbit polyclonal anti-Derlin-1 (IB: 1:4000, IP: 1 µg/ml) | Nishitoh et al, 2008 | N/A |
| Rabbit polyclonal anti-Derlin-2 (IB: 1:2000) | MBL | Cat. #PM019; RRID: AB_593007 |
| Mouse monoclonal anti-Derlin-2 (clone D-10, IB: 1:100, IP: 1 µg/ml) | Santa Cruz Biotechnology | Cat. #SC-398573 |

| Reagent/resource | Reference or source | Identifier or catalog number |
| --- | --- | --- |
| Rabbit polyclonal anti-Sec61α (IB: 1:5000) | Affinity BioReagents | Cat. #PA3-014 |
| Rabbit polyclonal anti-Sec61β (IB: 1:500) | Proteintech | Cat. #15087-1-AP; RRID: AB_2186411 |
| Rabbit polyclonal anti-eIF4A1 (IB: 1:2000) | Abcam | Cat. #ab31217; RRID: AB_732122 |
| Rabbit monoclonal anti-eIF4E (IB: 1:1000) | Cell Signaling Technology | Cat. #2067; RRID: AB_2097675 |
| Rabbit polyclonal anti-phospho-eIF2α (Ser52) (IB: 1:1000) | Invitrogen | Cat. #44-728 G; RRID: AB_1500038 |
| Rabbit polyclonal anti-eIF2α (IB: 1:400) | Santa Cruz Biotechnology | Cat. #SC-11386; RRID: AB_640075 |
| Rabbit polyclonal anti-ARIH1 (IB: 1:2000) | Proteintech | Cat. #14949-1-AP RRID: AB_3085447 |
| Mouse monoclonal anti-ARIH1 (clone C-7, IB: 1:100) | Santa Cruz Biotechnology | Cat. #SC-514551 |
| Rabbit polyclonal anti-4EHP (IB: 1:1000) | Gene Tex | Cat. #GTX103977; RRID: AB_2036842 |
| Rabbit polyclonal anti-KDEL (IB: 1:2000) | Enzo Life Sciences | Cat. #ADI-SPA-827; RRID: AB_10618036 |
| Mouse monoclonal anti-Puromycin (clone 12D10, IB: 1:2500) | Millipore | Cat. #MABE343; RRID: AB_2566826 |
| Rabbit monoclonal anti-ATF4 (IB: 1:1000) | Cell Signaling Technology | Cat. #11815; RRID: AB_26160025 |
| Mouse monoclonal anti-Actin (clone AC-40, IB: 1:3000) | Sigma-Aldrich | Cat. #A4700; RRID: AB_476730 |
| Rabbit polyclonal anti-calnexin (IF: 1:1000) | Abcam | Cat. #ab22595; RRID: AB_2069006 |
| APC anti-DYKDDDDK Tag antibody (IF: 1:450) | Biolegend | Cat. #637308 |
| Rabbit polyclonal anti-Sec61β (Sec61β_9AA3; Immunogen: PGPTPSGTN in human and mouse Sec61β, IB: 1:200, IP: 1 µg/ml) | This study (COSMO BIO) | N/A |
| Rabbit polyclonal anti-Sec61β (Sec61β_21AA3; Immunogen: SAGTGGMWRFYTEDSPGLKV in human, mouse, and zebrafish Sec61β, IB: 1:500) | This study (COSMO BIO) | N/A |
| Rabbit IgG control polyclonal antibody | Proteintech | Cat. #30000-0-AP; RRID: AB_2819035 |

| Reagent/resource | Reference or source | Identifier or catalog number |
|---|---|---|
| Mouse IgG2b isotype control monoclonal antibody | Proteintech | Cat. #66360-3-Ig; RRID: AB_2881740 |
| Anti-rabbit IgG, HRP-linked (IB: 1:4000) | Cell Signaling Technology | Cat. #7074; RRID: AB_2099233 |
| Anti-mouse IgG, HRP-linked (IB: 1:5000) | Cytiva | Cat. #NA931; RRID: AB_772210 |
| Anti-rat IgG, HRP-linked (IB: 1:5000) | Cytiva | Cat. #NA935; RRID: AB_772207 |
| Cy5-conjugated donkey anti-mouse IgG (H + L) (IF: 1:200) | Jackson | Cat. #715-175-151; RRID: AB_2340820 |
| CF647-conjugated donkey anti-rabbit IgG (H + L) (IF: 1:2000) | Biotium | Cat. #20047; RRID: AB_10559808 |
| Anti-Flag M2 affinity gel | Sigma-Aldrich | Cat. #A2220; RRID: AB_10063035 |
| HA-tagged protein purification gel | MBL | Cat. #3321 |
| Anti-Flag M2 magnetic beads | Sigma-Aldrich | Cat. #M8823; RRID: AB_2637089 |
| **Oligonucleotides and other sequence-based reagents** | | |
| RT-qPCR primers | | |
| qPCR primer: α1AT forward: 5′-GCCAT CTTCTTCCTG CCTGA-3′ | This study | N/A |
| qPCR primer: α1AT reverse: 5′-GCTGGCAGACCTTCT GTCTT-3′ | This study | N/A |
| qPCR primer: TTR forward: 5′-ATCCAAGTGTCC TCTGATGGTC-3′ | This study | N/A |
| qPCR primer: TTR reverse: 5′-TTTTCCCAGAGGC AAATGGC-3′ | This study | N/A |
| qPCR primer: PDI forward: 5′-TCAAGGTGCTTGTTG GGAAG-3′ | This study | N/A |
| qPCR primer: PDI reverse: 5′-AATGGGAGCCAAC TGTTTGC-3′ | This study | N/A |
| qPCR primer: BiP forward: 5′-CTTGCCGTTCAA GGTGGTTG-3′ | This study | N/A |
| qPCR primer: BiP reverse: 5′-TCCCAAATAAGCCTCAGCGG-3′ | This study | N/A |
| qPCR primer: OPN forward: 5′-GCCGA GGTGATAGTG TGGTT-3′ | This study | N/A |
| qPCR primer: OPN reverse: 5′-AAC GGGGATGGCCTT GTATG-3′ | This study | N/A |
| qPCR primer: DUSP6 forward: 5′-CGGATCACTGGA GCCAAAAC-3′ | This study | N/A |
| qPCR primer: DUSP6 reverse: 5′-ATGCCAGCCAAGCAA TGTAC-3′ | This study | N/A |
| qPCR primer: β-actin forward: 5′-CACCATTGGCAATGAG CGGTTC-3′ | This study | N/A |
| qPCR primer: β-actin reverse: 5′-AGGT CTTTGCGGAT GTCCACGT-3′ | This study | N/A |
| siRNAs | | |
| Stealth RNAi siRNA Negative Control Med GC Duplex | Invitrogen | Cat. #12935300 |
| Stealth siRNA Derlin-1 MSS228692: 5′-GGGUUAU CCUUGGAUUCAAC UAUAU-3′ | Invitrogen | Cat. #1320001 |
| Stealth siRNA Derlin-2 HSS121486: 5′-GGCCUUUACAA UAAUGCUCGUCUAU-3′ | Invitrogen | Cat. #1299001 |
| Stealth siRNA Derlin-3 HSS150566: 5′-CCCUGGGAUUCAGCUU CUUCUUCAA-3′ | Invitrogen | Cat. #1299001 |
| Stealth siRNA Sec61α HSS121072: 5′-UGGCAAUCAAA UUUCUGGAAGUCAU-3′ | Invitrogen | Cat. #1299001 |
| Stealth siRNA Sec61β #1 HSS145899: 5′-UC UUCUGUUCAUCGC UUCUGUAUUU-3′ | Invitrogen | Cat. #1299001 |
| Stealth siRNA Sec61β #2 HSS173999: 5′-GGG AUGUGGCGAUUCU ACACAGAAG-3′ | Invitrogen | Cat. #1299001 |
| Stealth siRNA Sec61β #3 HSS174000: 5′-ACC UGGGCUCAAA GUUGGCCCUGUU-3′ | Invitrogen | Cat. #1299001 |
| Stealth siRNA Sec61β UTR-targeting #1: 5′-CAGCGCCUUGCCACCCU CAUCUCCA-3′ | Invitrogen | NM_006808_stealth_50 |
| Stealth siRNA Sec61β UTR-targeting #2: 5′-CGUCAGCGCCUUGCC ACCCUCAUCU-3′ | Invitrogen | NM_006808_stealth_47 |
| Stealth siRNA ARIH1 #1 HSS119341: 5′-CC UGGUACAACUGU AACCGCUAUAA-3′ | Invitrogen | Cat. #1299001 |

| Reagent/resource | Reference or source | Identifier or catalog number |
|---|---|---|
| Stealth siRNA ARIH1 #2 HSS177850: 5'-GAU GACAGUGAAACCU CCAAUUGGA-3' | Invitrogen | Cat. #1299001 |
| Stealth siRNA ARIH1 UTR-targeting: 5'-CCAAGCCUUU GUGUGCCCAUGUUAU-3' | Invitrogen | NM_005744. 3_stealth_3032 |
| Stealth siRNA 4EHP #1 HSS114097: 5'-GCAGU AGCAAGAGAAA GGCUGUUGU-3' | Invitrogen | Cat. #1299001 |
| Stealth siRNA 4EHP #2 HSS190361: 5'-UGUCC GCUUUCAGGAAGACA UUAUU-3' | Invitrogen | Cat. #1299001 |
| Morpholino oligonucleotides (MOs) | | |
| Antisense MO: Sec61β-atg: 5'-AAAGCTGTGTTTTAGAC GACTGGCA-3' | Gene Tools, LLC | N/A |
| Control MO: Sec61β-atg-5mis: 5'-AAACCTGTCTTATAGA CCACTCGCA-3' | Gene Tools, LLC | N/A |
| Chemicals, enzymes and other reagents | | |
| Thapsigargin (Tg) | Nacalai Tesque | Cat. #33637-31 |
| Tunicamycin (Tun) | Nacalai Tesque | Cat. #35638-74 |
| MG132 | Calbiochem | Cat. #474790 |
| ISRIB | Sigma-Aldrich | Cat. #SML0843 |
| Polyethyleneimine (PEI)-MAX | Polysciences | Cat. #24765 |
| Lipofectamine RNAiMAX | Invitrogen | Cat. #13778150 |
| Lipofectamine 3000 | Invitrogen | Cat. #L300015 |
| Leupeptin | Nacalai Tesque | Cat. #4041 |
| Protease inhibitor cocktail | Nacalai Tesque | Cat. #25955-11 |
| cOmplete EDTA-free protease inhibitor cocktail | Roche | Cat. #11873580001 |
| N-Ethylmaleimide (NEM) | Nacalai Tesque | Cat. #15512-24 |
| Cycloheximide | Sigma-Aldrich | Cat. #C7698 |
| RNasin Plus RNase inhibitor | Promega | Cat. #N261A |
| Trypsin | Sigma-Aldrich | Cat. #4352157 |
| Tris-(2-carboxyethyl) phosphine (TCEP) | Thermo Fisher Scientific | Cat. #77720 |
| Lysyl-endopeptidase | Wako Pure Chemical Industries | Cat. #129-02541 |
| Protein G Sepharose | GE Healthcare | Cat. #17-0618-02 |
| RNAiso Plus | Takara-Bio | Cat. #9109 |
| Endoglycosidase H (Endo-H) | New England Biolabs | Cat. #P0702 |
| Succinil-Leu-Leu-Val-Tyr-7 amino-4-methylcoumarin (Suc-LLVY-AMC) | Calbiochem | Cat. #539142 |

| Reagent/resource | Reference or source | Identifier or catalog number |
|---|---|---|
| Prolong Diamond antifade reagent with DAPI | Invitrogen | Cat. #P36962 |
| EXPRE$^{35}$S$^{35}$S Protein Labeling Mix [$^{35}$S] | Revvity Health Sciences | Cat. #NEG772 |
| mMESSAGE mMACHINE SP6 kit | Invitrogen | Cat. #AM1340 |
| BCA protein assay (Takara-Bio, T9300A) | Takara-Bio | Cat. #T9300A |
| RevaTra Ace qPCR RT Master Mix with gDNA Remover | TOYOBO | Cat. #FSQ-301 |
| KAPA SYBR FAST qPCR Master Mix | KAPA BIOSYSTEMS | Cat. #KK4605 |
| Cell Counting Kit-8 | Dojindo Laboratories | Cat. #343-07623 |
| ProteoStat Aggresome detection kit | Enzo Life Sciences | Cat. #ENZ-51035-K100 |
| Duolink In Situ Detection Reagents FarRed | Sigma-Aldrich | DUO92013-100RXN |
| Duolink In Situ PLA Probe Anti-Rabbit MINUS | Sigma-Aldrich | DUO92005-100RXN |
| Duolink In Situ PLA Probe Anti-Mouse PLUS | Sigma-Aldrich | DUO92001-100RXN |
| Duolink In Situ Mounting Medium with DAPI | Sigma-Aldrich | DUO82040-5ML |
| Software | | |
| ImageQuant TL image analysis software ver. 8.1 | Cytiva https://www.cytivalifesciences.co.jp/catalog/1167.html | |
| Empiria Studio software ver. 3.0 | LICOR https://bio.licor.com/bio/help/empiria_studio/software/empiria_studio/about/release-notes.html | |
| EZR software ver. 1.54 | Kanda, 2013; https://www.jichi.ac.jp/saitama-sct/SaitamaHP.files/statmedEN.html | |
| R Studio ver.4.3.2 | Rstudio https://posit.co/download/rstudio-desktop/ | |
| DAVID 6.8 | Huang da et al, 2009 (Huang da et al, 2009) https://davidbioinformatics.nih.gov/ | |
| ProteinPilot software | Sciex https://sciex.com/products/software/proteinpilot-software | |
| Proteome Discoverer 2.2 software | Thermo Fisher Scientific https://www.fishersci.com/shop/products/NC2891056/NC2891056 | |
| Adobe Photoshop Elements 2021 | Adobe https://www.adobe.com/jp/products/premiere-elements.html | |
| FlowJo software | Tree star https://www.flowjo.com/ | |

| Reagent/resource | Reference or source | Identifier or catalog number |
|---|---|---|
| CytExpert software for DxFLEX 2.0 | Beckman Coulter https://www.beckman.jp/flow-cytometry/research-flow-cytometers/cytoflex/software | |

## Cell lines and cell culture

Wild-type and *Derlin-1*, *Derlin-2*, and *Derlin-3* triple knockout HEK293 cells (Kadowaki et al, 2015) and ³×Flag-tagged *Sec61β* knock-in HEK293 cells were cultured in Dulbecco's modified Eagle's medium (DMEM, Nacalai Tesque; 08459-64) containing 10% fetal bovine serum (FBS) and penicillin-streptomycin solution (Nacalai Tesque; 09367-34). ³×Flag-tagged *Sec61β* knock-in HEK293 cells were generated in accordance with the relevant guidelines of University of Miyazaki. All experimental protocols were approved by institutional guidelines of University of Miyazaki (735-4). To establish N-terminal tagging ³×Flag *Sec61β* knock-in HEK293 cells, CRISPR-Cas9 editing technology with homologous repair template was used. To obtain the vector for N-terminal tagging ³×Flag tag with hygromycin-resistant gene, pMK344 (Hyg B-P2A-mAID) was modified to replace the mAID gene with a ³×Flag sequence to create pMK344-3x Flag. Hyg B-P2A-3x Flag fragment was inserted at ATG codon site into donor plasmid containing 1.0 kbp of the homology arms near the Sec61β ATG codon. For sgRNA targeting the region near the ATG codon of *Sec61β*, complementary oligonucleotides including sgRNA target sites (underlines) CACCGCCTCATCTCCAATATGGTA (forward) and AAACTACCATATTGGAGATGAGGC (reverse) was cloned into the pX330 vector. HEK293 cells were co-transfected with pX330 vectors carrying target sequences for *Sec61β* and donor plasmid by PEI-Max. After selection with Hygromycin B (500 μg/ml), clone was confirmed by genomic PCR and the expression of ³×Flag- Sec61β protein was analyzed by immunoblotting. HepG2 cells were cultured in Eagle's Minimum Essential Medium (MEM, Nacalai Tesque: 21443-15) containing 10% FBS and penicillin-streptomycin solution. All cells were maintained in a 5% $CO_2$ atmosphere at 37 °C.

## Mice strains and maintenance

C57BL/6J mice were maintained in a specific pathogen-free environment. All mouse experiments were approved by the Animal Research Committee of the University of Miyazaki following institutional guidelines (2024-501-3). The experiments were conducted according to institutional guidelines. All efforts were made to minimize animal suffering and reduce the number of animals used.

## Zebrafish strains and maintenance

Adult zebrafish (*Danio rerio*) of AB strain were raised and maintained according to standard guidelines (Westerfield, 2000). The embryos were raised in E3 medium at 28.5 °C. All experimental procedures were performed in accordance with relevant guidelines and regulations and approved by the University of Miyazaki (2022-527-4).

## ER stress treatment

HEK293 cells were treated with 50 nM Tg (Nacalai Tesque) and/or 200 nM MG132 (Calbiochem) for 16 h. In immunoblotting experiments, HepG2 cells were treated with 200 nM Tg and/or 500 nM MG132 for 16 h. In RIP and PLA, HepG2 cells were treated with or without 50 nM Tg and 200 nM MG132 for 16 h. In proteasome activity assays, HepG2 cells were treated with 200 nM Tg or 2 μg/ml Tun (Nacalai Tesque) for 16 h. In protein aggregation assay, HepG2 cells were treated with 50 nM Tg for 10 h followed by confocal microscopy, or with 50 nM Tg for 4 h followed by flow cytometry.

## Transfection

For HEK293 cells, plasmid transfection was performed using polyethyleneimine (PEI)-Max (Polysciences; 24765) according to a previously described protocol (Longo et al, 2013) with minor modification, and siRNA transfection was carried out using Lipofectamine RNAiMAX (Invitrogen; 13778150) according to the manufacturer's instructions. For HepG2 cells, plasmid transfection was performed using either PEI-Max or Lipofectamine 3000 (Invitrogen; L300015) according to the manufacturer's protocol, and siRNA transfection was carried out using either Lipofectamine RNAiMAX or Lipofectamine 3000. The following Stealth siRNAs (Invitrogen) were transfected for 48–72 h: Stealth RNAi siRNA Negative Control Med GC Duplex (12935300), Derlin-1 (MSS228692), Derlin-2 (HSS121486), Derlin-3 (HSS150566), Sec61α (HSS121072), Sec61β (#1-HSS145899, #2-HSS173999, #3-HSS174000, UTR-targeting #1, and UTR-targeting #2), ARIH1 (#1-HSS119341, #2-HSS177850, and UTR-targeting), 4EHP (#1-HSS114097 and #2-HSS190361). 4EHP rescue construct was resistant to siRNA #1 against 4EHP. The siRNAs sequences are shown in the Reagents and Tools Table.

## Morpholino injections

Antisense MO was designed to inhibit against the 5'UTR (Sec61β-atg: aaagctgtgtgttttagacgactggca) of zebrafish *Sec61β* and the control MO carried 5 mismatch bases (Sec61β-atg-5mis: aaacctgtcttata-gaccactcgca). The MOs were injected at 1.2 μg/μL using an IM-30 Electric Micro-injector (Narishige, Tokyo, Japan). For rescue experiments, HA-tagged human ARIH1 lacking Ariadne domain [HA-ARIH1ΔAri(CC)] and ARIH1 mutant with a serine mutation of C357 in Ring2 domain [HA-ARIH1ΔAri(CS)] were cloned into the pCS2+ expression vector. Capped RNA was synthesized with the mMESSAGE mMACHINE SP6 kit (Invitrogen, AM1340) according to the manufacturer's protocol. Capped RNA (0.4 μg/μL) was co-injected with Sec61β-atg MO (1.2 μg/μL) into zebrafish embryos at 1- to 2-cell stages. The MO sequences are shown in the Reagents and Tools Table.

## Video recording behavior and analysis of motor function of zebrafish

Larval behaviors were observed and video-recorded using a dissection microscope (OLYMPUS SZ61) and an iPhone 12 mini attached to microscope with an adaptor. To examine the swimming performance, larvae were video-recorded with their head and yolk

restrained in an agarose gel leaving the trunk and tail free to move at 3 dpf. The movement of tail was captured by a slow mode video at 240 fps (iPhone 12 mini). For quantitative analysis of behaviors, we applied the head-tail angle as defined elsewhere (Naganawa and Hirata, 2011). In brief, a line was drawn at tangent to the rostral trunk of the fish. Then, a second line was drawn at a tangent to the tail of the fish. The angle between these two lines was estimated as the head-tail angle. The total body length of zebrafish larvae, defined as the distance from the mouth to the tip of the tail, was measured at 3 dpf. The behavior of the zebrafish was observed at 4 dpf, and a phenotype score was recorded manually. The phenotype score was defined by four levels as follows: no obvious abnormality (0), abnormal swimming while shaking head and slightly shorter trunk (1), abnormal swimming and morphology (2), and no swimming and abnormal morphology (3).

## Immunoprecipitation (IP)

Cells were lysed on ice in lysis buffer (20 mM Tris-HCl pH 7.5, 150 mM NaCl, 5 mM EGTA, 1% Triton X-100, and 12 mM β-glycerophosphate) containing 5 µg/ml leupeptine. For secreted TTR, samples were prepared by centrifuging the culture medium. The cell lysates or culture medium samples were immunoprecipitated (IPed) with anti-Flag M2 antibody affinity gel (Sigma-Aldrich, A2220, AB_10063035) and HA-tagged protein purification gel (MBL, 3321) or antibodies against Sec61β, Derlin-1, Derlin-2 and TTR using Protein G Sepharose (GE Healthcare, 17-0618-02). The beads were washed with high-salt buffer (20 mM Tris-HCl pH 7.5, 500 mM NaCl, 5 mM EGTA, and 1% Triton X-100) and/or low-salt buffer (20 mM Tris-HCl pH 7.5, 150 mM NaCl, and 5 mM EGTA), resolved by SDS-PAGE and immunoblotted with antibodies. The proteins were detected with the ECL system. Aliquots of whole cell lysates were immunoblotted with antibodies.

## Immunoblotting (IB)

Cell lysates were resolved on SDS-PAGE and blotted onto PVDF membranes. After blocking with 5% skim milk in TBS-T (50 mM Tris-HCl pH 8.0, 150 mM NaCl, and 0.05% Tween-20), the membranes were probed with antibodies. The proteins were detected with the ECL system. Band intensity was measured by ImageQuant TL (GE Healthcare).

## Glycosidase digestion

Cell lysates were incubated with denaturing buffer containing 0.5% SDS and 40 mM DTT at 98 °C for 10 min and with Endoglycosidase H (Endo-H, New England Biolabs, P0702) at 37 °C for 1 h. The reaction was stopped with the addition of SDS-PAGE sample buffer.

## Pulse labeling assay

HEK293 cells transfected with siRNAs and Flag-NHK$^{QQQ}$-HA, NHK$^{QQQ}$-Flag, Flag-TTR-HA, BiP- $^{2×}$Flag-KDEL, or Venus-CL1-Flag and treated with 50 nM Tg and 200 nM MG132 for 16 h were labeled with [$^{35}$S]-methionine/cysteine (EXPRE$^{35}$S$^{35}$S Protein Labeling Mix [$^{35}$S], Revvity, NEG772) for 3, 10, or 15 min in DMEM (Gibco, 21013024) lacking methionine and cysteine and containing

1 mM Sodium Pyruvate and 2 mM L-glutamine. For pulse-chase experiments, cells were stimulated with 50 nM Tg for 16 h, labeled with [$^{35}$S]-methionine/cysteine for 15 min, washed with PBS and chased in medium containing 50 nM Tg, methionine and cysteine for the indicated time periods. Cells were immediately washed with PBS and lysed on ice in lysis buffer containing 5 µg/ml leupeptine. Antibody against Flag or HA was used for IP. IPed samples were resolved by SDS-PAGE. Band radioactivity was measured using a Typhoon FLA7000 image analyzer (GE Healthcare) and quantified using ImageQuant TL.

## Puromycin labeling assay

HEK293 cells were treated with 2 µg/mL puromycin for 10 min, immediately washed with PBS and lysed. The puromycin-labeled proteins were IPed with anti-Flag M2 antibody affinity gel. IPed samples were resolved by SDS-PAGE and analyzed by IB with anti-puromycin and anti-Flag antibodies.

## RNA immunoprecipitation (RIP) and reverse transcription-quantitative PCR (RT-qPCR)

To immunoprecipitate $^{3×}$Flag-eIF4E-bound or Flag-4EHP-bound mRNAs, HepG2 cells on 15-cm plates were reverse transfected with or without siRNAs using Lipofectamine RNAiMAX, followed the next day by transfection with $^{3×}$Flag-eIF4E or Flag-4EHP using PEI-Max. The cells were treated with or without 50 nM Tg and 200 nM MG132 for 16 h, washed with PBS, and lysed on ice in lysis buffer A (50 mM HEPES-KOH pH 7.4, 2 mM EDTA, 10 mM pyrophosphate, 10 mM β-glycerophosphate, 40 mM NaCl, 1% Triton X-100) containing protease inhibitor cocktail (Nacalai Tesque, 25955-11) and RNasin Plus RNase inhibitor (Promega, N261A). The cell lysates were clarified by centrifugation at 14,000 rpm for 10 min at 4 °C. Protein concentration was measured by BCA protein assay (Takara-Bio, T9300A) and 2–4 mg of cell lysate was IPed with anti-Flag M2 antibody affinity gel for 1 h at 4 °C. The beads were washed twice with lysis buffer A, twice with buffer B (15 mM HEPES-KOH pH 7.4, 7.5 mM MgCl$_2$, 100 mM KCl and 1% Triton X-100) containing 2 mM DTT and resuspended in 100 µl buffer B. In total, 10 µl of the resuspension was used for IB and the remainder was used for RNA extraction with RNAiso Plus (Takara, 9109). For input samples, 100 µg of each cell lysate was used for IB and RNA extraction with RNAiso Plus. cDNA of the input and IPed samples was synthesized from purified RNA using RevaTra Ace qPCR RT Master Mix with gDNA Remover (TOYOBO, FSQ-301). qPCR was performed using a StepOnePlus Real-Time PCR System (Applied Biosystems) with KAPA SYBR FAST qPCR Master Mix (KAPA BIOSYSTEMS, KK4605). β-actin mRNA in the same sample was used as a normalization control for Flag-eIF4E-bound mRNA. The levels of Flag-4EHP-bound mRNA were normalized to the corresponding input mRNA. Sequences of the used primers are shown in the Reagents and Tools Table.

## Polysome profiling

HEK293 cells transfected with control or Sec61β siRNA on 15-cm plates were washed with cold PBS including 100 µg/mL cyclohex-imide (CHX) after stimulation with 50 nM Tg and 200 nM MG132 for 16 h and treatment with 100 µg/mL CHX for 5 min, and lysed

on ice in polysome lysis buffer (20 mM Tris pH 7.4, 150 mM NaCl, 5 mM MgCl$_2$ and 1% Triton X-100) containing 100 µg/mL CHX, 1 mM DTT, 5 µg/ml leupeptine and RNasin Plus RNase inhibitor. The cell lysates were clarified by centrifugation and layered onto a 10–50% sucrose gradient in polysome lysis buffer containing 100 µg/mL CHX, 1 mM DTT, 5 µg/ml leupeptine and RNasin Plus RNase inhibitor, prepared using a gradient master (BioComp). Samples were centrifuged for 2 h 45 min at 36,000 rpm in a SW41 rotor and collected from the top using a fraction collector (ATTO) while measuring absorbance at 254 nm on a UV monitor (ATTO).

## Proteasome activity assays

HepG2 cells transfected with siRNAs were washed with PBS and lysed in CHAPS buffer (50 mM Tris-HCl pH 7.5, 100 mM NaCl, 0.2% CHAPS, 5 mM EDTA, 1 mM EGTA) containing 5 µg/ml leupeptine and protease inhibitor cocktail (Nacalai Tesque, 25955-11) after stimulation with 200 nM Tg or 2 µg/mL Tm for 16 h, and cell extracts were clarified by centrifugation. For measurement of the chymotrypsin-like peptidase activity of the proteasome, the cell extracts were incubated with CHAPS buffer containing 100 µM Succinil-Leu-Leu-Val-Tyr-7 amino-4-methylcoumarin (Suc-LLVY-AMC; Calbiochem, 539142) and 1 mM DTT for 20 min at 37 °C. Fluorescence of the released AMC was measured with an excitation wavelength of 365 nm and emission wavelength of 465 nm by a multimode detector DTX800 (Beckman Coulter). The fluorescence intensity was normalized to cell viability in each condition. The cell viability was quantified using a Cell Counting Kit-8 (Dojindo Laboratories) by a multimode detector DTX800 according to the manufacturer's protocol.

## Protein aggregation assay by confocal microscopy and flow cytometry

To visualize protein aggregation in HepG2 cells, a ProteoStat Aggresome detection kit (Enzo Life Sciences, ENZ-51035-K100) was used. HepG2 cells were reverse transfected with siRNAs using Lipofectamine RNAiMAX and cultured on 12-mm-diameter glass coverslips (Fisher Scientific) for 3 days. After stimulation with 50 nM Tg for 10 h, cells were washed with PBS, fixed with 4% paraformaldehyde for 10 min at room temperature, permeabilized in 0.1% Triton X-100 for 5 min at room temperature, blocked with 1% BSA in PBS for 1 h at room temperature, and then incubated with or without anti-calnexin antibody for 12 h at 4 °C. After washing with PBS, the cells were incubated with ProteoStat dye (1:2000 dilution) in the presence or absence of CF647-conjugated anti-rabbit IgG for 1 h at room temperature. The cells were washed with PBS and mounted using Prolong Diamond antifade reagent with DAPI (Invitrogen, P36962). Fluorescence images were obtained using confocal laser microscopy (TSC-SP8, Leica).

To quantify the relative amount of protein aggregation, HepG2 cells were analyzed using ProteoStat Aggresome detection kit and flow cytometry. HepG2 cells were transfected with siRNAs using Lipofectamine RNAiMAX or co-transfected with siRNAs and ³×Flag-Sec61β or ³×Flag-Sec61βΔIDR using Lipofectamine 3000, cultured on 10-cm plates for 3 days. After stimulation with 50 nM Tg for 4 h, cells were washed with PBS, trypsinized, collected in 15 ml tube, washed with PBS and pelleted by centrifugation. Cells resuspended in 150 µl of PBS were gradually added to 1 ml of 4% paraformaldehyde with vortexing and fixed for 20 min at room temperature. After washing with PBS by centrifugation, cell pellet was resuspended in 150 µl of PBS, gradually added to 1 ml permeabilizing buffer (0.5% Triton X-100 and 3 mM EDTA) with vortexing and incubated for 20 min on ice. After washing 3 times with PBS, cells were resuspended in 500 µl of PBS and counted under microscopy. For quantification by flow cytometry, cells adjusted to $1 \times 10^6$ per ml in PBS were incubated with ProteoStat dye (1:5000–1:10,000 dilution) with or without APC anti-DYKDDDDK Tag antibody (1:450 dilution) for 30 min at room temperature in the dark. After washing with PBS, cells were filtered through the cell strainer cap (BD Biosciences, 352235) and analyzed using FACSVerse flow cytometer (BD Biosciences), or CytoFLEX flow cytometer (Beckman Coulter). Flow cytometric data were analyzed with FlowJo software (Tree Star) or CytExpert software (Beckman Coulter).

## Proximity ligation assay (PLA)

HepG2 cells on 12-mm-diameter glass coverslips were transfected with GFP-KDEL, Flag-Sec61β, and HA-ARIH1 or HA-4EHP using Lipofectamine 3000. The cells were treated with 50 nM Tg and 200 nM MG132 for 16 h, washed with PBS, and fixed in 4% paraformaldehyde at room temperature (RT) for 10 min. After permeabilization in 0.1% TritionX-100 at RT for 5 min and blocking with 1% bovine serum albumin in PBS, cells were subjected to PLA Duolink in situ Detection Reagents FarRed Fluorescence kit (Sigma-Aldrich) according to the manufacturer's instructions. Briefly, the cover glass was incubated with the primary antibodies against Flag (MBL, clone FLA-1) and HA (Cell Signaling Technology, clone C29F4) at 4 °C overnight and incubated with the two PLA probes, which are secondary anti-mouse and anti-rabbit antibodies conjugated to unique oligonucleotides, at RT for 1 h. The cover glass including the products followed by ligation and amplification was mounted using Duolink In Situ Mounting Medium with DAPI (Sigma-Aldrich) and visualized using a TCS SP8 confocal microscope (Leica).

## MS sample preparation

For analysis of Derlins-interacting proteins, HEK293 cells transfected with each Flag-tagged Derlin expression plasmid were cultured for 24 h, after which DMSO or Tg (final concentration 50 nM) and MG132 (final concentration 200 nM) were added. After 16 h, the cells were lysed with lysis buffer (20 mM HEPES-NaOH pH 7.5, containing 1% digitonin, 150 mM NaCl, 50 mM NaF, 1 mM Na$_3$VO$_4$, 5 µg/mL leupeptin, 5 µg/mL aprotinin, 3 µg/mL pepstatin A and 1 mM phenylmethylsulfonylfluoride) and centrifuged at $20,000 \times g$ for 10 min. The supernatants were subjected to IP using anti-Flag M2 magnetic beads (Sigma-Aldrich, M8823, AB_2637089). IPed beads were washed three times in washing buffer (10 mM HEPES-NaOH, pH 7.5, 150 mM NaCl and 0.1% Triton X-100) and eluted with Flag peptides. The samples were subjected to trichloroacetic acid precipitation. The resulting pellets were dissolved in 0.1 M ammonium bicarbonate pH 8.8 containing 7 M guanidine hydrochloride, reduced using 5 mM TCEP, and subsequently alkylated using 10 mM iodoacetamide. After alkylation, samples were digested with lysyl-endopeptidase for 3 h at 37 °C and then further digested with trypsin for 14 h at 37 °C.

For analysis of Sec61β-interacting proteins, HEK293T cells transfected with the Flag-Sec61β plasmid were cultured for 24 h,

after which DMSO or Tg (final concentration 50 nM) and MG132 (final concentration 200 nM) were added. After 16 h, the cells were lysed with lysis buffer (20 mM HEPES-KOH pH 7.5, 1% Triton X-100, 150 mM NaCl, Protease inhibitor cocktail [cOmplete EDTA-free protease inhibitor cocktail; Roche, 11873580001], 50 mM NaF and 1 mM $Na_3VO_4$) for 10 min on ice. The cell lysates were centrifuged at $20,000 \times g$ for 10 min at 4 °C, and the supernatants were transferred to a new 1.5 mL tube. Subsequent sample preparation was performed using the general-purpose humanoid robot (LabDroid) "Maholo" (Hayase et al, 2023; Liu et al, 2022; McClymont & Freemont, 2017; Ochiai et al, 2021; Sasamata et al, 2021; Yachie et al, 2017). The supernatants were subjected to IP using Dynabeads Protein G (Thermo Fisher Scientific, 10004D) conjugated with anti-Flag M2 antibody (Sigma-Aldrich, F1804, AB_262044). Magnetic separated beads were washed three times in washing buffer (10 mM HEPES-NaOH pH 7.5, 150 mM NaCl and 0.1% Triton X-100) and eluted with Flag peptides. Subsequent processes were performed in the same method as for sample preparation of Derlins-interacting proteins.

## Liquid chromatography-tandem MS (LC-MS/MS) analysis

For analysis of Derlins-interacting proteins, digested peptide samples were analyzed using a nanoscale LC-MS/MS system as previously described (Natsume et al, 2002). The peptide mixture was applied to a Mightysil-PR-18 (Kanto Chemical) frit-less column (45 × 0.150 mm ID) and separated using a 0–40% gradient of acetonitrile containing 0.1% formic acid for 80 min at a flow rate of 100 nL/min. Eluted peptides were sprayed directly into a Triple TOF 5600+ mass spectrometer (Sciex). MS and MS/MS spectra were obtained using the information-dependent mode. Up to 25 precursor ions above an intensity threshold of 50 counts/s were selected for MS/MS analyses from each survey scan. All MS/MS spectra were searched against protein sequences of NCBI nonredundant human protein dataset (NCBInr RefSeq Release 71, containing 179,460 entries) using the Protein Pilot software package (Sciex). Protein quantification was performed using the iBAQ method (Schwanhausser et al, 2011) without conversion to absolute amounts using universal proteomics standards (iBQ). The iBQ value was calculated by dividing the sum of the ion intensities of all the identified peptides of each protein by the number of theoretically measurable peptides. The mass spectrometry proteomics data have been deposited to the ProteomeXchange Consortium (http://proteomecentral.proteomexchange.org) via jPOSTrepo (Okuda et al, 2017) (PXD054522).

For analysis of Sec61β-interacting proteins, digested peptide samples were analyzed using LC-MS/MS system. MS analysis was performed twice per sample. For LC-MS/MS analyses, an EasynLC1200 system (Thermo Scientific) and Q Exactive HF-X (Thermo Scientific) were used. A C18 silica resin-packed capillary column with a diameter of 10 μM and a length of 12 cm (Nikkyo Technos) was used as the analytical column for LC. Solvents A (0.1% formic acid) and B (0.1% formic acid/80% acetonitrile) were used; peptide separation was performed at a flow rate of 300 nL/min with a gradient of B from 0% to 40% for 80 min. For MS and MS/MS measurement, the Data-Dependent Acquisition (DDA) mode was used. The mass resolution was 60,000 for MS and 15,000 for MS/MS, and the $m/z$ measurement range was 380–1500 for MS

and 200–2000 for MS/MS. The AGC target and Maximum IT were set to 3e6 and 60 ms for MS, and 1e5 and 45 ms for MS/MS, respectively. The MS/MS scan selected 20 precursor ions per MS scan with an exclusion time of 12 s. MS/MS data sets were analyzed by Proteome Discoverer 2.2 software (Thermo Fisher Scientific). Peptide identification was performed using the SEQUEST algorithm with Swiss-Prot (Human: 20,386 entries) as the protein database. The tolerances were specified as 10 ppm and 0.02 Da for precursor and fragment ions, respectively. Trypsin was selected as the digestion enzyme, and up to two missed cleavages were allowed. Oxidation (M), carbamidomethylation (C), and protein N-terminal acetylation were added as modifications. For protein quantification, a label-free quantification method using precursor ions was used. The mass spectrometry proteomics data have been deposited to the ProteomeXchange Consortium (http://proteomecentral.proteomexchange.org) via jPOSTrepo (PXD054523).

## Quantification and statistical analysis

All quantitative data represent the mean of at least three biological replicates. All data are presented as the mean ± SEM. Statistical differences between experimental and control groups were determined using two-tailed Student's $t$ test (for comparisons between two groups) or one-way ANOVA (for comparisons among three or more groups). For analysis of zebrafish phenotype scores, Fisher's exact test with Bonferroni's post hoc correction was applied. Zebrafish body length was analyzed using Kruskal–Wallis rank-sum test. Western blot signals were quantified with ImageQuant TL image analysis software version 8.1 (Cytiva) and Empiria Studio software version 3.0 (LICOR). All statistical analyses were performed using EZR software version 1.54 (Kanda, 2013). The statistical methods and $P$ values are shown in the figure legends.

# Data availability

The mass spectrometry proteomics data related to Figs. 1A and 3A have been deposited in the ProteomeXchange Consortium under accession code PXD054522 and PXD054523, respectively. Any additional information required to reanalyze the data reported in this paper is available from the corresponding author upon request.

The source data of this paper are collected in the following database record: biostudies:S-SCDT-10_1038-S44319-026-00690-y.

# Peer review information

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

## Acknowledgements

We thank Dr. T Inada (University of Tokyo) for valuable discussions, K Mikoshiba (Shanghai Tech University, Toho University) for the GFP-KDEL plasmid, M Watanabe, M Nagatomo, and laboratory members (University of Miyazaki) for technical assistance. A part of this work was conducted in the Frontier Science Research Center, University of Miyazaki. This study was supported by AMED (JP25gm6410024 [HK]), MEXT/JSPS KAKENHI (grant number, 25K10202 [HK], 23K24215 [HN], and 24K21975 [HN]), Kato Memorial Bioscience Foundation (HK), The Ichiro Kanehara Foundation for the Promotion of Medical Sciences and Medical Care (HK), Takeda Science Foundation (HK), Astellas Foundation for Research on Metabolic Disorders (HK), TMDU Nanken-Kyoten grant number 2023 kokunai 12 (HN), and Joint Usage and Joint Research Programs, Institute of Advanced Medical Sciences, Tokushima University (HK).

## Author contributions

**Hisae Kadowaki:** Conceptualization; Formal analysis; Supervision; Funding acquisition; Validation; Investigation; Visualization; Methodology; Writing— original draft; Project administration; Writing—review and editing. **Tomohisa Hatta:** Data curation; Formal analysis. **Kazuma Sugiyama:** Validation; Investigation. **Tomohiro Fukaya:** Formal analysis; Investigation. **Takao Fujisawa:** Data curation; Formal analysis. **Takashi Hamano:** Investigation. **Naoya Murao:** Formal analysis. **Yasunari Takami:** Resources. **Shuya Mitoma:** Formal analysis; Investigation. **Tohru Natsume:** Data curation; Methodology. **Katsuaki Sato:** Supervision; Investigation. **Hiromi Hirata:** Supervision; Investigation. **Tamayo Uechi:** Supervision; Investigation. **Hideki Nishitoh:** Conceptualization; Supervision; Funding acquisition; Investigation; Writing— original draft; Project administration; Writing—review and editing.

Source data underlying figure panels in this paper may have individual authorship assigned. Where available, figure panel/source data authorship is listed in the following database record: biostudies:S-SCDT-10_1038-S44319-026-00690-y.

## Disclosure and competing interests statement

TH is an employee of Robotic Biology Institute Inc. The remaining authors declare no competing interests.

# Expanded View Figures

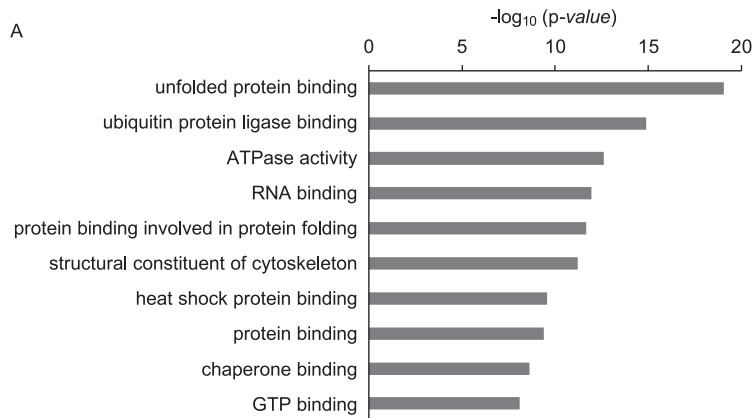

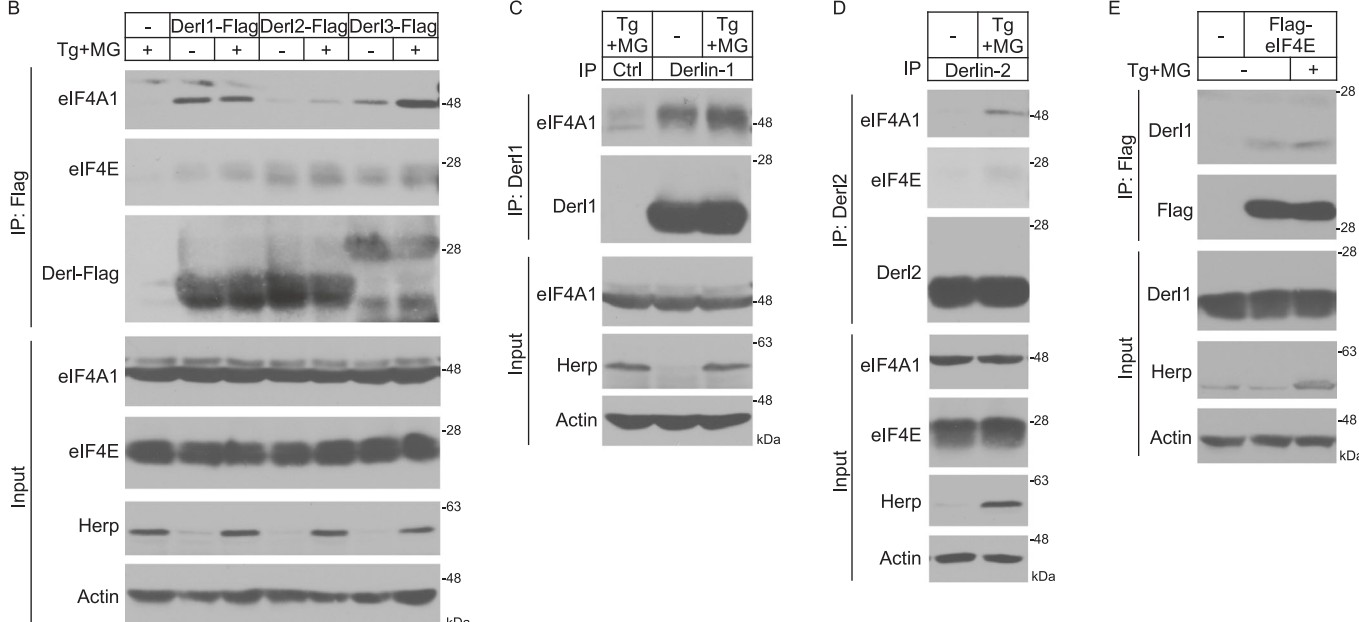

**Figure EV1.  Derlins enhance interactions with translation-related proteins during ER stress.**

(A) Gene Ontology (GO) analysis of Derlins-interacting proteins by thapsigargin (Tg) and MG132 treatment. The bar graph shows the top 10 GO molecular function terms with a false discovery rate of <0.05 calculated from the DAVID online tool. *P* values were calculated using the modified Fisher's exact test implemented in DAVID. eIF4A1 and Sec61β are included among 43 proteins identified as RNA binding in terms of molecular function. Related to Fig. 1A. (B) Interactions of exogenous Derlins with endogenous eIF4A1 and eIF4E. HEK293 cells transfected with indicated plasmids and treated with or without 50 nM Tg and 200 nM MG132 for 16 h were immunoprecipitated (IPed) with an anti-Flag antibody and immunoblotted with indicated antibodies. (C, D) Endogenous interactions of Derlin-1 (C) or Derlin-2 (D) with eIF4A1 (C, D) and eIF4E (D). HepG2 cells treated with or without 200 nM Tg and 500 nM MG132 for 16 h were IPed with an anti-Derlin-1 or an anti-Derlin-2 antibody and immunoblotted with indicated antibodies. (E) Interaction of endogenous Derlin-1 with exogenous eIF4E. HEK293 cells transfected with Flag-eIF4E and treated with or without 50 nM Tg and 200 nM MG132 for 16 h were IPed with an anti-Flag antibody and immunoblotted with indicated antibodies.

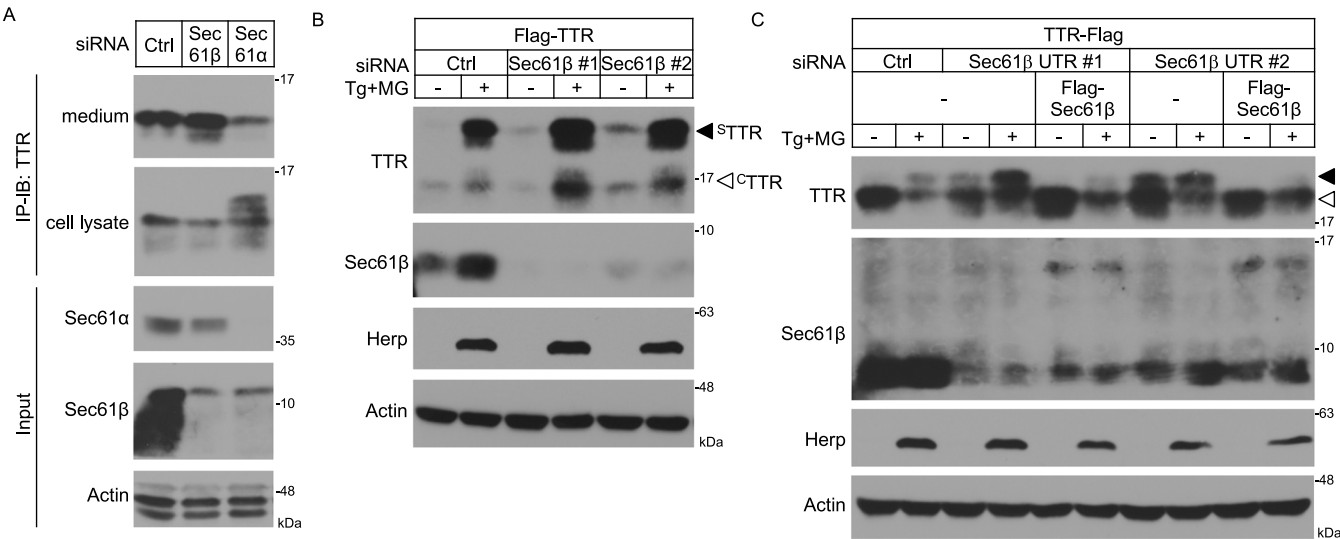

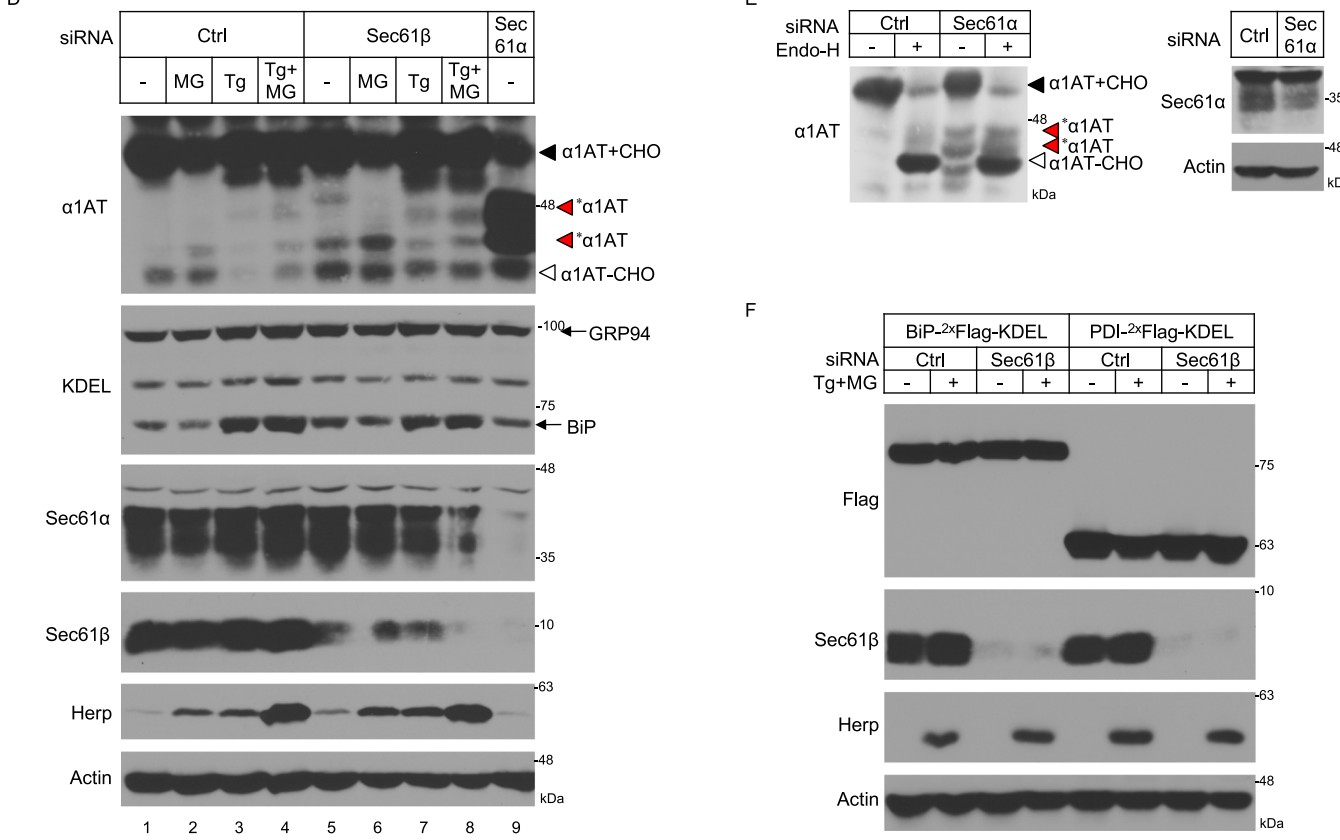

**Figure EV2.  Sec61β-mediated inhibition of ERpQC substrate accumulation.**

(A) Immunoprecipitation (IP) and immunoblotting (IB) of secreted TTR into the culture medium and intracellular TTR in HepG2 cells transfected with indicated siRNAs. All samples were immunoblotted with indicated antibodies. (B, C) IB of ERpQC substrate in HEK293 cells transfected with indicated siRNAs and plasmids and treated with or without 50 nM Tg and 200 nM MG132 for 16 h; samples were immunoblotted with indicated antibodies. Black arrowhead, signal peptide-uncleaved TTR ($^S$TTR); white arrowhead, signal peptide-cleaved TTR ($^C$TTR). (D) IB of α1AT in HepG2 cells transfected with indicated siRNAs and treated with or without 200 nM Tg and/or 500 nM MG132 in the indicated combinations for 16 h. Red arrowheads, untranslocated α1AT (˙α1AT); black arrowhead, glycosylated α1AT (α1AT + CHO); white arrowhead, deglycosylated α1AT (α1AT-CHO). (E) Left: IB of α1AT in Endo-H-treated lysates of HepG2 cells transfected with indicated siRNAs. The untranslocated α1AT was resistant to deglycosylation enzymes. Right: IB of Sec61α and actin in HepG2 cells transfected with indicated siRNAs. Red arrowheads, ˙α1AT; black arrowhead, α1AT + CHO; white arrowhead, α1AT-CHO. (F) IB of Flag-BiP and Flag-PDI in HEK293 cells transfected with indicated siRNAs and cDNAs and treated with or without 50 nM Tg and 200 nM MG132 for 16 h. All samples were immunoblotted with indicated antibodies.

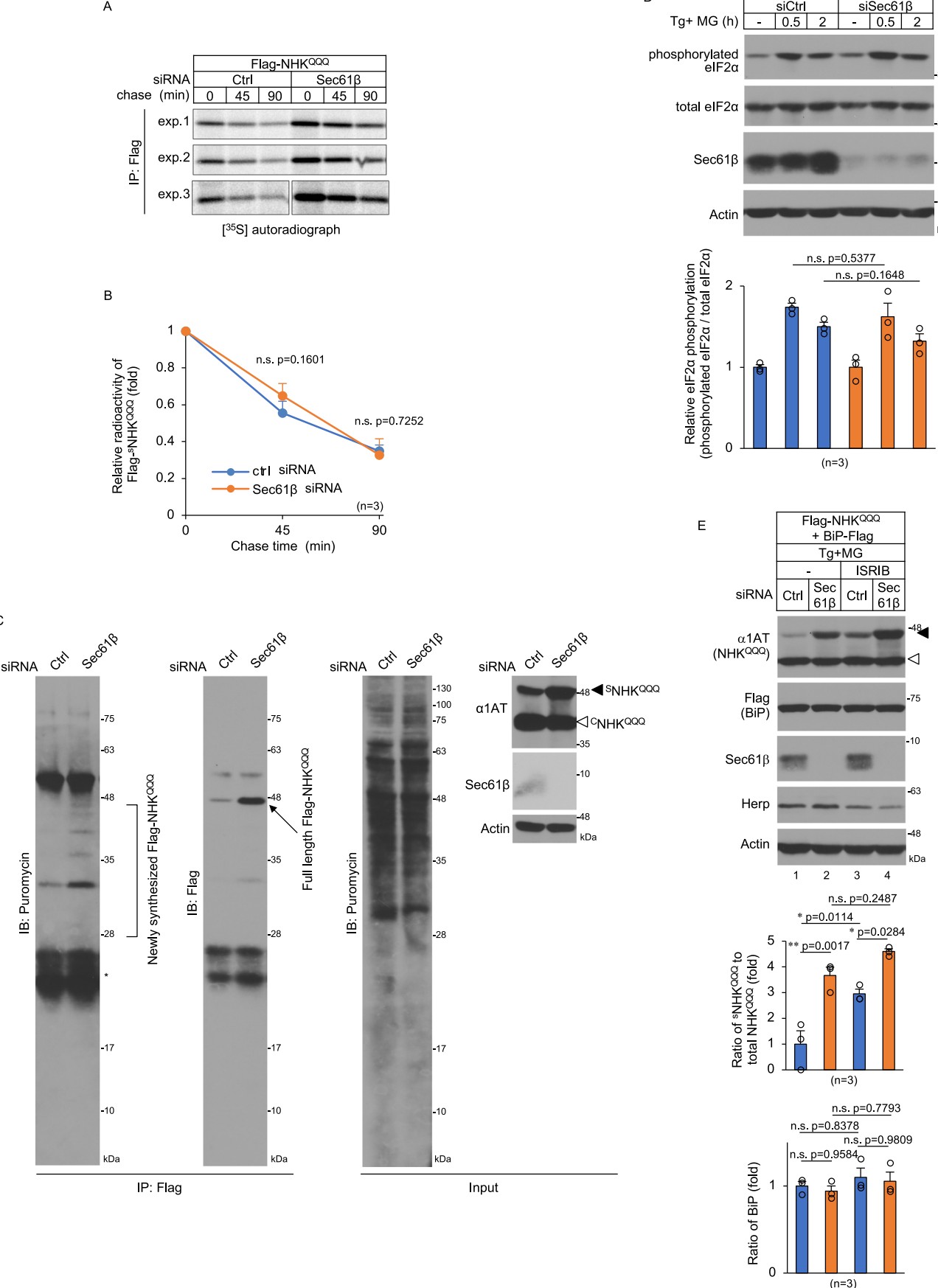

◄ **Figure EV3.   Inhibition of ERpQC substrate accumulation by Sec61β through translational repression rather than accelerated degradation.**

(A) Degradation of Flag-tagged signal peptide-uncleaved NHK$^{QQQ}$ ($^S$NHK$^{QQQ}$) after 15 min pulse of [$^{35}$S]-methionine/cysteine metabolic labeling, followed by the indicated periods of chase time. Cell lysates of HEK293 cells transfected with indicated siRNAs and Flag-NHK$^{QQQ}$ and stimulated with 50 nM Tg for 16 h were IPed with an anti-Flag antibody, resolved by SDS-PAGE and analyzed by autoradiography. (B) The relative radioactivity of Flag-NHK$^{QQQ}$ at different times of chase was calculated and shown as fold decreases relative to the intensity observed at 0 h chase ($n = 3$ biological replicates). (C) Newly synthesized Flag-NHK$^{QQQ}$ labeled with puromycin for 10 min followed by IP with an anti-Flag antibody and IB with indicated antibodies in HEK293 cells transfected with indicated siRNAs and Flag-NHK$^{QQQ}$ and treated with 50 nM Tg and 200 nM MG132 for 16 h. Black arrowhead, $^S$NHK$^{QQQ}$; white arrowhead, $^C$NHK$^{QQQ}$. (D) IB of phosphorylated and total eIF2α in HepG2 cells transfected with indicated siRNAs and treated with or without 200 nM Tg and 500 nM MG132 for 0.5 and 2 h; samples were immunoblotted with indicated antibodies. Bar graph: ratio of phosphorylated eIF2α to total eIF2α was calculated and shown as fold change relative to the intensity observed in the absence of Tg and MG132 for each knockdown ($n = 3$). (E) IB of ERpQC substrate in HEK293 cells transfected with indicated siRNAs and indicated plasmids, pretreated with 200 nM ISRIB for 3 h, and subsequently treated with 50 nM Tg and 200 nM MG132 for 16 h; samples were immunoblotted with indicated antibodies. Bar graph: ratio of the expression level of $^S$NHK$^{QQQ}$ to the total amount of NHK$^{QQQ}$ was calculated and shown as fold change relative to the intensity observed in siCtrl-transfected cells without ISRIB treatment ($n = 3$). Black arrowhead, $^S$NHK$^{QQQ}$; white arrowhead, $^C$NHK$^{QQQ}$. Data are means ± SEM. *$P < 0.05$, **$P < 0.01$; n.s., not significant. Two-tailed unpaired $t$ test for siCtrl vs siSec61β (B, D); one-way ANOVA with Tukey's multiple comparisons test (E).

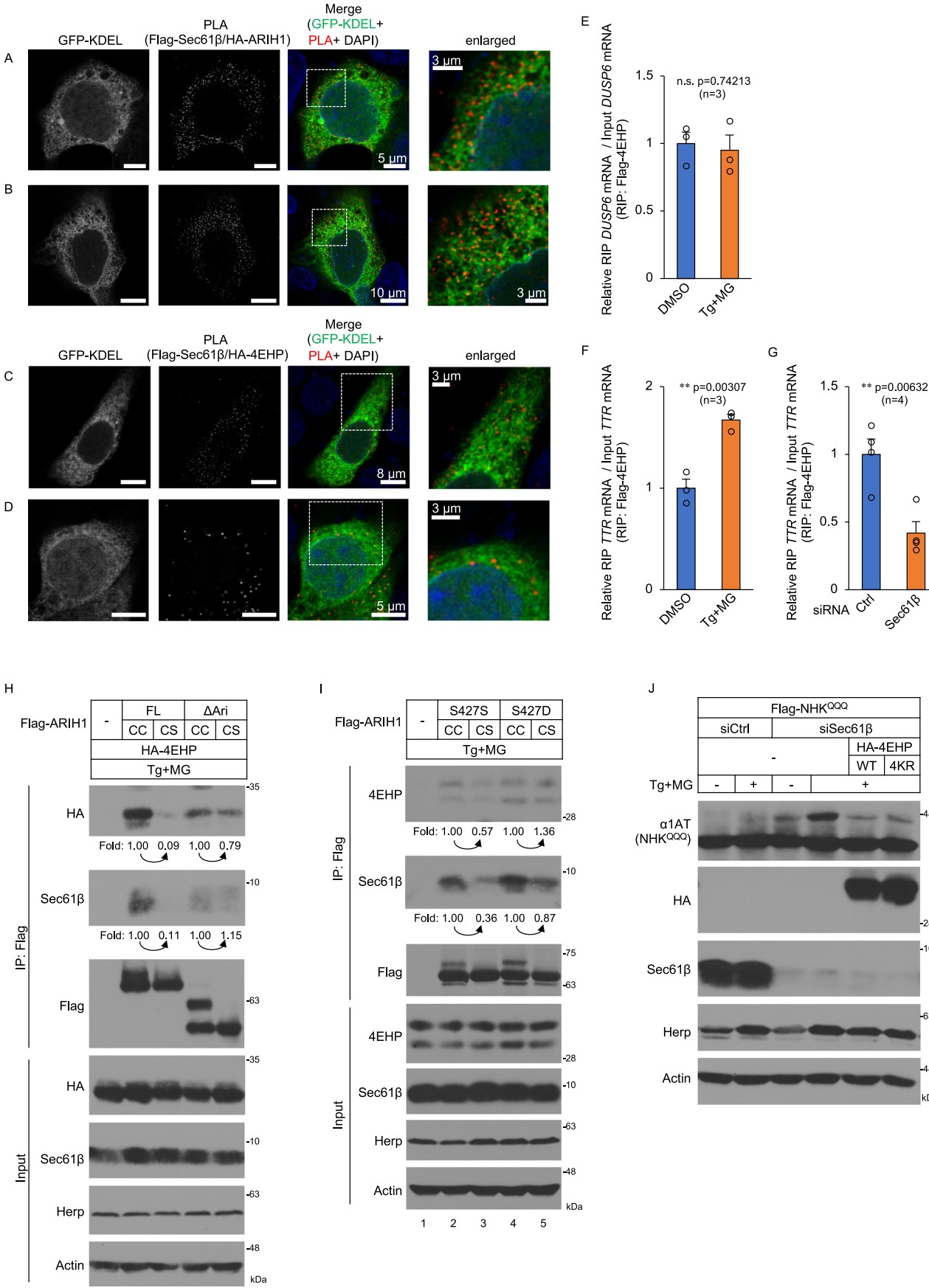

**Figure EV4. Sec61β–ARIH1–4EHP complex formation and 4EHP interaction with ERpQC substrate mRNAs.**

(A–D) Representative fluorescence images of interaction between Sec61β and ARIH1 (**A, B**) or 4EHP (**C, D**) detected by a proximity ligation assay (PLA). HepG2 cells transfected with GFP-KDEL, Flag-Sec61β and HA-ARIH1 (**A, B**) or HA-4EHP (**C, D**) were treated with 50 nM Tg and 200 nM MG132 for 16 h. PLA was performed using anti-Flag and anti-HA antibodies. GFP-KDEL (green, ER marker), PLA signal (red, Flag/HA) and DAPI (blue, nuclei) are shown. Scale bars, 3–10 μm. (**E–G**) RNA immunoprecipitation (RIP) of Flag-4EHP with *DUSP6* mRNA (**E**) and ERpQC substrate *TTR* mRNA (**F**). RIP of Flag-4EHP with *TTR* mRNA in Sec61β-deficient cells (**G**). HepG2 cells were transfected with Flag-4EHP alone (**E, F**) or together with siCtrl or siSec61β (**G**) and treated with or without 50 nM Tg and 200 nM MG132 for 16 h (**E, F**) or with 50 nM Tg and 200 nM MG132 for 16 h (**G**). Flag-4EHP was IPed using an anti-Flag antibody, and the levels of the indicated mRNAs (normalized to input) in Flag-4EHP-bound mRNA were analyzed by RT-qPCR (**E, F**; $n = 3$, **G**; $n = 4$). (**H, I**) Interaction of exogenous (**H**) or endogenous (**I**) 4EHP and endogenous Sec61β with ARIH1 mutants upon ER stress. IP with an anti-Flag antibody and IB with indicated antibodies in HEK293 cells transfected with indicated plasmids and treated with 50 nM Tg and 200 nM MG132 for 16 h. The amounts of co-IPed proteins with Flag-ARIH1 were normalized by the amount of IPed Flag-ARIH1 and the input amount of each protein and shown as fold changes relative to the intensity observed in IP with Flag-ARIH1(CC) retaining its E3 ligase activity. Details regarding the ARIH1 mutants are described in Fig. 3B. ΔAri, mutant lacking the inhibitory Ariadne domain; CS, catalytically inactive serine mutant of Cys357; S427D, phospho-mimetic mutant of Ser427 to Asp on the Ariadne domain. (**J**) IB of ERpQC substrate in HEK293 cells transfected with indicated siRNAs and plasmids and then treated with or without 50 nM Tg and 200 nM MG132 for 16 h; samples were immunoblotted with indicated antibodies. 4KR, non-ISGylatable 4EHP mutant (Lys121/130/134/222 to Arg). Data are means ± SEM. *$P < 0.05$, **$P < 0.01$; n.s., not significant. Two-tailed unpaired *t* test for DMSO vs Tg+MG (**E, F**) and for siCtrl vs siSec61β (**G**).

A

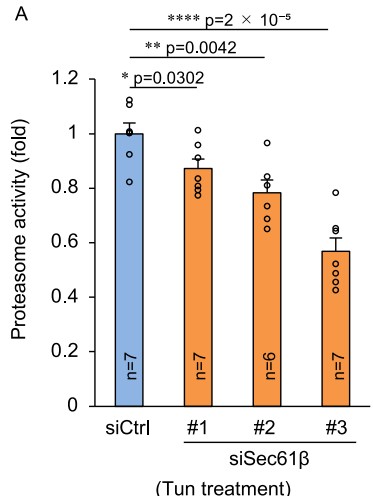

**** p=2 × 10⁻⁵
** p=0.0042
* p=0.0302

B

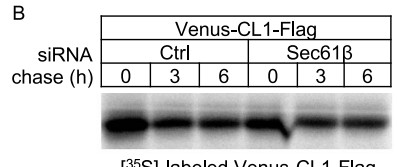

[³⁵S]-labeled Venus-CL1-Flag

C

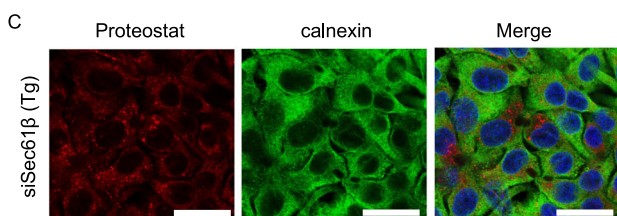

D

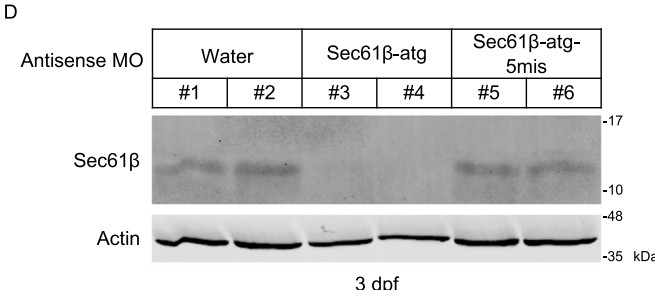

3 dpf

E

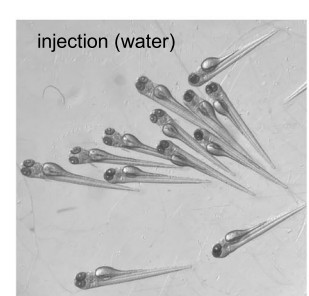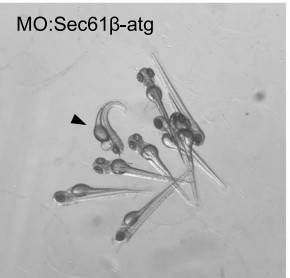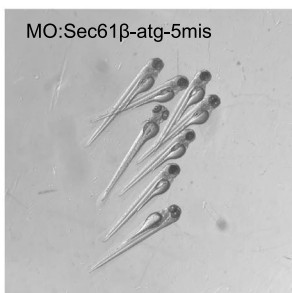

F

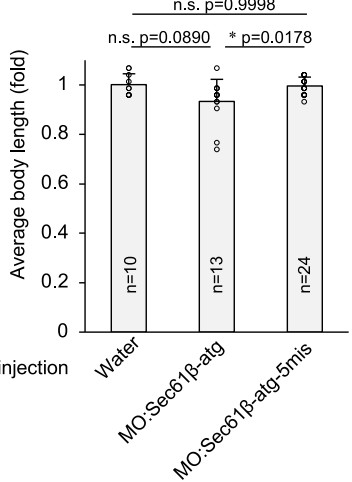

G

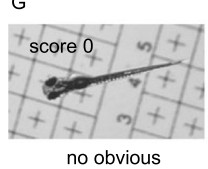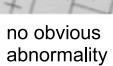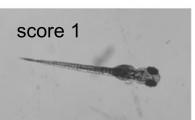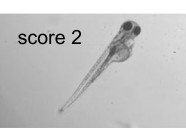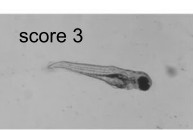

H

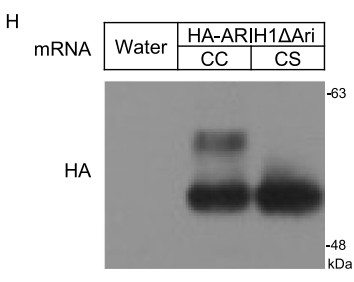

10 fish/lane

◀

**Figure EV5.  Sec61β-mediated maintenance of cytoplasmic proteostasis upon ER stress and characterization of phenotypes in zebrafish.**

(A) Proteasome chymotrypsin-like peptidase activity of cell extracts from HepG2 cells transfected with siCtrl or siSec61β (#1, #2, or #3) and treated with 2 μg/ml tunicamycin (Tun) for 16 h was measured using Suc-LLVY-AMC as a substrate. Fluorescence intensity was normalized to cell viability in each condition. Proteasome activity is shown as fold decrease relative to that of siCtrl-transfected cells ($n = 7, 7, 6,$ and 7 from left to right, respectively). (B) Degradation of CL1 degron chased for the indicated periods after 15 min pulse of [$^{35}$S]-methionine/cysteine metabolic labeling shown in Fig. 6B. Cell lysates of HEK293 cells transfected with indicated siRNAs and Venus-CL1-Flag and stimulated with 50 nM Tg for 16 h were IPed with an anti-Flag antibody, resolved by SDS-PAGE and analyzed by autoradiography. (C) Representative fluorescence images of HepG2 cells transfected with siSec61β and stimulated with 50 nM Tg for 6 h, followed by staining with ProteoStat (red, protein aggregation), calnexin (green, ER membranes) and DAPI (blue, nuclei). Scale bars, 25 μm. (D) Reduced expression of Sec61β in zebrafish by injection of antisense morpholino oligonucleotide (MO). Antisense MO was designed as described in Methods. MOs were injected into zebrafish embryos at 1- to 2-cell stages. The expression of Sec61β in zebrafish at 3 dpf was analyzed by IB using a polyclonal antibody against a peptide against zebrafish Sec61β (SAGTGGMWRFYTEDSPGLKV) raised in rabbits. Lysates were prepared by homogenizing 3 dpf fish for 60 s in lysis buffer (20 mM Tris-HCl pH 7.5, 150 mM NaCl, 5 mM EGTA, and 1% Triton X-100) supplemented with 5 μg/mL leupeptin (Nacalai Tesque; 43449-62) on ice using a Micro Smash (TOMY; MS-100) (4500 rpm, 4 °C). Lysates were resolved by SDS-PAGE and blotted onto polyvinylidene fluoride (PVDF) membranes. After blocking with 5% skim milk in TBS-T (50 mM Tris-HCl pH 8.0, 150 mM NaCl, and 0.05% Tween-20), the membranes were probed with antibody against to zebrafish Sec61β (1/1000) or actin (1/5000) diluted in 5% BSA in TBS-T overnight at 4 °C. Secondary antibodies [IRDye 800CW Donkey anti-Rabbit IgG (H + L) (1/10,000) and IRDye 680RD Donkey anti-Mouse IgG (H + L) (1/10,000)] were diluted in 5% skim milk in TBS-T, and membranes were incubated 2 h at room temperature. Images were revealed and analyzed using Odyssey CLx (LICOR) and Empiria Studio software 3.0 (LICOR). (E) Representative images showing the typical morphology of zebrafish larvae injected with water, Sec61β-atg MO or Sec61β-5mis MO at 3 dpf. Arrowhead indicates the abnormal morphology observed in rare Sec61β-deficient zebrafish. (F) Histogram showing the average body length of zebrafish larvae relative to that of water-injected controls at 3 dpf. Sec61β-deficient zebrafish ($n = 13$) showed a tendency toward reduced body length compared to control groups ($n = 10$ for water injection or $n = 24$ for Sec61β-atg-5mis MO injection). (G) The behavior of zebrafish was observed at 4 dpf, and a phenotype scores were recorded manually. The scoring criteria were defined as follows: no obvious abnormality (0), abnormal swimming with head shaking and slightly smaller size (1), abnormal swimming and morphology (2), and no swimming and abnormal morphology (3). (H) Exogenous expression of ARIH1 mutants in zebrafish by injection of synthesized mRNAs. Synthesized mRNAs for human ARIHΔAri(CC) or (CS) were co-injected into zebrafish embryos at 1- to 2-cell stages. Lysates were prepared by homogenizing 3 dpf 10 fish for 60 s in lysis buffer due to weak expression of exogenous proteins. The expression of HA-ARIHΔAri was analyzed by IB using a rat monoclonal antibody against HA (clone 3F10) and secondary antibody (HRP-linked anti-rat IgG antibody). The membranes were detected by an ECL system, and images were revealed and analyzed using ChemiDoc Touch (BioRad). Data are means ± SEM. *$P < 0.05$, **$P < 0.01$, ****$P < 0.0001$; n.s., not significant. Two-tailed unpaired $t$ test for siCtrl vs siSec61β (A); Kruskal–Wallis rank-sum test (F).

