## [Peer Review File · EMBO Reports]

Sec61 β maintains cytoplasmic proteostasis via ARIH1-mediated translational repression upon ER stress

Hisae Kadowaki, Tomohisa Hatta, Kazuma Sugiyama, Tomohiro Fukaya, Takao Fujisawa, Takashi Hamano, Naoya Murao, Yasunari Takami, shuya Mitoma, Tohru Natsume, Katsuaki Sato, Hiromi Hirata, Tamayo Uechi, and Hideki Nishitoh

Corresponding author(s): Hisae Kadowaki (kadowaki@med.miyazaki-u.ac.jp) , Hideki Nishitoh (nishitoh@med.miyazaki-u.ac.jp)

Review Timeline:

Submission Date:	30th May 25
Editorial Decision:	17th Jul 25
Revision Received:	9th Nov 25
Editorial Decision:	5th Dec 25
Revision Received:	16th Dec 25
Accepted:	8th Jan 26

Editor: Esther Schnapp

Transaction Report:

Dear Dr. Kadowaki,

Thank you for your patience while your ms was peer-reviewed at EMBO reports. We have now received the full set of referee reports as well as referee cross-comments, which are all pasted below.

As you will see, while referee 2 is more critical, both referees 1 and 3 agree that you should be given a chance to revise your ms and address all referee concerns.

I would thus like to invite you to revise your manuscript with the understanding that the referee concerns must be fully addressed and their suggestions taken on board. Please address all referee concerns in a complete point-by-point response. Acceptance of the manuscript will depend on a positive outcome of a second round of review. It is EMBO reports policy to allow a single round of major revision only and acceptance or rejection of the manuscript will therefore depend on the completeness of your responses included in the next, final version of the manuscript.

We realize that it is difficult to revise to a specific deadline. In the interest of protecting the conceptual advance provided by the work, we recommend a revision within 3 months (17th Oct 2025). Please discuss the revision progress ahead of this time with the editor if you require more time to complete the revisions.

- 1) A data availability section providing access to data deposited in public databases is missing. If you have not deposited any data, please add a sentence to the data availability section that explains that.
- 2) Your manuscript contains statistics and error bars based on $n=2$. Please use scatter blots in these cases. No statistics should be calculated if $n=2$.

3) We replaced Supplementary Information with Expanded View (EV) Figures and Tables that are collapsible/expandable online. A maximum of 5 EV Figures can be typeset. EV Figures should be cited as 'Figure EV1, Figure EV2' etc... in the text and their respective legends should be included in the main text after the legends of regular figures.

5) a complete author checklist, which you can download from our author guidelines <https://www.embopress.org/page/journal/14693178/authorguide>. Please insert information in the checklist that is also reflected in the manuscript. The completed author checklist will also be part of the RPF.

6) Please note that all corresponding authors are required to supply an ORCID ID for their name upon submission of a revised manuscript (<https://orcid.org/>). Please find instructions on how to link your ORCID ID to your account in our manuscript tracking system in our Author guidelines <https://www.embopress.org/page/journal/14693178/authorguide#authorshipguidelines>

7) Before submitting your revision, primary datasets produced in this study need to be deposited in an appropriate public database (see <https://www.embopress.org/page/journal/14693178/authorguide#datadeposition>). Please remember to provide a

reviewer password if the datasets are not yet public. The accession numbers and database should be listed in a formal "Data Availability" section placed after Materials & Method (see also <https://www.embopress.org/page/journal/14693178/authorguide#datadeposition>). Please note that the Data Availability Section is restricted to new primary data that are part of this study. * Note - All links should resolve to a page where the data can be accessed. *

12) All Materials and Methods need to be described in the main text using our 'Structured Methods' format, which is required for all research articles. According to this format, the Methods section includes a separate Reagents and Tools Table file (listing key reagents, experimental models, software and relevant equipment and including their sources and relevant identifiers) and a Methods and Protocols section describing the methods using a step-by-step protocol format. The aim is to facilitate adoption of the methodologies across labs. More information on how to adhere to this format as well as a downloadable template (.docx) for the Reagents and Tools Table can be found in our author guidelines:

An example of a Method paper with Structured Methods can be found here: <https://www.embopress.org/doi/full/10.1038/s44320-024-00037-6#sec-4>

I look forward to seeing a revised form of your manuscript when it is ready.

Referee #1:

Kadowagi and colleagues present compelling evidence that Sec61 β mediates translational repression of ERpQC substrates by recruiting the ARIH1 E3 ligase and the eIF4E-homologous protein 4EHP to the 5' cap structure of target mRNAs. The experiments are well controlled, and the results are clean and convincing.

However, several points require further clarification.

First, it remains unclear how the E3 ligase activity of ARIH1 contributes to the translational repression of target mRNAs. Is 4EHP ISGylated in their assay systems?

Additionally, treatment with thapsigargin and MG132 is known to induce phosphorylation of eIF2. Is this phosphorylation event involved in the Sec61 β -mediated translational suppression? What is the effect of ISRIB treatment on the translational activity regulated by Sec61 β ?

Finally, can the authors use imaging approaches to visualize the interaction between Sec61 β , ARIH1, and 4EHP in cells?

Referee #2:

Kadowaki et al. report on a novel role for Sec61 β in maintaining cytoplasmic homeostasis during ER stress via the cap-binding protein 4EHP-mediated selective translational repression of ERpQC substrates. The authors conclude that the translational repression is achieved through the E3 ligase ARIH1 and 4EHP as 4EHP recruitment blocks eIF4E dependent mRNA translation. The authors showed that loss of Sec61 β leads to increased synthesis of ERpQC substrates, impaired proteasome activity, and the accumulation of cytoplasmic aggregates. The authors used a Zebrafish model to show that Sec61 β deficiency leads to motor dysfunction, which can be rescued by exogenous ARIH1 expression, pointing to the physiological relevance of this mechanism. The findings suggest that translational repression of ERpQC substrates by the Sec61 β -ARIH1-4EHP complex is a critical mechanism for coordinating ER and cytoplasmic proteostasis under stress conditions.

While the authors address an important aspect of ER homeostasis and offer a novel mechanism for maintaining proteostasis during ER stress, the paper suffers from major deficiencies including limited experimentation and technical shortcomings as follows:

1. The conclusions of the paper are tenuous. Based on the literature it is difficult to envisage a molecular mechanism by which the 4EHP-ARIH1 would replace the eIF4F at the 5' cap structure. eIF4F exhibits a strong interaction with cap because of its association with eIF4G which binds to the mRNA in the vicinity of the cap. 4EHP binds very weakly to the cap. It is not clear (model, Fig.7) how the complex 4EHP-ARIH1 binds to the cap even if eIF4F abundance is reduced 2-3 fold.
2. The quality of the WBs in many figures is inadequate, as many WB are composites of different gel electrophoresis runs and some lanes are spliced.
 - a. This is evident for Sec61 β (main topic of the paper), Der11, Der12, and α 1AT (NHKQQQ).
 - b. Figure 5F, the input for 4EHP shows two bands, whereas the IP displays only a single faint band, which is not explained.
 - c. Fig 5C: The levels of ARIH1 in input increased in the flag-tagged (lines 3-5), also the amount of the Flag in the IP is more for lines 4 and 5, which could explain the increased interaction with ARIH1 in Tg and the Tg+MG. Also, the flag is not shown in the input.
3. The data presentation is confusing; for instance, Figure EV1 B is referenced before Fig, 1B, which data related only to endogenous Derlin-1 and not Derlin-2 or -3.
4. Important observations are not addressed. For example, Figure EV1C shows that eIF4A interacts with Der12 specifically under ER stress (Tg+MG), but this result is not discussed in the text.
5. Many experiments in the paper lack a rationale

a. P.4; The authors state "eIF4A1 and cap-binding protein eIF4E, which are components of the eIF4F complex together with the scaffold protein eIF4G, interacted with Derlins and these interactions were enhanced under the ER stress conditions (Fig. EV1C-E)". This is out of the blue: what is the rationale? Table EV2 lacks the fold-change value for eIF4A, and the authors do not explain why eIF4A was selected for further analysis.

The IPs were performed only with exogenously produced proteins. This is acceptable when mutant proteins are also studied. Otherwise, they must show the interactions of endogenous proteins here and in the other experiments.

6. Figure 3E, the reason for the use of mouse liver cells is not explained. While including this model could add value, it requires proper introduction and contextualization.

7. The authors switch from using (Tg) and (MG132) to tunicamycin without justification.

8. The rationale for the use of the swimming performance test is unclear. The authors neither explain why they chose to assess motor function specifically, nor do they reference any existing literature suggesting that Sec61 β affects motor abilities.

9. The introduction relies heavily on self-citations, with only three references not authored by the same group.

Referee #3:

ERpQC is a selective proteostasis mechanism by which certain proteins are re-routed from the ER translocon to the cytoplasmic degradation pathway. Previous studies by the authors established roles of Derlin-1, -2, -3 in ERpQC, but their mechanism of function had remained poorly understood. In this study, Kadowaki and colleagues show that Derlin-1, -2, -3 physically interact with Sec61b, and Sec61b specifically inhibits the accumulation of ERpQC substrates such as A1AT and TTR (without affecting secretory protein accumulation). Sec61b specifically regulates the translation of ERpQC substrates, and Sec61b physically interacts with the E3 ubiquitin ligase ARIH1. Derlins and ARIH1 regulate translation by recruiting 4EHP, a cap-binding protein that blocks eIF4E-mediated translation. The over-production of ERpQC in response to Sec61b loss impairs proteasome capacity. Consistently, knockdown of Sec61b in zebrafish causes motor impairment, which is suppressed by expressing ARIH1.

Overall, this is an interesting study that identifies new regulators of ERpQC. The study combines *ex vivo* and *in vivo* experiments to show detailed mechanisms and physiological functions. The experiments are well designed. The authors' ERpQC assays have internal controls of substrates that enter the secretory pathway (which are not affected by ERpQC impairment) and a higher molecular weight product that are specifically affected by ERpQC block. siRNA experiments have independent siRNAs and rescue experiments. The blots are of very high quality, and all major data are quantified. The quality of this manuscript is outstanding, and I have no major concerns.

Cross-comments from referee 1 :

I liked the work in the manuscript. If Reviewer #2 has had some concerns about the preparation of the images, this can be resolved by the provision of raw data, something that Nature publishing group always requests from the authors. Regarding the rest of the comments of Reviewer #2, I believe that everyone deserves the benefit of the doubt. In my opinion, it would be fair to offer the opportunity to the authors to respond to the comments of this Reviewer rigorously.

Cross-comments from referee 3:

It seems to me that Reviewer #2's specific points are less about the overall significance, but rather, about a few technical points that could be addressed by the authors. I would vote to give the authors a chance to respond. Below are some specific points:

"Conclusions are tenuous." Reviewer #2 points out that 4EHP has a weaker affinity to the 5' caps of mRNAs compared to eIF4E *in vitro*, and based on this, disagrees with the authors' model that 4EHP can replace eIF4E at the 5' cap. But, I wonder if we can rule out the idea of 4EHP binding to 5' caps of certain mRNAs just because of the lower affinity measured *in vitro*. The published literature clearly supports the idea that 4EHP, a cap-binding protein, binds to a small number of mRNAs to exert a biological role. Perhaps the authors can strengthen their model by providing additional supporting data for 4EHP binding to ERpQC mRNAs.

"The quality of the WBs is inadequate, as many WBs are composites of different gel electrophoresis." I went over the Figures again, and I couldn't find signs of piecing together different gels in a single blot. Maybe I have a lower resolution image. Perhaps your journal could check through image analysis on this point.

Other points appear to be technical and addressable by the authors.

We would like to express our sincere gratitude to the reviewers for their insightful and constructive comments. We have carefully considered all the points raised and conducted additional experiments to address most of them. Details regarding our responses can be found in the point-by-point reply below.

Blue Arial font: comments from referees.

Black Times New Roman font: our responses.

The mass spectrometry proteomics data associated with Figures 1A and 3A have been deposited in the ProteomeXchange Consortium under the following access codes.

PXD054522 (<https://repository.jpostdb.org/preview/472977770690eb697990f2>)

PXD054523 (<https://repository.jpostdb.org/preview/1271889182690eb69651955>)

These datasets are currently under controlled access and will be made publicly available upon publication. Editors and reviewers can access the data during the peer-review process using the following access keys: PXD054522 [2255] and PXD054523 [6918].

Comments from referee #1:

Kadowagi and colleagues present compelling evidence that Sec61 β mediates translational repression of ERpQC substrates by recruiting the ARIH1 E3 ligase and the eIF4E-homologous protein 4EHP to the 5' cap structure of target mRNAs. The experiments are well controlled, and the results are clean and convincing.

However, several points require further clarification.

First, it remains unclear how the E3 ligase activity of ARIH1 contributes to the translational repression of target mRNAs. Is 4EHP ISGylated in their assay systems?

Re) We appreciate the reviewer's thoughtful comment on our manuscript. As the reviewer pointed out, the ISGylation of 4EHP and the importance of its ISGylation sites (K121, K130, K134, and K222) in suppressing translation initiation have been reported (Okumura *et al.*, 2007). Although we examined whether 4EHP is ISGylated under ER stress conditions in HEK293 cells, no clear ISGylation was detected for exogenously expressed 4EHP (reference Figure 1). Moreover, non-ISGylatable 4EHP mutant (4KR; K121R/K130R/K134R/K222R) clearly inhibited the accumulation of ERpQC substrates in Sec61 β knockdown cells to a similar extent as wild-type 4EHP (revised Figure EV4J). These results suggest that ISGylation of 4EHP is unlikely to contribute to the translational repression of ERpQC substrates. Nevertheless, we do not yet have conclusive evidence that completely rules out the possibility that ISGylation or other posttranslational modifications of 4EHP may contribute to this process. This remains an important issue for future investigation, and we have carefully discussed this point in the revised manuscript (p9 line 34–p10 line 4, p14 lines 26–30, revised Figure EV4J).

Reference Figure 1 (data not shown in the revised manuscript)

Our current findings suggest that the E3 ligase activity of ARIH1 is crucial for both the formation of the Sec61 β -ARIH1-4EHP complex and the inhibition of ERpQC substrate accumulation (revised Figures 4F, 5A, EV4H, lanes 2,3). The E3 ligase activity of ARIH1 is autoinhibited by its C-terminal Ariadne domain, which masks the catalytic cysteine (C357) under basal conditions. A conformational change to open this catalytic site is required for its activation (Duda *et al*, 2013). To elucidate the relationship between this conformational change and complex formation, we generated a constitutively active ARIH1 mutant lacking the inhibitory Ariadne domain [ARIH1 Δ Ari(CC)], and its E3 ligase-inactive mutant bearing a serine substitution of C357 [ARIH1 Δ Ari(CS)]. Both ARIH1 mutants interacted with exogenously expressed 4EHP independently of their E3 ligase activity (revised Figure EV4H, lanes 4,5). Furthermore, a phospho-mimetic mutation on the Ariadne domain [ARIH1(S427D)] has been reported to relieve this autoinhibitory activity by opening the catalytic site (Reiter *et al*, 2022). Using this S427D mutant, we examined the requirement of E3 ligase activity for Sec61 β -ARIH1-4EHP complex formation. Wild-type ARIH1(S427S) required its E3 ligase activity to interact with endogenous 4EHP and Sec61 β (revised Figure EV4I, lanes 2,3), whereas the open-state mimic ARIH1(S427D) bound to these proteins independently of its E3 ligase activity (revised Figure EV4I, lanes 4,5). These findings suggest that the E3 ligase activity facilitates a conformational transition of ARIH1 from a closed (inactive) to an open (active) state, thereby enabling the recruitment of Sec61 β and 4EHP under ER stress conditions. Importantly, the increased accumulation of ERpQC substrate (^SNHK^{QQQ}) in ARIH1-deficient cells was suppressed by ARIH1 Δ Ari(CC), but not by the E3 ligase-inactive ARIH1 Δ Ari(CS), despite its ability to bind 4EHP (revised Figure 4F). Thus, the E3 ligase activity of ARIH1 appears to be essential not only for driving the conformational change required for 4EHP and Sec61 β binding, but also for repressing the translation of ERpQC substrates. Although the detailed mechanisms underlying the release of ARIH1

autoinhibition and modification of its E3 ligase substrates remain to be elucidated, we have discussed this point in the revised manuscript (p9 line 11–p10 line 10, revised Figure EV4H,I).

Additionally, treatment with thapsigargin and MG132 is known to induce phosphorylation of eIF2. Is this phosphorylation event involved in the Sec61 β -mediated translational suppression? What is the effect of ISRIB treatment on the translational activity regulated by Sec61 β ?

Re) We thank the reviewer for this valuable suggestion. We first examined whether ER stress-induced eIF2 α phosphorylation is associated with Sec61 β . In HepG2 cells, Sec61 β knockdown did not affect the extent of eIF2 α phosphorylation following thapsigargin and MG132 treatment compared with control knockdown (revised Figure EV3D), suggesting that ER stress-induced eIF2 α phosphorylation occurs independently of Sec61 β . Next, we used ISRIB, an inhibitor of the integrated stress response (ISR), to examine the involvement of eIF2 α phosphorylation-mediated translational attenuation in the Sec61 β -mediated translational repression of ERpQC substrates. Under ER stress conditions, ISRIB treatment increased the expression of ^SNHK^{QQQ} (revised Figure EV3E, lanes 1,3). Importantly, Sec61 β deficiency further enhanced the accumulation of ^SNHK^{QQQ}, but not that of ^CNHK^{QQQ} or BiP, in ISRIB-treated cells (revised Figure EV3E, lanes 3,4). Taken together, these results suggest that the PERK–eIF2 α pathway induces broad translational attenuation in the vicinity of the ER membrane under stress conditions, whereas Sec61 β -mediated translational regulation specifically suppresses the accumulation of ERpQC substrates through a mechanism distinct from the ISR. In other words, the ERpQC substrate may be subject to dual translational regulation mediated by the Sec61 β –ARIH1 axis in addition to the PERK–eIF2 α pathway. The corresponding results have been included, and we have described this point in detail in the revised manuscript (p6 line 29–p7 line 9, revised Figure EV3D,E).

Finally, can the authors use imaging approaches to visualize the interaction between Sec61 β , ARIH1, and 4EHP in cells?

Re) We appreciate the reviewer's insightful suggestion. To visualize the interactions among Sec61 β , ARIH1, and 4EHP in cells, we performed a proximity ligation assay (PLA) (Soderberg *et al*, 2006). Because suitable antibodies against ARIH1 and 4EHP were not available for the PLA, HepG2 cells were transfected to overexpress Flag-Sec61 β together with either HA-ARIH1 or HA-4EHP. The PLA was then carried out using anti-Flag and anti-HA antibodies to detect the close proximity of these two proteins. Under ER stress conditions, PLA signals between Flag-Sec61 β and either HA-ARIH1 (revised Figure EV4A,B) or HA-4EHP (revised Figure EV4C,D) were clearly observed along the reticular structures defined by co-expressed GFP-KDEL, an ER marker protein. Together with the co-immunoprecipitation data (revised Figure 3C,D,F), these findings suggest that Sec61 β interacts with ARIH1 and 4EHP on the ER membrane in ER-stressed cells. The corresponding results and description have been included in the revised manuscript (p8 lines 8–13, revised Figure EV4A–D).

Comments from referee #2:

Kadowaki et al. report on a novel role for Sec61 β in maintaining cytoplasmic homeostasis during ER stress via the cap-binding protein 4EHP-mediated selective translational repression of ERpQC substrates. The authors conclude that the translational repression is achieved through the E3 ligase ARIH1 and 4EHP as 4EHP recruitment blocks eIF4E dependent mRNA translation. The authors showed that loss of Sec61 β leads to increased synthesis of ERpQC substrates, impaired proteasome activity, and the accumulation of cytoplasmic aggresomes. The authors used a Zebrafish model to show that Sec61 β deficiency leads to motor dysfunction, which can be rescued by exogenous ARIH1 expression, pointing to the physiological relevance of this mechanism. The findings suggest that translational repression of ERpQC substrates by the Sec61 β -ARIH1-4EHP complex is a critical mechanism for coordinating ER and cytoplasmic proteostasis under stress conditions.

While the authors address an important aspect of ER homeostasis and offer a novel mechanism for maintaining proteostasis during ER stress, the paper suffers from major deficiencies including limited experimentation and technical shortcomings as follows:

1. The conclusions of the paper are tenuous. Based on the literature it is difficult to envisage a molecular mechanism by which the 4EHP-ARIH1 would replace the eIF4F at the 5' cap structure. eIF4F exhibits a strong interaction with cap because of its association with eIF4G which binds to the mRNA in the vicinity of the cap. 4EHP binds very weakly to the cap. It is not clear (model, Fig.7) how the complex 4EHP-ARIH1 binds to the cap even if eIF4F abundance is reduced 2-3 fold.

Re) We appreciate the reviewer's comments. As the reviewer pointed out, 4EHP has been reported to bind to cap analogs with lower affinity than eIF4E in *in vitro* experiments (Zuberek *et al*, 2007). However, 4EHP can compete with eIF4E through its interaction with the eIF4E-binding protein 4E-T (eIF4E transporter), which enhances the affinity of 4EHP for the 5' cap structure and thereby enables suppression of translation initiation (Chapat *et al*, 2017). Furthermore, several studies have demonstrated that 4EHP can be recruited to the 5' cap structure via the additional adaptor proteins GIGYF1 and GIGYF2, allowing for selective translational repression (Morita *et al*, 2012; Weber *et al*, 2020). Based on these findings, we propose that ARIH1 may act as an adaptor that stabilizes the interaction between 4EHP and the 5' cap structure of ERpQC substrate mRNAs on the ER membrane under stress conditions. We agree that it is important to verify whether 4EHP actually binds to the 5' cap structure of ERpQC substrate mRNAs. *Dual specificity phosphatase 6 (DUSP6)* mRNA, which encodes an ERK1/2 phosphatase, has been reported to be translationally repressed by 4EHP (Jafarnejad *et al*, 2018). To examine whether 4EHP associates with the 5' cap structure of ERpQC substrate mRNAs, we performed RNA immunoprecipitation (RIP) analysis using exogenously expressed Flag-4EHP in HepG2 cells and quantified the amounts of *ALAT* and *TTR* mRNAs bound to 4EHP under ER stress conditions. The amount of *DUSP6* mRNA bound to Flag-4EHP was not affected by ER stress (revised Figure EV4E), whereas the amounts of *ALAT* and *TTR* mRNAs bound to Flag-4EHP significantly increased under ER stress conditions (revised Figures 4D, EV4F). Moreover, Sec61 β knockdown suppressed the interaction between

4EHP and mRNAs of *α1AT* and *TTR* during ER stress (revised Figures 4E, EV4G). These results suggest that 4EHP is specifically recruited to the 5' cap structure of ERpQC substrate mRNAs via Sec61β under ER stress conditions, thereby contributing to their translational repression. However, the precise mechanism by which ARIH1 facilitates the association of 4EHP with the 5' cap structure and potentially displaces eIF4F remains unclear, and we cannot exclude the possibility that additional factors are involved. Therefore, our conclusion is limited to the possibility that 4EHP acts as a mediator in the Sec61β-ARIH1-dependent translational repression of ERpQC substrates. Based on this point, we have removed original Figure 3H and revised the schematic diagram accordingly (revised Figure 7). The corresponding results and descriptions have been included in the revised manuscript (p8 line 30–p9 line 10, p12 line 28–p13 line 6, revised Figures 4D,E, 7, EV4E–G). These points are also discussed in our response to reviewer 3's cross-comment.

Original Figure 3H > deleted

2. The quality of the WBs in many figures is inadequate, as many WB are composites of different gel electrophoresis runs and some lanes are spliced.

a. This is evident for Sec61β (main topic of the paper), Der1, Der2, and α1AT (NHKQQQ).

Re) We appreciate the reviewer's comments. We would like to clarify that the Western blot data were not composited from different gels and do not contain any spliced lanes. We are confident that the reviewer's concerns will be resolved by verifying the quality of the Western blots using the source data provided with the revised manuscript.

b. Figure 5F, the input for 4EHP shows two bands, whereas the IP displays only a single faint band, which is not explained.

Re) We apologize for the insufficient explanation regarding the two 4EHP bands in the input sample of the original Figure 5F. Two alternative splicing isoforms have been annotated for human 4EHP [245 a.a. (NM_004846.4) and 234 a.a. (NM_001276336.2)]. Indeed, the expression levels of both bands were reduced by 4EHP knockdown [revised Figure 4B,C (identical to original Figure 4B,C, respectively) and reference Figure 2, arrows]. Because the band representing 4EHP binding to Sec61β in the original Figure 5F was faint, we replaced the 1×Flag tag (original Figure 5F) with the 3×Flag tag (revised Figure 5F) to increase the amount of immunoprecipitated Flag-Sec61β and

reanalyzed the interaction to clarify which 4EHP isoform interacts with Sec61 β . The wild-type 3 \times Flag-Sec61 β , but not the Δ IDR mutant, interacted with both 4EHP isoforms (revised Figure 5F, arrows). The corresponding descriptions have been included in the revised manuscript (p33 lines 9–11 in legend of Figure 5F).

Reference Figure 2 (Asterisk indicates a non-specific band unrelated to 4EHP that appears due to overexpression of 3 \times Flag-Sec61 β and is not reduced by 4EHP siRNA transfection.)

Original Figure 5F

c. Fig 5C: The levels of ARIH1 in input increased in the flag-tagged (lines 3-5), also the amount of the Flag in the IP is more for lines 4 and 5, which could explain the increased interaction with ARIH1 in Tg and the Tg+MG. Also, the flag is not shown in the input.

Re) We appreciate the reviewer's comments. This comment most likely refers to Figure 3C, but please let us know if our understanding is incorrect. We agree with the reviewer's point and therefore reanalyzed the data using the 3 \times Flag-tagged *Sec61 β* knock-in cells (revised Figure 3C). We confirmed that the expression levels of ARIH1 and 3 \times Flag-Sec61 β in the input samples, as well as the amount of immunoprecipitated 3 \times Flag-Sec61 β , were comparable under all conditions. As a result, we observed increased binding between 3 \times Flag-Sec61 β and ARIH1 upon Tg and Tg+MG treatment. Figure 3C has been revised accordingly.

Original Figure 3C

3. The data presentation is confusing; for instance, Figure EV1 B is referenced before Fig, 1B, which data related only to endogenous Derlin-1 and not Derlin-2 or -3.

Re) We appreciate the reviewer's valuable comment. To validate the mass spectrometry results, we first analyzed the stress-dependent increase in the interactions of Derlin-1, -2, and -3 with Sec61 β , which were exogenously co-expressed in HEK293 cells, as shown in revised Figure 1B (identical to original Figure EV1B). ER stress-induced interactions of Derlin-2 and -3 with Sec61 β were clearly observed (revised Figure 1B). In contrast, exogenous Derlin-1 did not exhibit an ER stress-induced increase in binding, likely due to its strong interaction with exogenous Sec61 β under basal condition (revised Figure 1B). However, in HepG2 cells, an ER stress-induced increase in the endogenous interaction between Derlin-1 and Sec61 β was clearly observed in revised Figure 1C (identical to original Figure 1B). Since this result reflects a more physiological interaction, we initially presented only the data related to endogenous Derlin-1 in the main figure. However, to avoid potential confusion arising from the order of figure presentation, we have now included the exogenous interaction data of Derlin-1, -2, and -3 with Sec61 β in the main figure and have described these details in the revised manuscript (p4 lines 17–23, revised Figure 1B,C).

4. Important observations are not addressed. For example, Figure EV1C shows that eIF4A interacts with Derl2 specifically under ER stress (Tg+MG), but this result is not discussed in the text.

Re) We appreciate the reviewer's valuable comment. Following the reviewer's subsequent suggestion (point 5), we reanalyzed the interactions of eIF4A and eIF4E with Derlin-1, -2, and -3 under more physiological conditions. Similar to the interactions observed when Derlin-1, -2, or -3 was exogenously co-expressed with eIF4A1 (original Figure EV1C) or eIF4E (original Figure EV1D), all exogenous Derlins also interacted with endogenous eIF4A1 and eIF4E (revised Figure EV1B). The interactions of endogenous eIF4A1 with Derlin-2 and -3, as well as those of endogenous eIF4E with all Derlins, were enhanced by ER stress (revised Figure EV1B). Importantly, in HepG2 cells, endogenous interactions of eIF4A1 with Derlin-1 and -2, as well as of eIF4E with Derlin-2, were also observed in an ER stress-dependent manner (revised Figure EV1C,D). Collectively, these results suggest that Derlins physiologically interact with eIF4A and eIF4E. These results have been carefully described in the revised manuscript and original Figure EV1C and D have been revised accordingly (p4 lines 27–36, revised Figure EV1B–D).

Original Figure EV1C

Original Figure EV1D

5. Many experiments in the paper lack a rationale

a. P.4; The authors state "eIF4A1 and cap-binding protein eIF4E, which are components of the eIF4F complex together with the scaffold protein eIF4G, interacted with Derlins and these interactions were enhanced under the ER stress conditions (Fig. EV1C-E)". This is out of the blue: what is the rationale? Table EV2 lacks the fold-change value for eIF4A, and the authors do not explain why eIF4A was selected for further analysis.

Re) We appreciate the reviewer's thoughtful comments. As shown in Table EV1, eIF4A, but not eIF4E, is listed as a protein interacting with Derlin-1 (1.2-fold increase under ER stress) and Derlin-3 (1.1-fold increase under ER stress). We understand the reviewer's concern to relate to the insufficient explanation of our rationale for examining the interactions between eIF4E (which is not listed in Tables EV1 and EV2) and Derlins. If our understanding is incorrect, please let us know. Because eIF4E binds to the 5' cap structure as a component of the translation initiation complex eIF4F, together with eIF4A, we hypothesized that eIF4F might be recruited to Derlins. To test this hypothesis, we examined the interactions between eIF4E and Derlins. We fully agree that our previous explanation was insufficient and have now provided detailed clarification in the revised manuscript (p4 lines 23–36, revised Figure EV1B–E).

The IPs were performed only with exogenously produced proteins. This is acceptable when mutant proteins are also studied. Otherwise, they must show the interactions of endogenous proteins here and in the other experiments.

Re) We understand the reviewer's concern. We examined the endogenous interactions of Derlin-1 and -2 with eIF4A1 and eIF4E in HepG2 cells. ER stress-induced endogenous interactions between Derlin-1 or -2 and eIF4A1, as well as between Derlin-2 and eIF4E were observed (revised Figure EV1C,D). However, no endogenous interaction between Derlin-1 and eIF4E was detected, which may be due to the limited sensitivity of the antibody. Therefore, we analyzed the interactions between exogenous Derlin-1 and endogenous eIF4E (revised Figure EV1B), as well as between endogenous Derlin-1 and exogenous eIF4E [revised Figure EV1E (identical to original Figure EV1E)] and

confirmed that these interactions were enhanced under ER stress condition. These findings suggest that Derlin-1 and -2 physiologically interact with eIF4A1 and eIF4E, implying that Derlin-1 and -2 may recruit eIF4F (composed of eIF4A, eIF4E, and eIF4G) under ER stress conditions. The corresponding results and descriptions have been included in the revised manuscript (p4 lines 23–36, revised Figure EV1B–E). Unlike Derlin-1 and -2, Derlin-3 is expressed in only a limited range of cell types, and antibodies recognizing endogenous Derlin-3 protein were not available. However, the ER stress-induced enhancement of the interactions between exogenous Derlin-3 and endogenous eIF4A1 and eIF4E was clearly observed (revised Figure EV1B), suggesting that Derlin-3 may also recruit eIF4F under ER stress conditions. We have also described this point in the revised manuscript (p4 lines 33–34) and original Figure EV1C and D have been revised accordingly (revised Figure EV1B–E).

6. Figure 3E, the reason for the use of mouse liver cells is not explained. While including this model could add value, it requires proper introduction and contextualization.

7. The authors switch from using (Tg) and (MG132) to tunicamycin without justification.

Re) We appreciate the reviewer's insightful comments. The liver is a highly secretory organ characterized by active protein synthesis and extensive transport of proteins into the ER. Consequently, it is frequently used as an *in vivo* model for the physiological analysis of ER stress responses. It is well established that intraperitoneal injection of tunicamycin (Tun) induces acute ER stress in the mouse liver, and thus Tun has been extensively employed as an *in vivo* model for ER stress induction (Eura *et al*, 2020; Yamamoto *et al*, 2010; Zinszner *et al*, 1998). Accordingly, we administered Tun to mice to analyze the ER stress-dependent physiological interactions of Derlin-1 with Sec61 β and ARIH1. We have explained the rationale in the revised manuscript (p7 lines 31–36).

8. The rationale for the use of the swimming performance test is unclear. The authors neither explain why they chose to assess motor function specifically, nor do they reference any existing literature suggesting that Sec61 β affects motor abilities.

Re) We understand and appreciate the reviewer's concerns. At 3 days post-fertilization, a small number of Sec61 β -deficient larvae exhibited morphological abnormalities (revised Figure EV5E, arrowhead), and their body length tended to be shorter than that of water- or Sec61 β -atg-5mis MO-injected larvae (revised Figure EV5F). Although no previous studies have reported that Sec61 β deficiency or mutation affects motor function in zebrafish, careful behavioral observation revealed that Sec61 β -deficient larvae were unable to move their tails smoothly during swimming. Therefore, we quantitatively evaluated their motor function by measuring the head–tail angle. These points have been clarified and described in detail in the revised manuscript (p11 lines 22–29, revised Figure EV5E,F).

9. The introduction relies heavily on self-citations, with only three references not authored by the same group.

Re) We appreciate the reviewer's valuable comments. We agree that the introduction previously relied too

heavily on our own studies. In the revised manuscript, we have incorporated some references from other research groups to provide a broader and more balanced background in this field (p 2 line 29–p 3 line 7, Introduction). We hope that these additions help place our findings in a wider scientific context and improve the overall clarity and balance of the introduction.

Comments from referee #3:

ERpQC is a selective proteostasis mechanism by which certain proteins are re-routed from the ER translocon to the cytoplasmic degradation pathway. Previous studies by the authors established roles of Derlin-1, -2, -3 in ERpQC, but their mechanism of function had remained poorly understood. In this study, Kadowaki and colleagues show that Derlin-1, -2, -3 physically interact with Sec61b, and Sec61b specifically inhibits the accumulation of ERpQC substrates such as A1AT and TTR (without affecting secretory protein accumulation). Sec61b specifically regulates the translation of ERpQC substrates, and Sec61b physically interacts with the E3 ubiquitin ligase ARIH1. Derlins and ARIH1 regulate translation by recruiting 4EHP, a cap-binding protein that blocks eIF4E-mediated translation. The over-production of ERpQC in response to Sec61b loss impairs proteasome capacity. Consistently, knockdown of Sec61b in zebrafish causes motor impairment, which is suppressed by expressing ARIH1.

Overall, this is an interesting study that identifies new regulators of ERpQC. The study combines ex vivo and in vivo experiments to show detailed mechanisms and physiological functions. The experiments are well designed. The authors' ERpQC assays have internal controls of substrates that enter the secretory pathway (which are not affected by ERpQC impairment) and a higher molecular weight product that are specifically affected by ERpQC block. siRNA experiments have independent siRNAs and rescue experiments. The blots are of very high quality, and all major data are quantified. The quality of this manuscript is outstanding, and I have no major concerns.

Re) We sincerely thank the reviewer for the positive and encouraging comments. We are delighted that the study design, data quality, and the mechanistic insights into ERpQC were positively received and appreciated.

Cross-comments from referee 1:

I liked the work in the manuscript. If Reviewer #2 has had some concerns about the preparation of the images, this can be resolved by the provision of raw data, something that Nature publishing group always requests from the authors.

Regarding the rest of the comments of Reviewer #2, I believe that everyone deserves the benefit of the doubt. In my opinion, it would be fair to offer the opportunity to the authors to respond to the comments of

this Reviewer rigorously.

Re) We sincerely thank the reviewer for the constructive and supportive comments. We have provided all the requested raw data and have carefully addressed all concerns raised by Reviewer 2 through detailed explanations and, where necessary, additional experiments.

Cross-comments from referee 3:

It seems to me that Reviewer #2's specific points are less about the overall significance, but rather, about a few technical points that could be addressed by the authors. I would vote to give the authors a chance to respond. Below are some specific points:

Re) We sincerely appreciate the reviewer's thoughtful comments. We fully agree that the points raised by Reviewer 2 are important and have appropriately addressed them through additional experiments.

"Conclusions are tenuous." Reviewer #2 points out that 4EHP has a weaker affinity to the 5' caps of mRNAs compared to eIF4E in vitro, and based on this, disagrees with the authors' model that 4EHP can replace eIF4E at the 5' cap. But, I wonder if we can rule out the idea of 4EHP binding to 5' caps of certain mRNAs just because of the lower affinity measured in vitro. The published literature clearly supports the idea that 4EHP, a cap-binding protein, binds to a small number of mRNAs to exert a biological role. Perhaps the authors can strengthen their model by providing additional supporting data for 4EHP binding to ERpQC mRNAs.

Re) We sincerely appreciate the reviewer's valuable and insightful comments. As suggested, we examined whether 4EHP interacts with ERpQC substrate mRNAs by RNA immunoprecipitation (RIP) assay. The results of these experiments are presented in our response to Reviewer 2's comment 1. Briefly, the levels of ERpQC substrate mRNAs (*αLAT* and *TTR*), but not non-ERpQC substrate mRNA (*DUSP6*), bound to exogenously expressed Flag-4EHP increased in an ER stress-dependent manner (revised Figures 4D, EV4E,F). Furthermore, Sec61β knockdown reduced the binding of *αLAT* and *TTR* mRNAs to Flag-4EHP (revised Figures 4E, EV4G). Taken together, these findings suggest that 4EHP is recruited to the 5' cap structure of ERpQC substrate mRNAs via Sec61β under ER stress conditions, thereby contributing to their translational repression. The corresponding results and descriptions have been incorporated into the revised manuscript (p8 line 30–p9 line 10, revised Figures 4D,E, EV4E–G). However, the precise mechanism by which ARIH1 facilitates the association of 4EHP with the 5' cap structure and potentially displaces eIF4F remains unclear, and we do not yet have definitive evidence to exclude the involvement of additional factors. Therefore, our conclusion is limited to the possibility that 4EHP acts as a mediator in the Sec61β-ARIH1-dependent translational repression of ERpQC substrates. Based on this point, we have removed original Figure 3H and revised the schematic diagram accordingly (revised Figure 7). The corresponding results and explanations have

been added to the revised manuscript (p8 line 30–p9 line 10, p12 line 28–p13 line 6, revised Figures 4D,E, 7, EV4E–G).

"The quality of the WBs is inadequate, as many WBs are composites of different gel electrophoresis." I went over the Figures again, and I couldn't find signs of piecing together different gels in a single blot. Maybe I have a lower resolution image. Perhaps your journal could check through image analysis on this point.

Re) We appreciate the reviewer's careful comments. All immunoblotting images were obtained from single, continuous gels without any compositing or splicing. In accordance with the *EMBO Reports* data policy, the unprocessed, original blot images for all main figures have been provided as raw data.

Other points appear to be technical and addressable by the authors.

Re) We sincerely appreciate the reviewer's constructive suggestions. In the revised manuscript, we have thoroughly addressed all reviewer comments by providing additional data, clarifications, and detailed explanations.

References

- Chapat C, Jafarnejad SM, Matta-Camacho E, Hesketh GG, Gelbart IA, Attig J, Gkogkas CG, Alain T, Stern-Ginossar N, Fabian MR *et al* (2017) Cap-binding protein 4EHP effects translation silencing by microRNAs. *Proceedings of the National Academy of Sciences of the United States of America* 114: 5425-5430
- Duda DM, Olszewski JL, Schuermann JP, Kurinov I, Miller DJ, Nourse A, Alpi AF, Schulman BA (2013) Structure of HHARI, a RING-IBR-RING ubiquitin ligase: Autoinhibition of an Ariadne-family E3 and insights into ligation mechanism. *Structure* 21: 1030-1041
- Eura Y, Miyata T, Kokame K (2020) Derlin-3 Is Required for Changes in ERAD Complex Formation under ER Stress. *Int J Mol Sci* 21
- Jafarnejad SM, Chapat C, Matta-Camacho E, Gelbart IA, Hesketh GG, Arguello M, Garzia A, Kim SH, Attig J, Shapiro M *et al* (2018) Translational control of ERK signaling through miRNA/4EHP-directed silencing. *eLife* 7: e35034
- Morita M, Ler LW, Fabian MR, Siddiqui N, Mullin M, Henderson VC, Alain T, Fonseca BD, Karashchuk G, Bennett CF *et al* (2012) A novel 4EHP-GIGYF2 translational repressor complex is essential for mammalian development. *Mol Cell Biol* 32: 3585-3593
- Okumura F, Zou W, Zhang DE (2007) ISG15 modification of the eIF4E cognate 4EHP enhances cap structure-binding activity of 4EHP. *Genes Dev* 21: 255-260
- Reiter KH, Zelter A, Janowska MK, Riffle M, Shulman N, MacLean BX, Tamura K, Chambers MC, MacCoss MJ, Davis TN *et al* (2022) Cullin-independent recognition of HHARI substrates by a dynamic RBR catalytic domain. *Structure* 30: 1269-1284

- Soderberg O, Gullberg M, Jarvius M, Ridderstrale K, Leuchowius KJ, Jarvius J, Wester K, Hydbring P, Bahram F, Larsson LG *et al* (2006) Direct observation of individual endogenous protein complexes in situ by proximity ligation. *Nat Methods* 3: 995-1000
- Weber R, Chung MY, Keskeny C, Zinnall U, Landthaler M, Valkov E, Izaurralde E, Igreja C (2020) 4EHP and GIGYF1/2 Mediate Translation-Coupled Messenger RNA Decay. *Cell Rep* 33: 108262
- Yamamoto K, Takahara K, Oyadomari S, Okada T, Sato T, Harada A, Mori K (2010) Induction of liver steatosis and lipid droplet formation in ATF6alpha-knockout mice burdened with pharmacological endoplasmic reticulum stress. *Mol Biol Cell* 21: 2975-2986
- Zinszner H, Kuroda M, Wang X, Batchvarova N, Lightfoot RT, Remotti H, Stevens JL, Ron D (1998) CHOP is implicated in programmed cell death in response to impaired function of the endoplasmic reticulum. *Genes Dev* 12: 982-995
- Zuberek J, Kubacka D, Jablonowska A, Jemielity J, Stepinski J, Sonenberg N, Darzynkiewicz E (2007) Weak binding affinity of human 4EHP for mRNA cap analogs. *RNA* 13: 691-697

Dear Dr. Kadowaki,

Thank you for the submission of your revised manuscript. We have now received the enclosed reports from the referees that were asked to assess it. As you will see, referee 2 still has a few more suggestions that I would like you to address and incorporate before we can proceed with the official acceptance of your manuscript.

A few editorial requests will also need to be addressed:

- Please place the Disclosure statement to after the Acknowledgments.
- Affiliation 2 looks like a biotech company; employment in a biotech company should be stated in the Disclosure statement.
- The author credits need to be removed from the ms file. All credits need to be entered during online ms submission.
- In the FUNDING INFO, the following discrepancy should be resolved: JP24gm6410024 in the ms versus JP25gm6410024 in our online submission system.
- All main and EV FIGURES need to be uploaded as separate production quality Figure files.
- The EV Table Excel files have incorrect titles provided for the sheet name: Table S1-Table S3, this needs to be corrected in the sheet name. The blank Sheet 1 should be removed from Table EV2 file. Table EV1 should be a called Dataset EV1, and this name needs to be corrected in all places. Table EV2 could be a second sheet in Dataset EV1. Table EV3 should be called Dataset EV2 and all names need to be corrected for this file.
- The Appendix table should either be part of the Reagents & Tools table or should be made an EV table. The Appendix file can then be deleted.
- Manuscript in PDF and Merged PDF (uploaded as Related Manuscript Files) are not needed and should be removed.

* Figure Legends - Comments *

- Please note that the exact p values are not provided in the legends of figures 4c; 5a, c, d; 6g, h; EV 5a, please provide exact values as reasonable.
- Please indicate the statistical test used for data analysis in the legends of figures 1a; 3a; EV 1a; EV 3b
- Please note that the box plots need to be defined in terms of minima, maxima, centre, bounds of box and whiskers, and percentile in the legends of figures 6d, e
- Although 'n' is provided, please describe the nature of entity for 'n' in the legend of figure EV 3b

I would like to suggest some minor changes to the abstract that needs to be written in present tense. Please let me know whether you agree with this:

Disrupted proteostasis causes various degenerative diseases, and organelle homeostasis is therefore maintained by elaborate mechanisms. Endoplasmic reticulum (ER) stress-induced preemptive quality control (ERpQC) counteracts stress by reducing ER load through inhibiting the translocation of newly synthesized proteins into the ER for their rapid degradation in the cytoplasm. Here, we show that Sec61 β , a translocon component, prevents the overproduction of ERpQC substrates, allowing for their efficient degradation by the proteasome. Sec61 β inhibits the binding of translation initiation factor eIF4E to the mRNA 5' cap structure by recruiting E3 ligase ARIH1 and eIF4E-homologous protein 4EHP, resulting in selective translational repression of ERpQC substrates. Sec61 β deficiency causes overproduction of ERpQC substrates and reduces proteasome activity, leading to cytoplasmic aggresome formation. We also show that Sec61 β deficiency causes motor dysfunction in zebrafish, which is restored by exogenous ARIH1 expression. Collectively, translational repression of ERpQC substrates by the Sec61 β -ARIH1 complex contributes to maintain ER and cytoplasmic proteostasis.

EMBO press papers are accompanied online by A) a short (1-2 sentences) summary of the findings and their significance, B) 2-3 bullet points highlighting key results and C) a synopsis image that is exactly 550 pixels wide and 200-600 pixels high (the height is variable). The synopsis image should provide a sketch of the major findings, like a graphical abstract. Please note that text needs to be readable at the final size. Please send us this information along with the final manuscript.

Referee #1:

The authors addressed my criticisms with new data that explained previous concerns satisfactorily.

Referee #2:

The authors have addressed many, but not all, of my criticisms. Since I was the most critical reviewer, I think that the paper should be published after stressing its limitations. The authors alleviated my concerns that some panels are composites of different gel electrophoresis runs or contain spliced lanes. Based on the provided source files, there is no reason to suspect that gels being were combined or lanes spliced. While the authors have supplied the source data for each blot and could have presented higher-quality images, the available evidence suggests that the blot quality does not affect their conclusions, and the source files confirm that the gels are not composites. Nonetheless, the quality of many WBs remain suboptimal

Regarding the 4EHP bands in Fig. 5F, they fixed the problem by expressing 3x instead of 1x flag tag; they state the 2 bands correspond to alternative splicing isoforms of 4EHP [245 a.a. (NM_004846.4) and 234 a.a. (NM_001276336.2)]. This is rather baffling. A quick search of data bases reveals the possibility of many alternative splicing variants (ranging from 4 to 12). Why was only 1 additional to the canonical cDNA, chosen. Regardless of the interpretation of the band origins, the revised IP is of a better quality and clearly shows two bands. However, it should be mentioned that 2 bands appear in the published literature when just the canonical cDNA is used.

Regarding the introduction, the authors consent that it previously relied heavily on their own work and now cite additional studies. However, the introduction still lacks sufficient citation. For example, on page 2, a single reference is used to support multiple statements across lines 19-28, despite these points requiring separate citations.

We are deeply grateful for your renewed review. We have carefully addressed the remaining points raised and implemented additional revisions to enhance the clarity of the main text, figure legends, and citations. We believe these revisions have further strengthened the manuscript. Detailed responses to each comment are provided below.

Blue Arial font: comments from referees.

Black Times New Roman font: our responses.

Comments from referee #1:

The authors addressed my criticisms with new data that explained previous concerns satisfactorily.

Re) We appreciate the reviewer's positive assessment. We are pleased that the additional experiments and revisions have adequately addressed the reviewers' previous concerns.

Comments from referee #2:

The authors have addressed many, but not all, of my criticisms. Since I was the most critical reviewer, I think that the paper should be published after stressing its limitations.

Re) We appreciate your review of our revisions. In response to your previous comments, there are several points that we are currently unable to address experimentally. These have been documented in the manuscript as follows.

- Regarding the mechanism by which ARIH1 promotes the binding between the 5' cap structure of ERpQC substrate mRNAs and 4EHP, thereby displacing eIF4F, we acknowledge that our current experimental evidence has limitations. This point has been clearly stated in the Results section 'ARIH1 and 4EHP suppress the translation of ERpQC substrate mRNAs' (p9) and the Discussion section (p13).
- Regarding the two bands of 4EHP observed in association with Sec61 β , as indicated below, we are currently unable to determine whether these bands reflect splicing products or post-translationally modified forms. This point has been described in the revised Figure 5F legend (p34).
- Regarding endogenous interactions, it was not possible to demonstrate all stress-dependent endogenous interactions. Specifically, endogenous interactions between Derlin-1 and eIF4E and between Derlin-3 and either eIF4A1 or eIF4E, could not be detected, and these limitations are described in the Results section 'Derlins interact with translation-related proteins' (p4).

We hope that these explanations sufficiently clarify the limitations of our current study, as recommended.

The authors alleviated my concerns that some panels are composites of different gel electrophoresis runs or contain spliced lanes. Based on the provided source files, there is no reason to suspect that gels being were combined or lanes spliced. While the authors have supplied the source data for each blot and could have presented higher-quality images, the available evidence suggests that the blot quality does not affect their conclusions, and the source files confirm that the gels are not composites. Nonetheless, the quality of many WBs remain suboptimal.

Re) We are relieved by your assessment that, based on the source data provided, there is no reason to suspect that gels were combined, or lanes were spliced.

Regarding the 4EHP bands in Fig. 5F, they fixed the problem by expressing 3x instead of 1x flag tag; they state the 2 bands correspond to alternative splicing isoforms of 4EHP [245 a.a. (NM_004846.4) and 234 a.a. (NM_001276336.2)]. This is rather baffling. A quick search of data bases reveals the possibility of many alternative splicing variants (ranging from 4 to 12). Why was only 1 additional to the canonical cDNA, chosen. Regardless of the interpretation of the band origins, the revised IP is of a better quality and clearly shows two bands. However, it should be mentioned that 2 bands appear in the published literature when just the canonical cDNA is used.

Re) We thank the reviewer for pointing this out. In our initial revision, we interpreted the two bands of 4EHP as splicing isoforms. However, as the reviewer pointed out, we recognized that other possibilities, including post-translational modifications such as ubiquitination or ISGylation, should also be considered. This is based on the fact that two 4EHP bands have been observed in cells transfected with cDNA encoding 4EHP, as previously reported (Okumura *et al*, 2007; von Stechow *et al*, 2015). To avoid overinterpretation, we have revised the legend of Figure 5F (p34). We believe this revision provides a more accurate and balanced interpretation of the data.

Regarding the introduction, the authors consent that it previously relied heavily on their own work and now cite additional studies. However, the introduction still lacks sufficient citation. For example, on page 2, a single reference is used to support multiple statements across lines 19-28, despite these points requiring separate citations.

Re) We appreciate the reviewers' valuable comments. In the revised manuscript, we have incorporated several appropriate references to the relevant section (p2), ensuring that each scientific statement is supported by independent references. We believe these revisions address the reviewers' concerns and enhance the accuracy and overall completeness of the introduction.

References

- Okumura F, Zou W, Zhang DE (2007) ISG15 modification of the eIF4E cognate 4EHP enhances cap structure-binding activity of 4EHP. *Genes Dev* 21: 255-260
- von Stechow L, Typas D, Carreras Puigvert J, Oort L, Siddappa R, Pines A, Vrieling H, van de Water B, Mullenders LH, Danen EH (2015) The E3 ubiquitin ligase ARIH1 protects against genotoxic stress by initiating a 4EHP-mediated mRNA translation arrest. *Mol Cell Biol* 35: 1254-1268

Dr. Hisae Kadowaki
University of Miyazaki
medical Science
5200, Kihara, Kiyotake
Miyazaki 8891601
Japan

Dear Dr. Kadowaki,

I am very pleased to accept your manuscript for publication in the next available issue of EMBO reports. Thank you for your contribution to our journal.

You may qualify for financial assistance for your publication charges - either via a Springer Nature fully open access agreement or an EMBO initiative. Check your eligibility: <https://link.springer.com/journal/44319/how-to-publish-with-us>

Yours sincerely,

>>> Please note that it is EMBO Reports policy for the transcript of the editorial process (containing referee reports and your response letter) to be published as an online supplement to each paper. If you do NOT want this, you will need to inform the Editorial Office via email immediately. More information is available here: <https://link.springer.com/partners/embo-press/editorial-policies#Peer%20review>